# Probable Domain Generalization
# via Quantile Risk Minimization

**Cian Eastwood**[*1,2]  **Alexander Robey**[*3]  **Shashank Singh**[1]

**Julius von Kügelgen**[1,4]  **Hamed Hassani**[3]  **George J. Pappas**[3]  **Bernhard Schölkopf**[1]

[1] Max Planck Institute for Intelligent Systems, Tübingen
[2] University of Edinburgh  [3] University of Pennsylvania  [4] University of Cambridge

## Abstract

Domain generalization (DG) seeks predictors which perform well on unseen test distributions by leveraging data drawn from multiple related training distributions or domains. To achieve this, DG is commonly formulated as an average- or worst-case problem over the set of possible domains. However, predictors that perform well on average lack robustness while predictors that perform well in the worst case tend to be overly-conservative. To address this, we propose a new probabilistic framework for DG where the goal is to learn predictors that perform well *with high probability*. Our key idea is that distribution shifts seen during training should inform us of probable shifts at test time, which we realize by explicitly relating training and test domains as draws from the same underlying meta-distribution. To achieve probable DG, we propose a new optimization problem called *Quantile Risk Minimization* (QRM). By minimizing the $\alpha$-quantile of predictor's risk distribution over domains, QRM seeks predictors that perform well with probability $\alpha$. To solve QRM in practice, we propose the *Empirical QRM* (EQRM) algorithm and provide: (i) a generalization bound for EQRM; and (ii) the conditions under which EQRM recovers the causal predictor as $\alpha \to 1$. In our experiments, we introduce a more holistic quantile-focused evaluation protocol for DG, and demonstrate that EQRM outperforms state-of-the-art baselines on datasets from WILDS and DomainBed.

## 1 Introduction

Despite remarkable successes in recent years [1–3], machine learning systems often fail calamitously when presented with *out-of-distribution* (OOD) data [4–7]. Evidence of state-of-the-art systems failing in the face of distribution shift is mounting rapidly—be it due to spurious correlations [8–10], changing sub-populations [11–13], changes in location or time [14–16], or other naturally-occurring variations [17–23]. These OOD failures are particularly concerning in safety-critical applications such as medical imaging [24–28] and autonomous driving [29–31], where they represent one of the most significant barriers to the real-world deployment of machine learning systems [32–35].

Domain generalization (DG) seeks to improve a system's OOD performance by leveraging datasets from multiple environments or domains at training time, each collected under different experimental conditions [36–38] (see Fig. 1a). The goal is to build a predictor which exploits invariances across the training domains in the hope that these invariances also hold in related but distinct test domains [38–41]. To realize this goal, DG is commonly formulated as an average- [36, 42, 43] or worst-case [9, 44, 45] optimization problem over the set of possible domains. However, optimizing for average performance provably lacks robustness to OOD data [46], while optimizing for worst-domain performance tends to lead to overly-conservative solutions, with worst-case outcomes unlikely in practice [47, 48].

---

*Equal contribution. Correspondence to `c.eastwood@ed.ac.uk` or `arobey1@seas.upenn.edu`.
Code available at: https://github.com/cianeastwood/qrm

36th Conference on Neural Information Processing Systems (NeurIPS 2022).

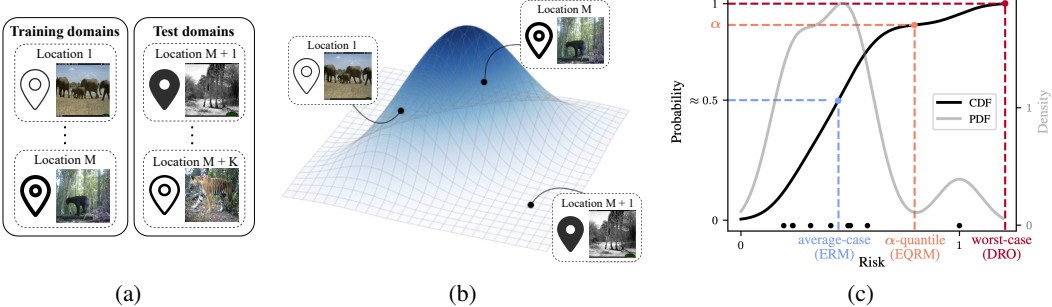

(a)                       (b)                       (c)

Figure 1: **Overview of Probable Domain Generalization and Quantile Risk Minimization.** (a) In domain generalization, training and test data are drawn from multiple related distributions or domains. For example, in the `iWildCam` dataset [50], which contains camera-trap images of animal species, the domains correspond to the different camera-traps which captured the images. (b) We relate training and test domains as draws from the same underlying (and often unknown) meta-distribution over domains $\mathbb{Q}$. (c) We consider a predictor's estimated risk distribution over training domains, naturally-induced by $\mathbb{Q}$. By minimizing the $\alpha$-quantile of this distribution, we learn predictors that perform well with high probability ($\approx \alpha$) rather than on average or in the worst case.

In this work, we argue that DG is neither an average-case nor a worst-case problem, but rather a probabilistic one. To this end, we propose a probabilistic framework for DG, which we call *Probable Domain Generalization* (§ 3), wherein the key idea is that distribution shifts seen during training should inform us of *probable* shifts at test time. To realize this, we explicitly relate training and test domains as draws from the same underlying meta-distribution (Fig. 1b), and then propose a new optimization problem called *Quantile Risk Minimization* (QRM). By minimizing the $\alpha$-quantile of predictor's risk distribution over domains (Fig. 1c), QRM seeks predictors that perform well *with high probability* rather than on average or in the worst case. In particular, QRM leverages the key insight that this $\alpha$-quantile is an upper bound on the test-domain risk which holds with probability $\alpha$, meaning that $\alpha$ is an interpretable conservativeness-hyperparameter with $\alpha = 1$ corresponding to the worst-case setting.

To solve QRM in practice, we introduce the *Empirical QRM* (EQRM) algorithm (§ 4). Given a predictor's empirical risks on the training domains, EQRM forms an estimated risk distribution using kernel density estimation (KDE, [49]). Importantly, KDE-smoothing ensures a right tail that extends beyond the largest training risk (see Fig. 1c), with this risk "extrapolation" [41] unlocking *invariant prediction* for EQRM (§ 4.1). We then provide theory for EQRM (§ 4.2, § 4.3) and demonstrate empirically that EQRM outperforms state-of-the-art baselines on real and synthetic data (§ 6).

**Contributions.** To summarize our main contributions:

- *A new probabilistic perspective and objective for DG:* We argue that predictors should be trained and tested based on their ability to perform well *with high probability*. We then propose Quantile Risk Minimization for achieving this *probable* form of domain generalization (§ 3).

- *A new algorithm:* We propose the EQRM algorithm to solve QRM in practice and ultimately learn predictors that generalize with probability $\alpha$ (§ 4). We then provide several analyses of EQRM:

  - *Learning theory:* We prove a uniform convergence bound, meaning the empirical $\alpha$-quantile risk tends to the population $\alpha$-quantile risk given sufficiently many domains and samples (Thm. 4.1).

  - *Causality.* We prove that EQRM learns predictors with invariant risk as $\alpha \to 1$ (Prop. 4.3), then provide the conditions under which this is sufficient to recover the causal predictor (Thm. 4.4).

  - *Experiments:* We demonstrate that EQRM outperforms state-of-the-art baselines on several standard DG benchmarks, including `CMNIST` [9] and datasets from WILDS [12] and DomainBed [38], and highlight the importance of assessing the tail or *quantile performance* of DG algorithms (§ 6).

## 2   Background: Domain generalization

**Setup.** In domain generalization (DG), predictors are trained on data drawn from multiple related training distributions or *domains* and then evaluated on related but unseen test domains. For example, in the `iWildCam` dataset [50], the task is to classify animal species in images, and the domains correspond to the different camera-traps which captured the images (see Fig. 1a). More formally, we consider datasets $D^e = \{(x_i^e, y_i^e)\}_{i=1}^{n_e}$ collected from $m$ different training domains or *environments* $\mathcal{E}_{\text{tr}} := \{e_1, \ldots, e_m\}$, with each dataset $D^e$ containing data pairs $(x_i^e, y_i^e)$ sampled i.i.d. from

$\mathbb{P}(X^e, Y^e)$. Then, given a suitable function class $\mathcal{F}$ and loss function $\ell$, the goal of DG is to learn a predictor $f \in \mathcal{F}$ that generalizes to data drawn from a larger set of all possible domains $\mathcal{E}_{\text{all}} \supset \mathcal{E}_{\text{tr}}$.

**Average case.** Letting $\mathcal{R}^e(f)$ denote the statistical risk of $f$ in domain $e$, and $\mathbb{Q}$ a distribution over the domains in $\mathcal{E}_{\text{all}}$, DG was first formulated [36, 37] as the following average-case problem:

$$\min_{f \in \mathcal{F}} \mathbb{E}_{e \sim \mathbb{Q}} \mathcal{R}^e(f) \qquad \text{where} \qquad \mathcal{R}^e(f) := \mathbb{E}_{\mathbb{P}(X^e, Y^e)}[\ell(f(X^e), Y^e)]. \qquad (2.1)$$

**Worst case.** Since predictors that perform well *on average* provably lack robustness [46], i.e. they can perform quite poorly on large subsets of $\mathcal{E}_{\text{all}}$, subsequent works [9, 22, 41, 44, 45, 51] have sought robustness by formulating DG as the following *worst-case* problem:

$$\min_{f \in \mathcal{F}} \max_{e \in \mathcal{E}_{\text{all}}} \mathcal{R}^e(f). \qquad (2.2)$$

As we only have access to data from a finite subset of $\mathcal{E}_{\text{all}}$ during training, solving (2.2) is not just challenging but in fact impossible [41, 52, 53] without restrictions on how the domains may differ.

**Causality and invariance in DG.** Causal works on DG [9, 41, 53–55] describe domain differences using the language of causality and the notion of *interventions* [56, 57]. In particular, they assume all domains share the same underlying *structural causal model* (SCM) [56], with different domains corresponding to different interventions (see Appendix A.1 for formal definitions and a simple example). Assuming the mechanism of $Y$ remains fixed or invariant but all $X$s may be intervened upon, recent works have shown that only the causal predictor has invariant: (i) predictive distributions [54], coefficients [9] or risks [41] across domains; and (ii) generalizes to arbitrary interventions on the $X$s [9, 54, 55]. These works then leverage some form of invariance across domains to discover causal relationships which, through the invariant mechanism assumption, generalize to new domains.

## 3 Quantile Risk Minimization

In this section we introduce *Quantile Risk Minimization* (QRM) for achieving *Probable Domain Generalization*. The core idea is to replace the worst-case perspective of (2.2) with a probabilistic one. This approach is founded on a great deal of work in classical fields such as control theory [58, 59] and smoothed analysis [60], wherein approaches that yield high-probability guarantees are used in place of worst-case approaches in an effort to mitigate conservatism and computational limitations. This mitigation is of particular interest in domain generalization since generalizing to arbitrary domains is impossible [41, 52, 53]. Thus, motivated by this classical literature, our goal is to obtain predictors that are robust *with high probability* over domains drawn from $\mathcal{E}_{\text{all}}$, rather than in the worst case.

**A distribution over environments.** We start by assuming the existence of a probability distribution $\mathbb{Q}(e)$ over the set of all environments $\mathcal{E}_{\text{all}}$. For instance, in the context of medical imaging, $\mathbb{Q}$ could represent a distribution over potential changes to a hospital's setup or simply a distribution over candidate hospitals. Given that such a distribution $\mathbb{Q}$ exists[2], we can think of the risk $\mathcal{R}^e(f)$ as a *random variable* for each $f \in \mathcal{F}$, where the randomness is engendered by the draw of $e \sim \mathbb{Q}$. This perspective gives rise to the following analogue of the optimization problem in (2.2):

$$\min_{f \in \mathcal{F}} \operatorname*{ess\,sup}_{e \sim \mathbb{Q}} \mathcal{R}^e(f) \quad \text{where} \quad \operatorname*{ess\,sup}_{e \sim \mathbb{Q}} \mathcal{R}^e(f) = \inf \left\{ t \geq 0 : \Pr_{e \sim \mathbb{Q}} \{ \mathcal{R}^e(f) \leq t \} = 1 \right\} \quad (3.1)$$

Here, ess sup denotes the *essential-supremum* operator from measure theory, meaning that for each $f \in \mathcal{F}$, $\operatorname{ess\,sup}_{\mathbb{Q}} \mathcal{R}^e(f)$ is the least upper bound on $\mathcal{R}^e(f)$ that holds for almost every $e \sim \mathbb{Q}$. In this way, the ess sup in (3.1) is the measure-theoretic analogue of the max operator in (2.2), with the subtle but critical difference being that the ess sup in (3.1) can neglect domains of measure zero under $\mathbb{Q}$. For example, for discrete $\mathbb{Q}$, (3.1) ignores domains which are impossible (i.e. have probability zero) while (2.2) does not, laying the foundation for ignoring domains which are *improbable*.

**High-probability generalization.** Although the minimax problem in (3.1) explicitly incorporates the distribution $\mathbb{Q}$ over environments, this formulation is no less conservative than (2.2). Indeed, in many cases, (3.1) is equivalent to (2.2); see Appendix B for details. Therefore, rather than considering the worst-case problem in (3.1), we propose the following generalization of (3.1) which requires that predictors generalize with probability $\alpha$ rather than in the worst-case:

$$\min_{f \in \mathcal{F}, t \in \mathbb{R}} t \qquad \text{subject to} \qquad \Pr_{e \sim \mathbb{Q}} \{ \mathcal{R}^e(f) \leq t \} \geq \alpha \qquad (3.2)$$

---

[2]As $\mathbb{Q}$ is often unknown, our analysis does not rely on using an explicit expression for $\mathbb{Q}$.

The optimization problem in (3.2) formally defines what we mean by *Probable Domain Generalization*. In particular, we say that *a predictor $f$ generalizes with risk $t$ at level $\alpha$* if $f$ has risk at most $t$ with probability at least $\alpha$ over domains sampled from $\mathbb{Q}$. In this way, the conservativeness parameter $\alpha$ controls the strictness of generalizing to unseen domains.

**A distribution over risks.** The optimization problem presented in (3.2) offers a principled formulation for generalizing to unseen distributional shifts governed by $\mathbb{Q}$. However, $\mathbb{Q}$ is often unknown in practice and its support $\mathcal{E}_{\text{all}}$ may be high-dimensional or challenging to define [22]. While many previous works have made progress by limiting the scope of possible shift types over domains [19, 22, 45], in practice, such structural assumptions are often difficult to justify and impossible to test. For this reason, we start our exposition of QRM by offering an alternative view of (3.2) which elucidates how a predictor's *risk distribution* plays a central role in achieving probable domain generalization.

To begin, note that for each $f \in \mathcal{F}$, the distribution over domains $\mathbb{Q}$ naturally induces[3] a distribution $\mathbb{T}_f$ over the risks in each domain $\mathcal{R}^e(f)$. In this way, rather than considering the randomness of $\mathbb{Q}$ in the often-unknown and (potentially) high-dimensional space of possible shifts (Fig. 1b), one can consider it in the real-valued space of risks (Fig. 1c). This is analogous to statistical learning theory, where the analysis of convergence of empirical risk minimizers (i.e., of functions) is substituted by that of a weaker form of convergence, namely that of scalar risk functionals—a crucial step for VC theory [61]. From this perspective, the statistics of $\mathbb{T}_f$ can be thought of as capturing the sensitivity of $f$ to different environmental shifts, summarizing the effect of different intervention types, strengths, and frequencies. To this end, (3.2) can be equivalently rewritten in terms of the risk distribution $\mathbb{T}_f$ as follows:

$$\min_{f \in \mathcal{F}} F_{\mathbb{T}_f}^{-1}(\alpha) \quad \text{where} \quad F_{\mathbb{T}_f}^{-1}(\alpha) := \inf \left\{ t \in \mathbb{R} : \Pr_{R \sim \mathbb{T}_f} \{R \leq t\} \geq \alpha \right\}. \quad \text{(QRM)}$$

Here, $F_{\mathbb{T}_f}^{-1}(\alpha)$ denotes the inverse CDF (or quantile[4]) function of the risk distribution $\mathbb{T}_f$. By means of this reformulation, we elucidate how solving (QRM) amounts to finding a predictor with minimal $\alpha$-quantile risk. That is, (QRM) requires that a predictor $f$ satisfy the probabilistic constraint for at least an $\alpha$-fraction of the risks $R \sim \mathbb{T}_f$, or, equivalently, for an $\alpha$-fraction of the environments $e \sim \mathbb{Q}$. In this way, $\alpha$ can be used to interpolate between typical ($\alpha = 0.5$, median) and worst-case ($\alpha = 1$) problems in an interpretable manner. Moreover, if the mean and median of $\mathbb{T}_f$ coincide, $\alpha = 0.5$ gives an average-case problem, with (QRM) recovering several notable objectives for DG as special cases.

**Proposition 3.1.** *For $\alpha = 1$, (QRM) is equivalent to the worst-case problem of (3.1). For $\alpha = 0.5$, it is equivalent to the average-case problem of (2.1) if the mean and median of $\mathbb{T}_f$ coincide $\forall f \in \mathcal{F}$:*

$$\min_{f \in \mathcal{F}} \mathbb{E}_{R \sim \mathbb{T}_f} R = \min_{f \in \mathcal{F}} \mathbb{E}_{e \sim \mathbb{Q}} \mathcal{R}^e(f) \quad (3.3)$$

**Connection to DRO.** While fundamentally different in terms of objective and generalization capabilities (see § 4), we draw connections between QRM and distributionally robust optimization (DRO) in Appendix F by considering an alternative problem which optimizes the *superquantile*.

## 4 Algorithms for Quantile Risk Minimization

We now introduce the *Empirical QRM* (EQRM) algorithm for solving (QRM) in practice, akin to Empirical Risk Minimization (ERM) solving the Risk Minimization (RM) problem [63].

### 4.1 From QRM to Empirical QRM

In practice, given a predictor $f$ and its empirical risks $\hat{\mathcal{R}}^{e_1}(f), \ldots, \hat{\mathcal{R}}^{e_m}(f)$ on the $m$ training domains, we must form an *estimated* risk distribution $\widehat{\mathbb{T}}_f$. In general, given no prior knowledge about the form of $\mathbb{T}_f$ (e.g. Gaussian), we use *kernel density estimation* (KDE, [49, 64]) with Gaussian kernels and either the Gaussian-optimal rule [65] or Silverman's rule-of-thumb [65] for bandwidth selection. Fig. 1c depicts the PDF and CDF for 10 training risks when using Silverman's rule-of-thumb. Armed

---

[3]$\mathbb{T}_f$ can be formally defined as the push-forward measure of $\mathbb{Q}$ through the risk functional $\mathcal{R}^e(f)$; see App. B.

[4]In financial optimization, when concerned with a distribution over potential losses, the $\alpha$-quantile value is known as the *value at risk* (VaR) at level $\alpha$ [62].

with a predictor's estimated risk distribution $\widehat{\mathbb{T}}_f$, we can approximately solve (QRM) using the following empirical analogue:

$$\min_{f \in \mathcal{F}} F_{\widehat{\mathbb{T}}_f}^{-1}(\alpha) \qquad (4.1)$$

Note that (4.1) depends only on known quantities so we can compute and minimize it in practice, as detailed in Alg. 1 of Appendix E.1.

**Smoothing permits risk extrapolation.** Fig. 2 compares the KDE-smoothed CDF (black) to the unsmoothed empirical CDF (gray). As shown, the latter places zero probability mass on risks greater than our largest training risk, thus implicitly assuming that test risks cannot be larger than training risks. In contrast, the KDE-smoothed CDF permits "risk extrapolation" [41] since its right tail extends beyond our largest training risk, with the estimated $\alpha$-quantile risk going to infinity as $\alpha \to 1$ (when kernels have full support). Note that different bandwidth-selection methods encode different assumptions about right-tail heaviness and thus about projected OOD risk. In § 4.3, we discuss how, as $\alpha \to 1$, this KDE-smoothing allows EQRM to learn predictors with invariant risk over domains. In Appendix C, we discuss different bandwidth-selection methods for EQRM.

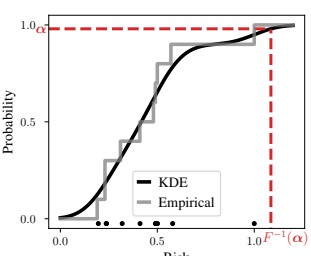

Figure 2: Risk CDFs.

## 4.2 Theory: Generalization bound

We now give a simplified version of our main generalization bound—Thm. D.1—which states that, given sufficiently many domains and samples, the empirical $\alpha$-quantile risk is a good estimate of the population $\alpha$-quantile risk. In contrast to previous results for DG, we bound the *proportion of test domains* for which a predictor performs well, rather than the average error [36, 42], and make no assumptions about the shift type, e.g. covariate shift [37]. The full version, stated and proved in Appendix D, provides specific finite-sample bounds on $\epsilon_1$ and $\epsilon_2$ below, depending on the hypothesis class $\mathcal{F}$, the empirical estimator $F_{\widehat{\mathbb{T}}_f}^{-1}(\alpha)$, and the assumptions on the possible risk profiles of hypotheses $f \in \mathcal{F}$.

**Theorem 4.1** (Simplified form of Thm. D.1, uniform convergence)**.** *Given m domains and n samples in each, then with high probability over the training data,*

$$\sup_{f \in \mathcal{F}} \left| F_{\mathbb{T}_f}^{-1}(\alpha - \epsilon_2) - F_{\widehat{\mathbb{T}}_f}^{-1}(\alpha) \right| \le \epsilon_1, \qquad (4.2)$$

*where $\epsilon_1 \to 0$ as $n \to \infty$ and $\epsilon_2 \to 0$ as $m \to \infty$.*

While many domains are required for this to bound be tight, i.e. for $\alpha$ to *precisely* estimate the true quantile, our empirical results in § 6 demonstrate that EQRM performs well in practice given only a few domains. In such settings, $\alpha$ still controls conservativeness, but with a less precise interpretation.

## 4.3 Theory: Causal recovery

We now prove that EQRM can recover the causal predictor in two parts. First, we prove that, as $\alpha \to 1$, EQRM learns a predictor with minimal, invariant risk over domains. For Gaussian estimators of the risk distribution $\mathbb{T}_f$, some intuition can be gained from Eq. (A.3) of Appendix A.2.1, noting that $\alpha \to 1$ puts increasing weight on the sample standard deviation of risks over domains $\hat{\sigma}_f$, eventually forcing it to zero. For kernel density estimators, a similar intuition applies so long as the bandwidth has a certain dependence on $\hat{\sigma}_f$, as detailed in Appendix A.2.2. Second, we prove that learning such a *minimal invariant-risk predictor* is sufficient to recover the causal predictor under weaker assumptions than Peters et al. [54] and Krueger et al. [41]. Together, these two parts provide the conditions under which EQRM successfully performs "causal recovery", i.e., correctly recovers the true causal coefficients in a linear causal model of the data.

**Definition 4.2.** A predictor $f$ is said to be an *invariant-risk predictor* if its risk is equal almost surely across domains (i.e., $\text{Var}_{e \sim \mathbb{Q}}[\mathcal{R}^e(f)] = 0$). A predictor is said to be a *minimal invariant-risk predictor* if it achieves the minimal possible risk across all possible invariant-risk predictors.

**Proposition 4.3** (EQRM learns a minimal invariant-risk predictor as $\alpha \to 1$, informal version of Props. A.4 and A.5)**.** *Assume: (i) $\mathcal{F}$ contains an invariant-risk predictor with finite training risks;*

*and (ii) no arbitrarily-negative training risks. Then, as $\alpha \rightarrow 1$, Gaussian and kernel EQRM predictors (the latter with certain bandwidth-selection methods) converge to minimal invariant-risk predictors.*

Props. A.4 and A.5 are stated and proved in Appendices A.2.1 and A.2.2 respectively. In addition, for the special case of Gaussian estimators of $\mathbb{T}_f$, Appendix A.2.1 relates our $\alpha$ parameter to the $\beta$ parameter of VREx [41, Eq. 8]. We next specify the conditions under which learning such a minimal invariant-risk predictor is sufficient to recover the causal predictor.

**Theorem 4.4** (The causal predictor is the only minimal invariant-risk predictor). *Assume that: (i) $Y$ is generated from a linear SEM, $Y = \beta^\mathsf{T} X + N$, with $X$ observed and coefficients $\beta \in \mathbb{R}^d$; (ii) $\mathcal{F}$ is the class of linear predictors, indexed by $\hat{\beta} \in \mathbb{R}^d$; (iii) the loss $\ell$ is squared-error; (iv) the risk $\mathbb{E}[(Y - \beta^T X)^2]$ of the causal predictor $\beta$ is invariant across domains; and (v) the system of equations*

$$
\begin{aligned}
0 \geq & \, x^\mathsf{T} \mathrm{Cov}_{X \sim e_1}(X, X) x + 2 x^\mathsf{T} \mathrm{Cov}_{N, X \sim e_1}(X, N) \\
= & \cdots \\
= & \, x^\mathsf{T} \mathrm{Cov}_{X \sim e_m}(X, X) x + 2 x^\mathsf{T} \mathrm{Cov}_{N, X \sim e_m}(X, N)
\end{aligned}
\tag{4.3}
$$

*has the unique solution $x = 0$. If $\hat{\beta}$ is a minimal invariant-risk predictor, then $\hat{\beta} = \beta$.*

**Assumptions (i–iii).** The assumptions that $Y$ is drawn from a linear structural equation model (SEM) and that the loss is squared-error, while restrictive, are needed for all comparable causal recovery results [41, 54]. In fact, these assumptions are weaker than both Peters et al. [54, Thm. 2] (assume a linear *Gaussian* SEM for $X$ *and* $Y$) and Krueger et al. [41, Thm. 1] (assume a linear SEM for $X$ *and* $Y$).

**Assumption (iv).** The assumption that the risk of the causal predictor is invariant across domains, often called *domain homoskedasticity* [41], is necessary for any method inferring causality from the *invariance of risks* across domains. For methods based on the *invariance of functions*, namely the conditional mean $\mathbb{E}[Y|\mathrm{Pa}(Y)]$ [9, 66], this assumption is not required. Appendix G.1.2 compares methods based on invariant risks and to those based on invariant functions.

**Assumption (v).** In contrast to both Peters et al. and Krueger et al., we do not require specific types of interventions on the covariates. Instead, we require that a more general condition be satisfied, namely that the system of $d$-variate quadratic equations in (4.3) has a unique solution. Intuitively, $\mathrm{Cov}(X, X)$ captures how correlated the covariates are and ensures they are sufficiently uncorrelated to distinguish each of their influences on $Y$, while $\mathrm{Cov}(X, N)$ captures how correlated descendant covariates are with $Y$ (via $N$). Together, these terms capture the idea that *predicting $Y$ from the causal covariates must result in the minimal invariant-risk*: the first inequality ensures the risk is *minimal* and the subsequent $m - 1$ equalities that it is *invariant*. While this generality comes at the cost of abstraction, Appendix A.2.3 provides several concrete examples with different types of interventions in order to aid understanding and illustrate how this condition generalizes existing causal-recovery results based on invariant risks [41, 54]. Appendix A.2.3 also provides a proof of Thm. 4.4 and further discussion.

## 5 Related work

**Robust optimization in DG.** Throughout this paper, we follow an established line of work (see e.g., [9, 41, 51]) which formulates the DG problem through the lens of robust optimization [44]. To this end, various algorithms have been proposed for solving constrained [22] and distributionally robust [45] variants of the worst-case problem in (2.2). Indeed, this robust formulation has a firm foundation in the broader machine learning literature, with notable works in adversarial robustness [67–71] and fair learning [72, 73] employing similar formulations. Unlike these past works, we consider a robust but non-adversarial formulation for DG, where predictors are trained to generalize with high probability rather than in the worst case. Moreover, the majority of this literature—both within and outside of DG—relies on specific structural assumptions (e.g. covariate shift) on the types of possible interventions or perturbations. In contrast, we make the weaker and more flexible assumption of i.i.d.-sampled domains, which ultimately makes use of the observed domain-data to determine the types of shifts that are *probable*. We further discuss this important difference in § 7.

**Other approaches to DG.** Outside of robust optimization, many algorithms have been proposed for the DG setting which draw on insights from a diverse array of fields, including approaches based on tools from meta-learning [40, 43, 74–76], kernel methods [77, 78], and information theory [51]. Also prominent are works that design regularizers to generalize OOD [79–81] and works that seek

domain-invariant representations [82–84]. Many of these works employ hyperparameters which are difficult to interpret, which has no doubt contributed to the well-established model-selection problem in DG [38]. In contrast, in our framework, $\alpha$ can be easily interpreted in terms of quantiles of the risk distribution. In addition, many of these works do not explicitly relate the training and test domains, meaning they lack theoretical results in the non-linear setting (e.g. [9, 41, 43, 85]). For those which do, they bound either average error over test domains [36, 42, 86] or worst-case error under specific shift types (e.g. covariate [22]). As argued above, the former lacks robustness while the latter can be both overly-conservative and difficult to justify in practice, where shift types are often unknown.

**High-probability generalization.** As noted in § 3, relaxing worst-case problems in favor of probabilistic ones has a long history in control theory [58, 59, 87–89], operations research [90], and smoothed analysis [60]. Recently, this paradigm has been applied to several areas of machine learning, including perturbation-based robustness [91, 92], fairness [93], active learning [94], and reinforcement learning [95, 96]. However, it has not yet been applied to domain generalization.

**Quantile minimization.** In financial optimization, the quantile and superquantile functions [62, 97, 98] are central to the literature surrounding portfolio risk management, with numerous applications spanning banking regulations and insurance policies [99, 100]. In statistical learning theory, several recent papers have derived uniform convergence guarantees in terms of alternative risk functionals besides expected risk [94, 101–103]. These results focus on functionals that can be written in terms of expectations over the loss distribution (e.g., the superquantile). In contrast, our uniform convergence guarantee (Theorem D.1) shows uniform convergence of the quantile function, which *cannot* be written as such an expectation; this necessitates stronger conditions to obtain uniform convergence, which ultimately suggest regularizing the estimated risk distribution (e.g. by kernel smoothing).

**Invariant prediction and causality.** Early work studied the problem of learning from multiple cause-effect datasets that share a functional mechanism but differ in noise distributions [39]. More generally, given (data from) multiple distributions, one can try to identify components which are stable, robust, or *invariant*, and find means to transfer them across problems [104–108]. As discussed in § 2, recent works have leveraged different forms of invariance across domains to discover causal relationships which, under the invariant mechanism assumption [57], generalize to new domains [9, 41, 54, 55, 109–111]. In particular, VREx [41] leveraged *invariant risks* (like EQRM) while IRM [9] leveraged *invariant functions* or coefficients—see Appendix G.1.2 for a detailed comparison of these approaches.

## 6 Experiments

We now evaluate our EQRM algorithm on synthetic datasets (§ 6.1), real-world datasets from WILDS (§ 6.2), and few-domain datasets from DomainBed (§ 6.3). Appendix G reports further results, while Appendix E reports further experimental details.

### 6.1 Synthetic datasets

**Linear regression.** We first consider a linear regression dataset based on the following linear SCM:

$$X_1 \leftarrow N_1, \qquad Y \leftarrow X_1 + N_Y, \qquad X_2 \leftarrow Y + N_2,$$

with $N_j \sim \mathcal{N}(0, \sigma_j^2)$. Here we have two features: one cause $X_1 = X_{\text{cause}}$ and one effect $X_2 = X_{\text{effect}}$ of $Y$. By fixing $\sigma_1^2 = 1$ and $\sigma_Y^2 = 2$ across domains but sampling $\sigma_2 \sim \text{LogNormal}(0, 0.5)$, we create a dataset in which $X_2$ is more predictive of $Y$ than $X_1$ but less stable. Importantly, as we know the true distribution over domains $\mathbb{Q}(e) = \text{LogNormal}(\sigma_2^e; 0, 0.5)$, we know the true risk quantiles. Fig. 3 depicts results for different $\alpha$'s with $m = 1000$ domains and $n = 200000$ samples in each, using the mean-squared-error (MSE) loss. Here we see that: **A:** for each true quantile (x-axis), the corresponding $\alpha$ has the lowest risk (y-axis), confirming that the empirical $\alpha$-quantile risk is a good estimate of the population $\alpha$-quantile risk; **B:** As $\alpha \to 1$, the estimated risk distribution of $f_\alpha$ approaches an invariant (or Dirac delta) distribution centered on the risk of the causal predictor; **C:** the regression coefficients approach those of the causal predictor as $\alpha \to 1$, trading predictive performance for robustness; and **D:** reducing the number of domains $m$ reduces the accuracy of the estimated $\alpha$-quantile risks. In Appendix G.1, we additionally: (i) depict the risk CDFs corresponding to plot **B** above, and discuss how they depict the predictors' risk-robustness curves (G.1.1); and (ii) discuss the solutions of EQRM on datasets in which $\sigma_1^2$, $\sigma_2^2$ and/or $\sigma_Y^2$ change over domains, compared to existing invariance-seeking algorithms like IRM [9] and VREx [41] (G.1.2).

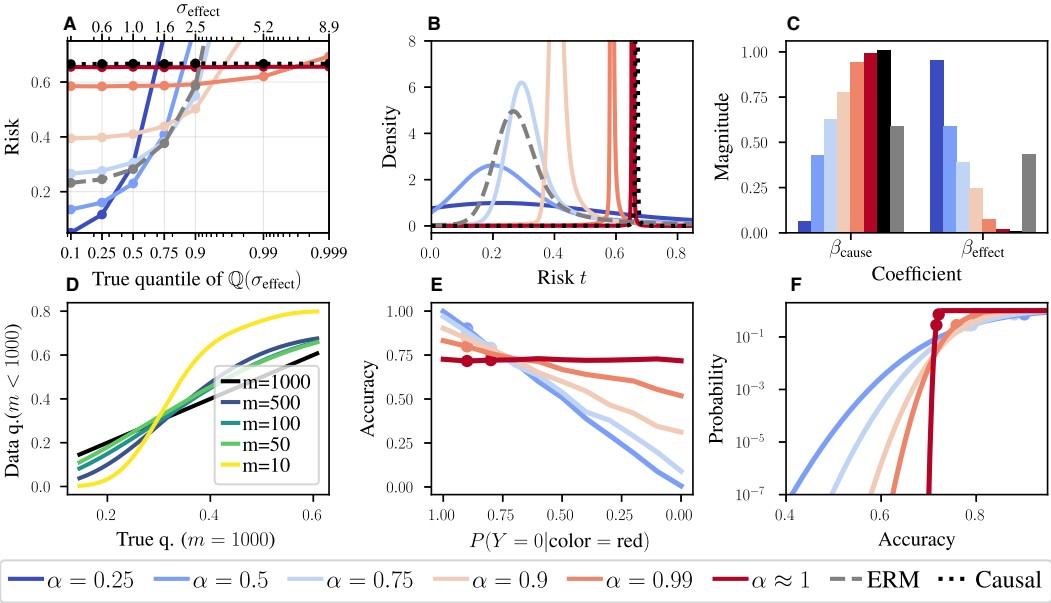

Figure 3: **EQRM on a toy linear regression dataset (A–D) and on ColoredMNIST (E–F). A:** Test risk at different quantiles or degrees of "OODness". For each quantile (x-axis), the corresponding $\alpha$ has the lowest risk (y-axis). **B:** Estimated risk distributions (corresponding CDFs in Appendix G.1.1). **C:** Regression coefficients approach those of the causal predictor ($\beta_{\text{cause}} = 1$, $\beta_{\text{effect}} = 0$) as $\alpha \to 1$. **D:** Q-Q plot comparing the "true" risk quantiles (estimated with $m = 1000$) against estimated ones ($m < 1000$), for $\alpha = 0.9$. **E:** Performance of different $\alpha$'s over increasingly OOD test domains, with dots showing training-domain accuracies. **F:** KDE-estimated accuracy-CDFs depicting accuracy-robustness curves. Larger $\alpha$'s make lower accuracies less likely.

**ColoredMNIST.** We next consider the `ColoredMNIST` or `CMNIST` dataset [9]. Here, the `MNIST` dataset is used to construct a binary classification task (0–4 or 5–9) in which digit color (red or green) is a highly-informative but spurious feature. In particular, the two training domains are constructed such that red digits have an 80% and 90% chance of belonging to class 0, while the single test domain is constructed such that they only have a 10% chance. The goal is to learn an invariant predictor which uses only digit shape—a stable feature having a 75% chance of correctly determining the class in all 3 domains. We compare with IRM [9], GroupDRO [45], SD [112], IGA [113] and VREx [41] using: (i) random initialization (Xavier method [114]); and (ii) random initialization followed by several iterations of ERM. The ERM initialization or pretraining directly corresponds to the delicate penalty "annealing" or warm-up periods used by most penalty-based methods [9, 41, 112, 113]. For all methods, we use a 2-hidden-layer MLP with 390 hidden units, the Adam optimizer, a learning rate of 0.0001, and dropout with $p = 0.2$. We sweep over five penalty weights for the baselines and five $\alpha$'s for EQRM. See Appendix E.2 for more experimental details. Table 1 shows that: (i) all methods struggle without ERM pretraining, explaining the need for penalty-annealing strategies in previous works and corroborating the results of [115, Table 1]; (ii) with ERM pretraining, EQRM matches or outperforms baseline methods, even approaching oracle performance (that of ERM trained on grayscale digits). These results suggest ERM pretraining as an effective strategy for DG methods.

In addition, Fig. 3 depicts the behavior of EQRM with different $\alpha$s. Here we see that: **E:** increasing $\alpha$ leads to more consistent performance across domains, eventually forcing the model to ignore color and focus on shape for invariant-risk prediction; and **F:** a predictor's (estimated) accuracy-CDF depicts its accuracy-robustness curve, just as its risk-CDF depicts its risk-robustness curve. Note that $\alpha = 0.5$ gives the best worst-case (i.e. worst-domain) risk over the two training domains—the preferred solution of DRO [45]—while $\alpha \to 1$ sacrifices risk for increased invariance or robustness.

## 6.2 Real-world datasets

We now evaluate our methods on the real-world or *in-the-wild* distribution shifts of WILDS [12]. We focus our evaluation on `iWildCam` [50] and `OGB-MolPCBA` [116, 117]—two large-scale classification datasets which have numerous test domains and thus facilitate a comparison of the test-domain risk distributions and their quantiles. Additional comparisons (e.g. using average accuracy) can be found in Appendix G.3. Our results demonstrate that, across two distinct data types (images and molecular graphs), EQRM offers superior tail or quantile performance.

Table 1: CMNIST test accuracy.

| Algorithm | Initialization | |
|---|---|---|
| | Rand. | ERM |
| ERM | $27.9 \pm 1.5$ | $27.9 \pm 1.5$ |
| IRM | $\mathbf{52.5 \pm 2.4}$ | $69.7 \pm 0.9$ |
| GrpDRO | $27.3 \pm 0.9$ | $29.0 \pm 1.1$ |
| SD | $49.4 \pm 1.5$ | $70.3 \pm 0.6$ |
| IGA | $50.7 \pm 1.4$ | $57.7 \pm 3.3$ |
| V-REx | $\mathbf{55.2 \pm 4.0}$ | $71.6 \pm 0.5$ |
| EQRM | $53.4 \pm 1.7$ | $71.4 \pm 0.4$ |
| Oracle | $72.1 \pm 0.7$ | |

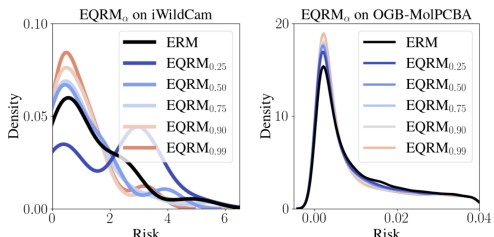

Figure 4: Test-domain risk distributions.

Table 2: EQRM test risks on iWildCam.

| Alg. | Mean risk | Quantile risk | | | | | | |
|---|---|---|---|---|---|---|---|---|
| | | 0.0 | 0.25 | 0.50 | 0.75 | 0.90 | 0.99 | 1.0 |
| ERM | 1.31 | 0.015 | 0.42 | 0.76 | 2.25 | 2.73 | 4.99 | 5.25 |
| IRM | 1.53 | 0.098 | 0.52 | 1.24 | 1.86 | 2.36 | 6.95 | 7.46 |
| GroupDRO | 1.73 | 0.091 | 0.68 | 1.65 | 2.18 | 3.36 | 5.29 | 5.54 |
| CORAL | 1.27 | 0.024 | 0.45 | 0.73 | 2.12 | 2.66 | 4.50 | 4.98 |
| $EQRM_{0.25}$ | 2.03 | 0.024 | 0.46 | 2.70 | 3.01 | 3.48 | 5.03 | 5.26 |
| $EQRM_{0.50}$ | 1.11 | **0.004** | 0.24 | 0.68 | 1.71 | 2.15 | 4.04 | 4.11 |
| $EQRM_{0.75}$ | 1.05 | 0.009 | **0.21** | 0.68 | 1.50 | 2.35 | 4.88 | 5.45 |
| $EQRM_{0.90}$ | **0.98** | 0.047 | 0.28 | **0.63** | **1.26** | **1.81** | 4.11 | 4.48 |
| $EQRM_{0.99}$ | 0.99 | 0.12 | 0.35 | 0.64 | 1.30 | 2.00 | **3.44** | **3.55** |

Table 3: EQRM test risks on OGB-MolPCBA.

| Alg. | Mean risk | Quantile risk | | | | | | |
|---|---|---|---|---|---|---|---|---|
| | | 0.0 | 0.25 | 0.50 | 0.75 | 0.90 | 0.99 | 1.0 |
| ERM | **0.051** | 0.0 | 0.004 | 0.017 | 0.060 | 0.13 | 0.49 | 16.04 |
| IRM | 0.073 | 0.098 | 0.52 | 1.24 | 1.86 | 2.36 | 6.95 | 7.46 |
| GroupDRO | 0.21 | 0.091 | 0.68 | 1.65 | 2.18 | 3.36 | 5.29 | **5.54** |
| CORAL | 0.055 | 0.0 | 0.12 | 0.32 | 1.23 | 2.01 | 5.76 | 7.44 |
| $EQRM_{0.25}$ | 0.054 | 0.0 | 0.003 | 0.016 | 0.059 | 0.13 | 0.48 | 15.46 |
| $EQRM_{0.50}$ | 0.052 | 0.0 | 0.003 | 0.015 | 0.059 | 0.13 | 0.48 | 11.33 |
| $EQRM_{0.75}$ | 0.052 | 0.0 | 0.003 | 0.015 | 0.059 | 0.13 | 0.47 | 12.15 |
| $EQRM_{0.90}$ | 0.052 | 0.0 | 0.003 | 0.015 | 0.059 | 0.12 | 0.47 | 10.81 |
| $EQRM_{0.99}$ | 0.053 | 0.0 | 0.003 | **0.014** | **0.055** | **0.11** | **0.46** | 7.16 |

**iWildCam.** We first consider the iWildCam image-classification dataset, which has 243 training domains and 48 test domains. Here, the label $Y$ is one of 182 different animal species and the domain $e$ is the camera trap which captured the image. In Table 2, we observe that $EQRM_{\alpha}$ does indeed tend to optimize the $\alpha$-risk quantile, with larger $\alpha$s during training resulting in lower test-domain risks at the corresponding quantiles. In the left pane of Fig. 4, we plot the (KDE-smoothed) test-domain risk distribution for ERM and EQRM. Here we see a clear trend: as $\alpha$ increases, the tails of the risk distribution tend to drop below ERM, which corroborates the superior quantile performance reported in Table 2. Note that, in Table 2, EQRM tends to record lower *average* risks than ERM. This has several plausible explanations. First, the number of testing domains (48) is relatively small, which could result in a biased sample with respect to the training domains. Second, the test domains may not represent i.i.d. draws from $\mathbb{Q}$, as WILDS [12] test domains tend to be more challenging.

**OGB-MolPCBA.** We next consider the OGB-MolPCBA (or OGB) dataset, which is a molecular graph-classification benchmark containing 44,930 training domains and 43,793 test domains with an average of 3.6 samples per domain. Table 3 shows that ERM achieves the lowest *average* test risk on OGB, in contrast to the iWildCam results, while $EQRM_{\alpha}$ still achieves stronger quantile performance. Of particular note is the fact that our methods significantly outperform ERM with respect to worst-case performance (columns/quantiles labeled 1.0); when $QRM_{\alpha}$ is run with large values of $\alpha$, we reduce the worst-case risk by more than a factor of two. In Fig. 4, we again see that the risk distributions of $EQRM_{\alpha}$ have lighter tails than that of ERM.

**A new evaluation protocol for DG.** The analysis provided in Tables 2-3 and Fig. 4 diverges from the standard evaluation protocol in DG [12, 38]. Rather than evaluating an algorithm's performance *on average* across test domains, we seek to understand *the distribution of its performance*—particularly in the tails by means of the quantile function. This new evaluation protocol lays bare the importance of multiple test domains in DG benchmarks, allowing predictors' risk distributions to be analyzed and compared. Indeed, as shown in Tables 2-3, solely reporting a predictor's average or worst risk over test domains can be misleading when assessing its ability to generalize OOD, indicating that the performance of DG algorithms was likely never "", as reported in [38], but rather invisible through the lens of average performance. This underscores the necessity of incorporating tail- or quantile-risk measures into a more holistic evaluation protocol for DG, ultimately providing a more nuanced and complete picture. In practice, which measure is preferred will depend on the application. For example, medical applications could have a human-specified robustness-level or quantile-of-interest.

## 6.3 DomainBed datasets

Finally, we consider the benchmark datasets of DomainBed [38], in particular VLCS [118], PACS [119], OfficeHome [120], TerraIncognita [5] and DomainNet [121]. As each of these datasets contain just 4 or 6 domains, it is not possible to meaningfully compare tail or quantile performance. Nonetheless, in line with much recent work, and to compare EQRM to a range of standard baselines on few-domain datasets, Table 4 reports DomainBed results in terms of the average performance

Table 4: DomainBed results. Model selection: training-domain validation set.

| Algorithm | VLCS | PACS | OfficeHome | TerraIncognita | DomainNet | Avg |
|---|---|---|---|---|---|---|
| ERM | $77.5 \pm 0.4$ | $85.5 \pm 0.2$ | $66.5 \pm 0.3$ | $46.1 \pm 1.8$ | $40.9 \pm 0.1$ | 63.3 |
| IRM | $78.5 \pm 0.5$ | $83.5 \pm 0.8$ | $64.3 \pm 2.2$ | $47.6 \pm 0.8$ | $33.9 \pm 2.8$ | 61.6 |
| GroupDRO | $76.7 \pm 0.6$ | $84.4 \pm 0.8$ | $66.0 \pm 0.7$ | $43.2 \pm 1.1$ | $33.3 \pm 0.2$ | 60.9 |
| Mixup | $77.4 \pm 0.6$ | $84.6 \pm 0.6$ | $68.1 \pm 0.3$ | $47.9 \pm 0.8$ | $39.2 \pm 0.1$ | 63.4 |
| MLDG | $77.2 \pm 0.4$ | $84.9 \pm 1.0$ | $66.8 \pm 0.6$ | $47.7 \pm 0.9$ | $41.2 \pm 0.1$ | 63.6 |
| CORAL | $78.8 \pm 0.6$ | $86.2 \pm 0.3$ | $68.7 \pm 0.3$ | $47.6 \pm 1.0$ | $41.5 \pm 0.1$ | **64.6** |
| ARM | $77.6 \pm 0.3$ | $85.1 \pm 0.4$ | $64.8 \pm 0.3$ | $45.5 \pm 0.3$ | $35.5 \pm 0.2$ | 61.7 |
| VREx | $78.3 \pm 0.2$ | $84.9 \pm 0.6$ | $66.4 \pm 0.6$ | $46.4 \pm 0.6$ | $33.6 \pm 2.9$ | 61.9 |
| EQRM | $77.8 \pm 0.6$ | $86.5 \pm 0.2$ | $67.5 \pm 0.1$ | $47.8 \pm 0.6$ | $41.0 \pm 0.3$ | 64.1 |

across each choice of test domain. While EQRM outperforms most baselines, including ERM, we reiterate that comparing algorithms solely in terms of average performance can be misleading (see final paragraph of § 6.2). Full implementation details are given in Appendix E.3, with further results in Appendix G.2 (additional baselines, per-dataset results, and test-domain model selection).

# 7    Discussion

**Interpretable model selection.** $\alpha$ approximates the probability with which our predictor will generalize with risk below the associated $\alpha$-quantile value. Thus, $\alpha$ represents an interpretable parameterization of the risk-robustness trade-off. Such interpretability is critical for model selection in DG, and for practitioners with application-specific requirements on performance and/or robustness.

**The assumption of i.i.d. domains.** For $\alpha$ to approximate the probability of generalizing, training and test domains must be i.i.d.-sampled. While this is rarely true in practice—e.g. hospitals have shared funders, service providers, etc.—we can better satisfy this assumption by subscribing to a new data collection process in which we collect training-domain data which is representative of how the underlying system tends to change. For example: (i) randomly select 100 US hospitals; (ii) gather and label data from these hospitals; (iii) train our system with the desired $\alpha$; (iv) deploy our system to all US hospitals, where it will be successful with probability $\approx \alpha$. While this process may seem expensive, time-consuming and vulnerable (e.g. to new hospitals), it offers a promising path to machine learning systems which *generalize with high probability*. Moreover, it is worth noting the alternative: prior works achieve generalization by assuming that only particular types of shifts can occur, e.g. covariate shifts [22, 122, 123], label shifts [123, 124], concept shifts [125], measurement shifts [19], mean shifts [126], shifts which leave the mechanism of $Y$ invariant [9, 39, 41, 54], etc. In real-world settings, where the underlying shift mechanisms are often unknown, such assumptions are both difficult to justify and impossible to test. Future work could look to relax the i.i.d.-domains assumption by leveraging knowledge of domain dependencies (e.g. time).

**The wider value of risk distributions.** As demonstrated in § 6, a predictor's risk distribution has value beyond quantile-minimization—it estimates the probability associated with each level of risk. Thus, regardless of the algorithm used, risk distributions can be used to analyze trained predictors.

# 8    Conclusion

We have presented Quantile Risk Minimization for achieving *Probable* Domain Generalization, motivated by the argument that the goal of domain generalization should be to learn predictors which perform well *with high probability* rather than *on-average* or *in the worst case*. By explicitly relating training and test domains as draws from the same underlying meta-distribution, we proposed to learn predictors with minimal $\alpha$-quantile risk under the training domains. We then introduced the EQRM algorithm, for which we proved a generalization bound and recovery of the causal predictor as $\alpha \rightarrow 1$, before demonstrating that EQRM outperforms state-of-the-art baselines on real and synthetic data.

## Acknowledgments and Disclosure of Funding

The authors thank Chris Williams and Ian Mason for providing feedback on an earlier draft, as well as Lars Lorch, David Krueger, Francesco Locatello and members of the MPI Tübingen causality group for helpful discussions and comments. This work was supported by the German Federal Ministry of Education and Research (BMBF): Tübingen AI Center, FKZ: 01IS18039A, 01IS18039B; and by the Machine Learning Cluster of Excellence, EXC number 2064/1 – Project number 390727645.

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
