# Appendices

## Table of Contents

# A Causality

## A.1 Definitions and example

As in previous causal works on DG [9, 41, 53–55], our causality results assume all domains share the same underlying *structural causal model* (SCM) [56], with different domains corresponding to different interventions. For example, the different camera-trap deployments depicted in Fig. 1a may induce changes in (or interventions on) equipment, lighting, and animal-species prevalence rates.

**Definition A.1.** An *SCM*[5] $\mathcal{M} = (\mathcal{S}, \mathbb{P}_N)$ consists of a collection of $d$ *structural assignments*

$$\mathcal{S} = \{X_j \leftarrow g_j(\mathrm{Pa}(X_j), N_j)\}_{j=1}^d, \tag{A.1}$$

where $\mathrm{Pa}(X_j) \subseteq \{X_1, \ldots, X_d\} \setminus \{X_j\}$ are the *parents* or *direct causes* of $X_j$, and $\mathbb{P}_N = \prod_{j=1}^d \mathbb{P}_{N_j}$, a joint distribution over the (jointly) independent noise variables $N_1, \ldots, N_d$. An SCM $\mathcal{M}$ induces a ("causal") graph $\mathcal{G}$ which is obtained by creating a node for each $X_j$ and then drawing a directed edge from each parent in $\mathrm{Pa}(X_j)$ to $X_j$. We assume this graph to be acyclic.

We can draw samples from the *observational distribution* $\mathbb{P}_\mathcal{M}(X)$ by first sampling a noise vector $n \sim \mathbb{P}_N$, and then using the structural assignments to generate a data point $x \sim \mathbb{P}_\mathcal{M}(X)$, recursively computing the value of every node $X_j$ whose parents' values are known. We can also manipulate or *intervene* upon the structural assignments of $\mathcal{M}$ to obtain a related SCM $\mathcal{M}^e$.

**Definition A.2.** An *intervention e* is a modification to one or more of the structural assignments of $\mathcal{M}$, resulting in a new SCM $\mathcal{M}^e = (\mathcal{S}^e, \mathbb{P}_N^e)$ and (potentially) new graph $\mathcal{G}^e$, with structural assignments

$$\mathcal{S}^e = \{X_j^e \leftarrow g_j^e(\mathrm{Pa}^e(X_j^e), N_j^e)\}_{j=1}^d. \tag{A.2}$$

We can draw samples from the *intervention distribution* $\mathbb{P}_{\mathcal{M}^e}(X^e)$ in a similar manner to before, now using the modified structural assignments. We can connect these ideas to DG by noting that each intervention $e$ creates a new domain or *environment* $e$ with interventional distribution $\mathbb{P}(X^e, Y^e)$.

**Example A.3.** Consider the following linear SCM, with $N_j \sim \mathcal{N}(0, \sigma_j^2)$:

$$X_1 \leftarrow N_1, \qquad\qquad Y \leftarrow X_1 + N_Y, \qquad\qquad X_2 \leftarrow Y + N_2.$$

Here, interventions could replace the structural assignment of $X_1$ with $X_1^e \leftarrow 10$ and change the noise variance of $X_2$, resulting in a set of training environments $\mathcal{E}_{\mathrm{tr}} = \{\text{fix } X_1 \text{ to } 10, \text{ replace } \sigma_2 \text{ with } 10\}$.

## A.2 EQRM recovers the causal predictor

**Overview.** We now prove that EQRM recovers the causal predictor in two stages. First, we prove the formal versions of Prop. 4.3, i.e. that EQRM learns a minimal invariant-risk predictor as $\alpha \to 1$ when using the following estimators of $\mathbb{T}_f$: (i) a Gaussian estimator (Prop. A.4 of Appendix A.2.1); and (ii) kernel-density estimators with certain bandwidth-selection methods (Prop. A.5 of Appendix A.2.2). Second, we prove Thm. 4.4, i.e. that learning a minimal invariant-risk predictor is sufficient to recover the causal predictor under weaker assumptions than those of Peters et al. [54, Thm 2] and Krueger et al. [41, Thm 1] (Appendix A.2.3). Throughout this section, we consider the "population" setting within each domain (i.e., $n \to \infty$); in general, with only finitely-many observations from each domain, only approximate versions of these results are possible.

**Notation.** Given $m$ training risks $\{\mathcal{R}^{e_1}(f), \ldots, \mathcal{R}^{e_m}(f)\}$ corresponding to the risks of a fixed predictor $f$ on $m$ training domains, let

$$\hat{\mu}_f = \frac{1}{m} \sum_{i=1}^m \mathcal{R}^{e_i}(f)$$

denote the sample mean and

$$\hat{\sigma}_f^2 = \frac{1}{m-1} \sum_{i=1}^m (\mathcal{R}^{e_i}(f) - \hat{\mu}_f)^2$$

the sample variance of the risks of $f$.

---

[5] A Non-parametric Structural Equation Model with Independent Errors (NP-SEM-IE) to be precise.

### A.2.1 Gaussian estimator

When using a Gaussian estimator for $\widehat{\mathbb{T}}_f$, we can rewrite the EQRM objective of (4.1) in terms of the standard-Normal inverse CDF $\Phi^{-1}$ as

$$\hat{f}_\alpha := \underset{f \in \mathcal{F}}{\arg\min} \ \hat{\mu}_f + \Phi^{-1}(\alpha) \cdot \hat{\sigma}_f. \tag{A.3}$$

Informally, we see that $\alpha \to 1 \implies \Phi^{-1}(\alpha) \to \infty \implies \hat{\sigma}_f \to 0$. More formally, we now show that, as $\alpha \to 1$, minimizing (A.3) leads to a predictor with minimal invariant-risk:

**Proposition A.4** (Gaussian QRM learns a minimal invariant-risk predictor as $\alpha \to 1$). *Assume*

1. $\mathcal{F}$ *contains an invariant-risk predictor* $f_0 \in \mathcal{F}$ *with finite mean risk (i.e.,* $\hat{\sigma}_{f_0} = 0$ *and* $\hat{\mu}_{f_0} < \infty$*), and*
2. *there are no arbitrarily negative mean risks (i.e.,* $\mu_* := \inf_{f \in \mathcal{F}} \mu_f > -\infty$*).*

*Then, for the Gaussian QRM predictor $\hat{f}_\alpha$ given in Eq. (A.3),*

$$\lim_{\alpha \to 1} \hat{\sigma}_{\hat{f}_\alpha} = 0 \quad and \quad \limsup_{\alpha \to 1} \hat{\mu}_{\hat{f}_\alpha} \leq \hat{\mu}_{f_0}.$$

Prop. A.4 essentially states that, if an invariant-risk predictor exists, then Gaussian EQRM equalizes risks across the $m$ domains, to a value at most the risk of the invariant-risk predictor. As we discuss in Appendix A.2.3, an invariant-risk predictor $f_0$ (Assumption 1. of Prop. A.4 above) exists under the assumption that the mechanism generating the labels $Y$ does not change between domains and is contained in the hypothesis class $\mathcal{F}$, together with a homoscedasticity assumption (see Appendix G.1.2). Meanwhile, Assumption 2. of Prop. A.4 above is quite mild and holds automatically for most loss functions used in supervised learning (e.g., squared loss, cross-entropy, hinge loss, etc.). We now prove Prop. A.4.

*Proof.* By definitions of $\hat{f}_\alpha$ and $f_0$,

$$\hat{\mu}_{\hat{f}_\alpha} + \Phi^{-1}(\alpha) \cdot \hat{\sigma}_{\hat{f}_\alpha} \leq \hat{\mu}_{f_0} + \Phi^{-1}(\alpha) \cdot \hat{\sigma}_{f_0} = \hat{\mu}_{f_0}. \tag{A.4}$$

Since for $\alpha \geq 0.5$ we have that $\Phi^{-1}(\alpha)\hat{\sigma}_{\hat{f}_\alpha} \geq 0$, it follows that $\hat{\mu}_{\hat{f}_\alpha} \leq \hat{\mu}_{f_0}$. Moreover, rearranging and using the definition of $\mu_*$, we obtain

$$\hat{\sigma}_{\hat{f}_\alpha} \leq \frac{\hat{\mu}_{f_0} - \hat{\mu}_{\hat{f}_\alpha}}{\Phi^{-1}(\alpha)} \leq \frac{\hat{\mu}_{f_0} - \mu_*}{\Phi^{-1}(\alpha)} \to 0 \quad as \quad \alpha \to 1.$$

$\square$

**Connection to VREx.** For the special case of using a Gaussian estimator for $\widehat{\mathbb{T}}_f$, we can equate the EQRM objective of (A.3) with the $\mathcal{R}_{\text{VREx}}$ objective of [41, Eq. 8]. To do so, we rewrite $\mathcal{R}_{\text{VREx}}$ in terms of the sample mean and variance:

$$\underset{f \in \mathcal{F}}{\arg\min} \ \mathcal{R}_{\text{VREx}}(f) = \underset{f \in \mathcal{F}}{\arg\min} \ m \cdot \hat{\mu}_f + \beta \cdot \hat{\sigma}_f^2. \tag{A.5}$$

Note that as $\beta \to \infty$, $\mathcal{R}_{\text{VREx}}$ learns a minimal invariant-risk predictor under the same assumptions, and by the same argument, as Prop. A.4. Dividing this objective by the positive constant $m > 0$, we can rewrite it in a form that allows a direct comparison of our $\alpha$ parameter and this $\beta$ parameter:

$$\underset{f \in \mathcal{F}}{\arg\min} \ \hat{\mu}_f + \left( \frac{\beta \cdot \hat{\sigma}_f}{m} \right) \cdot \hat{\sigma}_f. \tag{A.6}$$

Comparing (A.6) and (A.3), we note the relation $\beta = m \cdot \Phi^{-1}(\alpha) / \hat{\sigma}_f$ for a fixed $f$. For different $f$s, a particular setting of our parameter $\alpha$ corresponds to different settings of Krueger et al.'s $\beta$ parameter, depending on the sample standard deviation over training risks $\hat{\sigma}_f$.

### A.2.2 Kernel density estimator

We now consider the case of using a kernel density estimate, in particular,

$$\hat{F}_{\text{KDE},f}(x) = \frac{1}{m} \sum_{i=1}^{m} \Phi\left(\frac{x - R^{e_i}(f)}{h_f}\right) \tag{A.7}$$

to estimate the cumulative risk distribution.

**Proposition A.5** (Kernel EQRM learns a minimal risk-invariant predictor as $\alpha \to 1$). *Let*

$$\hat{f}_\alpha := \arg\min_{f \in \mathcal{F}} \hat{F}_{\text{KDE},f}^{-1}(\alpha),$$

*be the kernel EQRM predictor, where $\hat{F}_{\text{KDE},f}^{-1}$ denotes the quantile function computed from the kernel density estimate over (empirical) risks of $f$ with a standard Gaussian kernel. Suppose we use a data-dependent bandwidth $h_f$ such that $h_f \to 0$ implies $\hat{\sigma}_f \to 0$ (e.g., the "Gaussian-optimal" rule $h_f = (4/3m)^{0.2} \cdot \hat{\sigma}_f$ [65]). As in Proposition A.4, suppose also that*

1. *$\mathcal{F}$ contains an invariant-risk predictor $f_0 \in \mathcal{F}$ with finite training risks (i.e., $\hat{\sigma}_{f_0} = 0$ and each $R^{e_i}(f_0) < \infty$), and*
2. *there are no arbitrarily negative training risks (i.e., $R_* := \inf_{f \in \mathcal{F}, i \in [m]} R^{e_i}(f) > -\infty$).*

*For any $f \in \mathcal{F}$, let $R_f^* := \min_{i \in [m]} R^{e_i}(f)$ denote the smallest of the (empirical) risks of $f$ across domains. Then,*

$$\lim_{\alpha \to 1} \hat{\sigma}_{\hat{f}_\alpha} = 0 \quad \text{and} \quad \limsup_{\alpha \to 1} R_{\hat{f}_\alpha}^* \leq R_{f_0}^*.$$

As in Prop. A.4, Assumption 1 depends on invariance of the label-generating mechanism across domains (as discussed further in Appendix A.2.3 below), while Assumption 2 automatically holds for most loss functions used in supervised learning. We now prove Prop. A.5.

*Proof.* By our assumption on the choice of bandwidth, it suffices to show that, as $\alpha \to 1$, $h_{\hat{f}_\alpha} \to 0$.

Let $\Phi$ denote the standard Gaussian CDF. Since $\Phi$ is non-decreasing, for all $x \in \mathbb{R}$,

$$\hat{F}_{\text{KDE},\hat{f}_\alpha}(x) = \frac{1}{m} \sum_{i=1}^{m} \Phi\left(\frac{x - R^{e_i}(\hat{f}_\alpha)}{h_{\hat{f}_\alpha}}\right) \leq \Phi\left(\frac{x - R_{\hat{f}_\alpha}^*}{h_{\hat{f}_\alpha}}\right).$$

In particular, for $x = \hat{F}_{\text{KDE},\hat{f}_\alpha}^{-1}(\alpha)$, we have

$$\alpha = \hat{F}_{\text{KDE},\hat{f}_\alpha}(\hat{F}_{\text{KDE},\hat{f}_\alpha}^{-1}(\alpha)) \leq \Phi\left(\frac{\hat{F}_{\text{KDE},\hat{f}_\alpha}^{-1}(\alpha) - R_f^*}{h_{\hat{f}_\alpha}}\right).$$

Inverting $\Phi$ and rearranging gives

$$R_f^* + h_{\hat{f}_\alpha} \cdot \Phi^{-1}(\alpha) \leq \hat{F}_{\text{KDE},\hat{f}_\alpha}^{-1}(\alpha).$$

Hence, by definitions of $\hat{f}_\alpha$ and $f_0$,

$$R_f^* + h_{\hat{f}_\alpha} \cdot \Phi^{-1}(\alpha) \leq \hat{F}_{\text{KDE},\hat{f}_\alpha}^{-1}(\alpha) \leq \hat{F}_{\text{KDE},f_0}^{-1}(\alpha) = R_{f_0}^*. \tag{A.8}$$

Since, for $\alpha \geq 0.5$ we have that $h_{\hat{f}_\alpha} \cdot \Phi^{-1}(\alpha) \geq 0$, it follows that $R_{\hat{f}_\alpha}^* \leq R_{f_0}^*$. Moreover, rearranging Inequality (A.8) and using the definition of $R_*$, we obtain

$$h_{\hat{f}_\alpha} \leq \frac{R_{f_0}^* - R_{\hat{f}_\alpha}^*}{\Phi^{-1}(\alpha)} \leq \frac{R_{f_0}^* - R_*}{\Phi^{-1}(\alpha)} \to 0$$

as $\alpha \to 1$. $\qquad\square$

### A.2.3 Causal recovery

We now discuss and prove our main result, Thm. 4.4, regarding the conditions under which the causal predictor is the only minimal invariant-risk predictor. Together with Props. A.4 and A.5, this provides the conditions under which EQRM successfully performs "causal recovery", i.e., correctly recovers the true causal coefficients in a linear causal model of the data. As discussed in Appendix G.1.2, EQRM recovers the causal predictor by seeking *invariant risks* across domains, which differs from seeking *invariant functions* or coefficients (as in IRM [9]). As we discuss below, Thm. 4.4 generalizes related results in the literature regarding causal recovery based on *invariant risks* [41, 54].

**Assumption (v).** In contrast to both Peters et al. [54] and Krueger et al. [41], we do not require specific types of interventions on the covariates. In particular, our main assumption on the distributions of the covariates across domains, namely that the system of $d$-variate quadratic equations in (4.3) has a unique solution, is more general than these comparable results. For example, whereas both Peters et al. [54] and Krueger et al. [41] require one or more separate interventions for *every* covariate $X_j$, Example 4 below shows that we only require interventions on the subset of covariates that are effects of $Y$, while weaker conditions suffice for other covariates. Although this generality comes at the cost of abstraction, we now provide some concrete examples with different types of interventions to aid understanding. Note that, to simplify calculations and provide a more intuitive form, (4.3) of Thm. 4.4 assumes, without loss of generality, that all covariates are standardized to have mean 0 and variance 1, except where interventions change these. We can, however, rewrite (4.3) of Thm. 4.4 in a slightly more general form which does not require this assumption of standardized covariates:

$$
\begin{aligned}
0 \geq & x^\mathsf{T} \mathbb{E}_{X \sim e_1}[XX^\mathsf{T}]x + 2x^\mathsf{T} \mathbb{E}_{N,X \sim e_1}[NX] \\
= & \cdots \\
= & x^\mathsf{T} \mathbb{E}_{X \sim e_m}[XX^\mathsf{T}]x + 2x^\mathsf{T} \mathbb{E}_{N,X \sim e_m}[NX].
\end{aligned} \tag{A.9}
$$

We now present a number of concrete examples or special cases in which Assumption (v) of Thm. 4.4 would be satisfied, using this slightly more general form. In each example, we assume that variables are generated according to an SCM with an acyclic causal graph, as described in Appendix A.1.

1. *No effects of $Y$.* In the case that there are no effects of $Y$ (i.e., no $X_j$ is a causal descendant of $Y$, and hence each $X_j$ is uncorrelated with $N$), it suffices for there to exists at least one environment $e_i$ in which the covariance $\mathrm{Cov}_{X \sim e}[X]$ has full rank. These are standard conditions for identifiability in linear regression. More generally, it suffices for $\sum_{i=1}^{m} \mathrm{Cov}_{X \sim e_i}[X]$ to have full rank; this is the same condition one would require if simply performing linear regression on the pooled data from all $m$ environments. Intuitively, this full-rank condition guarantees that the observed covariate values are sufficiently uncorrelated to distinguish the effect of each covariate on the response $Y$. However, it does not necessitate interventions on the covariates, which are necessary to identify the *direction of causation* in a linear model; hence, this full-rank condition fails to imply causal recovery in the presence of effects of $Y$. See Appendix G.1.2 for a concrete example of this failure.

2. Hard *interventions.* For each covariate $X_j$, compared to some baseline environment $e_0$, there is some environment $e_{X_j}$ arising from a hard single-node intervention $do(X_j = z)$, with $z \neq 0$. If $X_j$ is any leaf node in the causal DAG, then in $e_{X_j}$, $X_j$ is uncorrelated with $N$ and with each $X_k$ ($k \neq j$), so the inequality in (A.9) gives

$$
0 \geq x^\mathsf{T} \mathbb{E}_{X \sim e_{X_j}}[XX^\mathsf{T}]x = x_j^2 z^2 + x_{-j}^\mathsf{T} \mathbb{E}_{X \sim e_0}[XX^\mathsf{T}]x_{-j}.
$$

Since the matrix $\mathbb{E}_{X \sim e}[XX^\mathsf{T}]$ is positive semidefinite (and $z \neq 0$ implies $z^2 > 0$), it follows that $x_j = 0$. The terms in (A.9) containing $x_j$ thus vanish, and iterating this argument for parents of leaf nodes in the causal DAG, and so on, gives $x = 0$. This condition is equivalent to that in Theorem 2(a) of Peters et al. [54] and is a strict improvement over Corollary 2 of Yin et al. [66] and Theorem 1 of Krueger et al. [41], which respectively require two and three distinct hard interventions on each variable.

3. *Shift interventions.* For each covariate $X_j$, compared to some baseline environment $e_0$, there is some environment $e_{X_j}$ consisting of the shift intervention $X_j \leftarrow g_j(\mathrm{Pa}(X_j), N_j) + z$, for some $z \neq 0$. Recalling that we assumed each covariate was centered (i.e., $\mathbb{E}_{X \sim e_0}[X_k] = 0$) in $e_0$, if $X_j$ is any leaf node in the causal DAG, then every other covariate remains centered in $e_{X_j}$ (i.e.,

$\mathbb{E}_{X \sim e_{X_j}}[X_k] = 0$ for each $k \neq j$). Hence, the excess risk is

$$x^{\mathsf{T}}\mathbb{E}_{X \sim e_{X_j}}[XX^{\mathsf{T}}]x + 2x^{\mathsf{T}}\mathbb{E}_{N,X \sim e_{X_j}}[NX] = x_j^2 z^2 + x^{\mathsf{T}}\mathbb{E}_{X \sim e_0}[XX^{\mathsf{T}}]x + 2x^{\mathsf{T}}\mathbb{E}_{N,X \sim e_0}[NX].$$

Since, by (A.9),

$$x^{\mathsf{T}}\mathbb{E}_{X \sim e_0}[XX^{\mathsf{T}}]x + 2x^{\mathsf{T}}\mathbb{E}_{N,X \sim e_0}[NX] = x^{\mathsf{T}}\mathbb{E}_{X \sim e_{X_j}}[XX^{\mathsf{T}}]x + 2x^{\mathsf{T}}\mathbb{E}_{N,X \sim e_{X_j}}[NX],$$

it follows that $x_j^2 z^2 = 0$, and so, since $z \neq 0$, $x_j = 0$. As above, the terms in (A.9) containing $x_j$ thus vanish, and iterating this argument for parents of leaf nodes in the causal DAG, and so on, gives $x = 0$. This condition is equivalent to the additive setting of Theorem 2(b) of Peters et al. [54].

4. *Noise interventions.* Suppose that each covariate is related to its causal parents through an additive noise model; i.e.,

$$X_j = g_j(\mathrm{Pa}(X_j)) + N_j,$$

where $\mathbb{E}[N_j] = 0$ and $0 < \mathbb{E}[N^2] < \infty$. Theorem 2(b) of Peters et al. [54] considers "noise" interventions, of the form

$$X_j \leftarrow g_j(\mathrm{Pa}(X_j)) + \sigma N_j,$$

where $\sigma^2 \neq 1$. Suppose that, for each covariate $X_j$, compared to some baseline environment $e_0$, there exists an environment $e_{X_j}$ consisting of the above noise intervention. If $X_j$ is any leaf node in the causal DAG, then, since we assumed $\mathbb{E}_{X \sim e_0}[X_j^2] = 1$,

$$x^{\mathsf{T}}\mathbb{E}_{X \sim e_{X_j}}[XX^{\mathsf{T}}]x + 2x^{\mathsf{T}}\mathbb{E}_{N,X \sim e_{X_j}}[NX]$$
$$= (\sigma^2 - 1)x_j^2\mathbb{E}[N_j^2] + x^{\mathsf{T}}\mathbb{E}_{X \sim e_0}[XX^{\mathsf{T}}]x + 2x^{\mathsf{T}}\mathbb{E}_{N,X \sim e_0}[NX].$$

Hence, the system (A.9) implies $0 = (\sigma^2 - 1)x_j^2\mathbb{E}[N_j^2]$. Since $\sigma^2 \neq 1$ and $\mathbb{E}[N_j^2] > 0$, it follows that $x_j = 0$.

5. *Scale interventions.* For each covariate $X_j$, compared to some baseline environment $e_0$, there exist two environments $e_{X_j,i}$ ($i \in \{1,2\}$) consisting of scale interventions $X_j \leftarrow \sigma_i g_j(\mathrm{Pa}(X_j), N_j)$, for some $\sigma_i \neq \pm 1$, with $\sigma_1 \neq \sigma_2$. If $X_j$ is any leaf node in the causal DAG, then, since we assumed $\mathbb{E}_{X \sim e_0}[X_j^2] = 1$,

$$x^{\mathsf{T}}\mathbb{E}_{X \sim e_{X_j}}[XX^{\mathsf{T}}]x + 2x^{\mathsf{T}}\mathbb{E}_{N,X \sim e_{X_j}}[NX]$$
$$= (\sigma_i^2 - 1)x_j^2 + 2(\sigma_i - 1)x_j\mathbb{E}_{X \sim e_0}[X_j X_{-j}^{\mathsf{T}}]x_{-j}^{\mathsf{T}} + x^{\mathsf{T}}\mathbb{E}_{X \sim e_0}[XX^{\mathsf{T}}]x$$
$$+ 2(\sigma_i - 1)x_j\mathbb{E}_{N,X \sim e_0}[X_j N] + 2x^{\mathsf{T}}\mathbb{E}_{N,X \sim e_0}[NX].$$

Hence, the system (A.9) implies

$$0 = (\sigma_i^2 - 1)x_j^2 + 2(\sigma_i - 1)x_j\left(\mathbb{E}_{X \sim e_0}[X_j X_{-j}^{\mathsf{T}}]x_{-j}^{\mathsf{T}} + \mathbb{E}_{N,X \sim e_0}[X_j N]\right).$$

Since $\sigma_i^2 \neq 1$, if $x_j \neq 0$, then solving for $x_j$ gives

$$x_j = -2\frac{\mathbb{E}_{X \sim e_0}[X_j X_{-j}^{\mathsf{T}}]x_{-j}^{\mathsf{T}} + \mathbb{E}_{N,X \sim e_0}[X_j N]}{\sigma_i + 1}.$$

Since $\sigma_1 \neq \sigma_2$, this is possible only if $x_j = 0$. This provides an example where a single intervention per covariate would be insufficient to guarantee causal recovery, but two distinct interventions per covariate suffice.

6. *Sufficiently uncorrelated causes and intervened-upon effects.* Suppose that, within the true causal DAG, $\mathrm{De}(Y) \subseteq [d]$ indexes the *descendants*, or *effects* of $Y$ (e.g., in Figure 5, $\mathrm{De}(Y) = \{5,6,7\}$). Suppose that for every $j \in \mathrm{De}(Y)$, compared to a single baseline environment $e_0$, there is

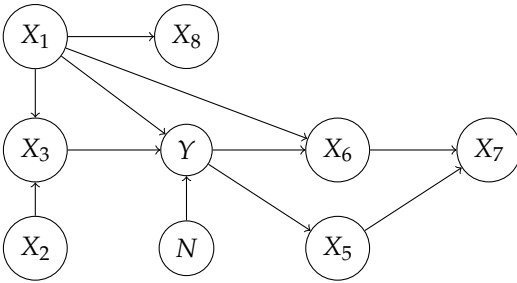

Figure 5: Example causal DAG with various types of covariates. $X_1$ and $X_3$ are the parents of $Y$, and so the true causal coefficient $\beta$ has only two non-zero coordinates $\beta_1$ and $\beta_3$. $X_1$, $X_2$, and $X_3$ are ancestors of $Y$. $X_5$, $X_6$, and $X_7$ are effects, or descendants, of $Y$ and are the only covariates for which $\mathbb{E}[X_j N]$ can be nonzero; hence, $X_5$, $X_6$, and $X_7$ are the only covariates on which interventions are generally necessary.

a environment $e_{X_j}$ consisting of either a $do(X_j = z)$ intervention or a shift intervention $X_j \leftarrow g_j(\mathrm{Pa}(X_j), N_j) + z$, with $z \neq 0$ and that the matrix

$$\sum_{i=1}^{m} \mathrm{Cov}_{X \sim e_i} \left[ X_{[d] \backslash \mathrm{De}(Y)} \right] \tag{A.10}$$

has full rank. Then, as argued in the previous two cases, for each $j \in \mathrm{De}(Y)$, $x_j = 0$. Moreover, for any $j \in [d] \backslash \mathrm{De}(Y)$, $\mathbb{E}[X_j N] = 0$, and so the system of equations (A.9) reduces to

$$0 \geq x_{[d] \backslash \mathrm{De}(Y)}^{\intercal} \, \mathbb{E}_{X \sim e_1} \left[ X_{[d] \backslash \mathrm{De}(Y)} X_{[d] \backslash \mathrm{De}(Y)}^{\intercal} \right] x_{[d] \backslash \mathrm{De}(Y)}^{\intercal}$$
$$= \cdots$$
$$= x_{[d] \backslash \mathrm{De}(Y)}^{\intercal} \, \mathbb{E}_{X \sim e_m} \left[ X_{[d] \backslash \mathrm{De}(Y)} X_{[d] \backslash \mathrm{De}(Y)}^{\intercal} \right] x_{[d] \backslash \mathrm{De}(Y)}^{\intercal}.$$

Since each $\mathbb{E}_{X \sim e_m} \left[ X_{[d] \backslash \mathrm{De}(Y)} X_{[d] \backslash \mathrm{De}(Y)}^{\intercal} \right]$ is positive semidefinite, the solution $x = 0$ to this reduced system of equations is unique if (and only if) the matrix (A.10) has full rank. This example demonstrates that interventions are only needed for effect covariates, while a weaker full-rank condition suffices for the remaining ones. In many practical settings, it may be possible to determine *a priori* that a particular covariate $X_j$ is not a descendant of $Y$; in this case, the practitioner need not intervene on $X_j$, as long as sufficiently diverse observational data on $X_j$ is available. To the best of our knowledge, this does not follow from any existing results in the literature, such as Theorem 2 of Peters et al. [54] or Corollary 2 of [66].

We conclude this section with the proof of Thm. 4.4:

*Proof.* Under the linear SEM setting with squared-error loss, for any estimator $\hat{\beta}$,

$$\mathcal{R}^e(\hat{\beta}) = \mathbb{E}_{N, X \sim e} \left[ \left( (\beta - \hat{\beta})^{\intercal} X + N \right)^2 \right]$$
$$= \mathbb{E}_{X \sim e} \left[ \left( (\beta - \hat{\beta})^{\intercal} X \right)^2 \right] + 2\mathbb{E}_{N, X \sim e} \left[ (\beta - \hat{\beta})^{\intercal} N X \right] + \mathbb{E}_N \left[ N^2 \right].$$

Since the second moment of the noise term $\mathbb{E}_N[N^2]$ is equal to the risk $\mathbb{E}_{(X,Y) \sim e}[(Y - \beta^T X)^2]$ of the causal predictor $\beta$, by the definition of $Y = \beta^T X + N$, we have that $\mathbb{E}_N[N^2]$ is invariant across environments. Thus, minimizing the squared error risk $\mathcal{R}^e(\hat{\beta})$ is equivalent to minimizing the excess risk

$$\mathbb{E}_{X \sim e} \left[ \left( (\beta - \hat{\beta})^{\intercal} X \right)^2 \right] + 2\mathbb{E}_{N, X \sim e} \left[ (\beta - \hat{\beta})^{\intercal} N X \right]$$
$$= (\beta - \hat{\beta})^{\intercal} \mathbb{E}_{X \sim e}[X X^{\intercal}] (\beta - \hat{\beta}) + 2(\beta - \hat{\beta})^{\intercal} \mathbb{E}_{N, X \sim e} [N X]$$

over estimators $\hat{\beta}$. Since the true coefficient $\beta$ is an invariant-risk predictor with $0$ excess risk, if $\hat{\beta}$ is a minimal invariant-risk predictor, it has at most $0$ invariant excess risk; i.e.,

$$
\begin{aligned}
0 &\geq (\beta - \hat{\beta})^\mathsf{T} \mathbb{E}_{X \sim e_1}[XX^\mathsf{T}](\beta - \hat{\beta}) + 2(\beta - \hat{\beta})^\mathsf{T} \mathbb{E}_{N, X \sim e_1}[NX] \\
&= \cdots \\
&= (\beta - \hat{\beta})^\mathsf{T} \mathbb{E}_{X \sim e_m}[XX^\mathsf{T}](\beta - \hat{\beta}) + 2(\beta - \hat{\beta})^\mathsf{T} \mathbb{E}_{N, X \sim e_m}[NX] \,.
\end{aligned}
\tag{A.11}
$$

By Assumption (v), the unique solution to this is $\beta - \hat{\beta} = 0$; i.e., $\hat{\beta} = \beta$. $\qquad\square$

# B   On the equivalence of different DG formulations

In Section 3, we claimed that under mild conditions, the minimax domain generalization problem in (2.2) is equivalent to the essential supremum problem in (3.1). In this subsection, we formally describe the conditions under which these two problems are equivalent. We also highlight several examples in which the assumptions needed to prove this equivalence hold.

Specifically, this appendix is organized as follows. First, in § B.1 we offer a more formal analysis of the equivalence between the probable domain general problems in (3.2) and (QRM). Next, in § B.2, we connect the domain generalization problem in (2.2) to the essential supremum problem in (3.1).

## B.1   Connecting formulations for QRM via a push-forward measure

To begin, we consider the abstract measure space $(\mathcal{E}_{\text{all}}, \mathcal{A}, \mathbb{Q})$, where $\mathcal{A}$ is a $\sigma$-algebra defined on the subsets of $\mathcal{E}_{\text{all}}$. Recall that in our setting, the domains $e \in \mathcal{E}_{\text{all}}$ are assumed to be drawn from the distribution $\mathbb{Q}$. Given this setting, in § 3 we introduced the probable domain generalization problem in (3.2), which we rewrite below for convenience:

$$
\min_{f \in \mathcal{F}, t \in \mathbb{R}} t \qquad \text{subject to} \qquad \Pr_{e \sim \mathbb{Q}} \{\mathcal{R}^e(f) \leq t\} \geq \alpha.
\tag{B.1}
$$

Our objective is to formally show that this problem is equivalent to (QRM). To do so, for each $f \in \mathcal{F}$, let consider a second measurable space $(\mathbb{R}_+, \mathcal{B})$, where $\mathbb{R}_+$ denotes the set of non-negative real numbers and $\mathcal{B}$ denotes the Borel $\sigma$-algebra over this space. For each $f \in \mathcal{F}$, we can now define the $(\mathbb{R}_+, \mathcal{B})$-valued random variable[6] $G_f : \mathcal{E}_{\text{all}} \to \mathbb{R}_+$ via

$$
G_f : e \mapsto \mathcal{R}^e(f) = \mathbb{E}_{\mathbb{P}(X^e, Y^e)}[\ell(f(X^e), Y^e)].
\tag{B.2}
$$

Concretely, $G_f$ maps an domain $e$ to the corresponding risk $\mathcal{R}^e(f)$ of $f$ in that domain. In this way, $G_f$ effectively summarizes $e$ by its effect on our predictor's risk, thus projecting from the often-unknown and potentially high-dimensional space of possible distribution shifts or interventions to the one-dimensional space of observed, real-valued risks. However, note that $G_f$ is not necessarily injective, meaning that two domains $e_1$ and $e_2$ may be mapped to the same risk value under $G_f$.

The utility of defining $G_f$ is that it allows us to formally connect (3.2) with (QRM) via a push-forward measure through $G_f$. That is, given any $f \in \mathcal{F}$, we can define the measure[7]

$$
\mathbb{T}_f =^d G_f \# \mathbb{Q}
\tag{B.3}
$$

where $\#$ denotes the *push-forward* operation and $=^d$ denotes equality in distribution. Observe that the relationship in (B.3) allows us to explicitly connect $\mathbb{Q}$—the often unknown distribution over (potentially high-dimensional and/or non-Euclidean) domain shifts in Fig. 1b—to $\mathbb{T}_f$—the distribution over real-valued risks in Fig. 1c, from which we can directly observe samples. In this way, we find that for each $f \in \mathcal{F}$,

$$
\Pr_{e \sim \mathbb{Q}} \{\mathcal{R}^e(f) \leq t\} = \Pr_{R \sim \mathbb{T}_f} \{R \leq t\}.
\tag{B.4}
$$

This relationship lays bare the connection between (3.2) and (QRM), in that the domain or environment distribution $\mathbb{Q}$ can be replaced by a distribution over risks $\mathbb{T}_f$.

---

[6]For brevity, we will assume that $G_f$ is always measurable with respect to the underlying $\sigma$-algebra $\mathcal{A}$.

[7]Here $\mathbb{T}_f$ is defined over the induced measurable space $(\mathbb{R}_+, \mathcal{B})$.

## B.2 Connecting (2.2) to the essential supremum problem (3.1)

We now study the relationship between (2.2) and (3.1). In particular, in § B.2.1 and § B.2.2, we consider the distinct settings wherein $\mathcal{E}_{\text{all}}$ comprises continuous and discrete spaces respectively.

### B.2.1 Continuous domain sets $\mathcal{E}_{\text{all}}$

When $\mathcal{E}_{\text{all}}$ is a continuous space, it can be shown that (2.2) and (3.1) are *equivalent* whenever: (a) the map $G_f$ defined in Section B.1 is continuous; and (b) the measure $\mathbb{Q}$ satisfies very mild regularity conditions.

**The case when $\mathbb{Q}$ is the Lebesgue measure.** Our first result concerns the setting in which $\mathcal{E}_{\text{all}}$ is a subset of Euclidean space and where $\mathbb{Q}$ is chosen to be the Lebesgue measure on $\mathcal{E}_{\text{all}}$. We summarize this result in the following proposition.

**Proposition B.1.** *Let us assume that the map $G_f$ is continuous for each $f \in \mathcal{F}$. Further, let $\mathbb{Q}$ denote the Lebesgue measure over $\mathcal{E}_{\text{all}}$; that is, we assume that domains are drawn uniformly at random from $\mathcal{E}_{\text{all}}$. Then (2.2) and (3.1) are equivalent.*

*Proof.* To prove this claim, it suffices to show that under the assumptions in the statement of the proposition, it holds for any $f \in \mathcal{F}$ that

$$\sup_{e \in \mathcal{E}_{\text{all}}} R^e(f) = \operatorname*{ess\,sup}_{e \sim \mathbb{Q}} R^e(f). \tag{B.5}$$

To do so, let us fix an arbitrary $f \in \mathcal{F}$ and write

$$A := \sup_{e \in \mathcal{E}_{\text{all}}} R^e(f) \quad \text{and} \quad B := \operatorname*{ess\,sup}_{e \sim \mathbb{Q}} R^e(f). \tag{B.6}$$

At a high-level, our approach is to show that $B \leq A$, and then that $A \leq B$, which together will imply the result in (B.5). To prove the first inequality, observe that by the definition of the supremum, it holds that $R^e(f) \leq A \; \forall e \in \mathcal{E}_{\text{all}}$. Therefore, $\mathbb{Q}\{e \in \mathcal{E}_{\text{all}} : R^e(f) > A\} = 0$, which directly implies that $B \leq A$. Now for the second inequality, let $\epsilon > 0$ be arbitrarily chosen. Consider that due to the continuity of $G_f$, there exists an $e_0 \in \mathcal{E}_{\text{all}}$ such that

$$R^{e_0}(f) + \epsilon > A. \tag{B.7}$$

Now again due to the continuity of $G_f$, we can choose a ball $\mathcal{B}_\epsilon \subset \mathcal{E}_{\text{all}}$ centered at $e_0$ such that $|R^e(f) - R^{e_0}(f)| \leq \epsilon \; \forall e \in \mathcal{B}_\epsilon$. Given such a ball, observe that $\forall e \in \mathcal{B}_\epsilon$, it holds that

$$R^e(f) \geq R^{e_0}(f) - \epsilon > A - 2\epsilon \tag{B.8}$$

where the first inequality follows from the reverse triangle inequality and the second inequality follows from (B.7). Because $\mathbb{Q}\{e \in \mathcal{B}_\epsilon : R^e(f) > A - 2\epsilon\} > 0$, it directly follows that $A - 2\epsilon \leq B$. As $\epsilon > 0$ was chosen arbitrarily, this inequality holds for any $\epsilon > 0$, and thus we can conclude that $A \leq B$, completing the proof. $\qquad\square$

**Generalizing Prop. B.1 to other measure $\mathbb{Q}$.** We note that this proof can be generalized to measures $\mathbb{Q}$ other than the Lebesgue measure. Indeed, the result holds for any measure $\mathbb{Q}$ taking support on $\mathcal{E}_{\text{all}}$ for which it holds that $\mathbb{Q}$ places non-zero probability mass on any closed subset of $\mathcal{E}_{\text{all}}$. This would be the case, for instance, if $\mathbb{Q}$ was a truncated Gaussian distribution with support on $\mathcal{E}_{\text{all}}$. Furthermore, if we let $\mathbb{L}$ denote the Lebesgue measure on $\mathcal{E}_{\text{all}}$, then another more general instance of this property occurs whenever $\mathbb{L}$ is absolutely continuous with respect to $\mathbb{Q}$, i.e., whenever $\mathbb{L} \ll \mathbb{Q}$.

**Corollary B.2.** *Let us assume that $\mathbb{Q}$ places nonzero mass on every open ball with radius strictly larger than zero. Then under the continuity assumptions of Prop. B.1, it holds that (2.2) and (3.1) are equivalent.*

*Proof.* The proof of this fact follows along the same lines as that of Prop. B.1. In particular, the same argument shows that $B \leq A$. Similarly, to show that $A \leq B$, we can use the same argument, noting that $\mathbb{Q}\{e \in \mathcal{B}_\epsilon : R^e(f) > A - 2\epsilon\}$ continues to hold, due to our assumption that $\mathbb{Q}$ places nonzero mass on $\mathcal{B}_\epsilon$. $\qquad\square$

**Examples.** We close this subsection by considering several real-world examples in which the conditions of Prop. B.1 hold. In particular, we focus on examples in the spirit of "Model-Based Domain Generalization" [22]. In this setting, it is assumed that the variation from domain to domain is parameterized by a *domain transformation model* $x^e \mapsto D(x^e, e') =: x^{e'}$, which maps the covariates $x^e$ from a given domain $e \in \mathcal{E}_{\text{all}}$ to another domain $e' \in \mathcal{E}_{\text{all}}$. As discussed in [22], domain transformation models cover settings in which inter-domain variation is due to *domain shift* [122, §1.8]. Indeed, under this model (formally captured by Assumptions 4.1 and 4.2 in [22]), the domain generalization problem in (2.2) can be equivalently rewritten as

$$\min_{f \in \mathcal{F}} \max_{e \in \mathcal{E}_{\text{all}}} \mathbb{E}_{(X,Y)}[\ell(f(D(X,e)), Y)]. \tag{B.9}$$

For details, see Prop. 4.3 in [22]. In this problem, $(X, Y)$ denote an underlying pair of random variables such that

$$\mathbb{P}(X^e) =^d D \# (\mathbb{P}(X), \delta(e)) \quad \text{and} \quad \mathbb{P}(Y^e) =^d \mathbb{P}(Y) \tag{B.10}$$

for each $e \in \mathcal{E}_{\text{all}}$ where $\delta(e)$ is a Dirac measure placed at $e \in \mathcal{E}_{\text{all}}$. Now, turning our attention back to Prop. B.1, we can show the following result for (B.9).

**Remark B.3.** Let us assume that the map $e \mapsto D(\cdot, e)$ is continuous with respect to a metric $d_{\mathcal{E}_{\text{all}}}(e, e')$ on $\mathcal{E}_{\text{all}}$ and that $x \mapsto \ell(x, \cdot)$ is continuous with respect to the absolute value. Further, assume that each predictor $f \in \mathcal{F}$ is continuous in the standard Euclidean metric on $\mathbb{R}^d$. Then (2.2) and (3.1) are equivalent.

*Proof.* By Prop. B.1, it suffices to show that the map

$$G_f : e \mapsto \mathbb{E}_{(X,Y)}[\ell(f(D(X,e)), Y)] \tag{B.11}$$

is a continuous function. To do so, recall that the composition of continuous functions is continuous, and therefore we have, by the assumptions listed in the above remark, that the map $e \mapsto \ell(f(D(x,e)), y)$ is continuous for each $(x, y) \sim (X, Y)$. To this end, let us define the function $h_f(x, y, e) := \ell(f(D(x,e)), y)$ and let $\epsilon > 0$. By the continuity of $h_f$ in $e$, there exists a $\delta = \delta(\epsilon) > 0$ such that $|h_f(x, y, e) - h_f(x, y, e')| < \epsilon$ whenever $d_{\mathcal{E}_{\text{all}}}(e, e') < \delta$. Now observe that

$$\left| \mathbb{E}_{(X,Y)}[h_f(X, Y, e)] - \mathbb{E}_{(X,Y)}[h_f(X, Y, e')] \right| \tag{B.12}$$

$$= \left| \int_{\mathcal{E}_{\text{all}}} h_f(X, Y, e) \mathrm{d}\mathbb{P}(X, Y) - \int_{\mathcal{E}_{\text{all}}} h_f(X, Y, e') \mathrm{d}\mathbb{P}(X, Y) \right| \tag{B.13}$$

$$= \left| \int_{\mathcal{E}_{\text{all}}} (h_f(X, Y, e) - h_f(X, Y, e')) \mathrm{d}\mathbb{P}(X, Y) \right| \tag{B.14}$$

$$\leq \int_{\mathcal{E}_{\text{all}}} \left| h_f(X, Y, e) - h_f(X, Y, e') \right| \mathrm{d}\mathbb{P}(X, Y). \tag{B.15}$$

Therefore, whenever $d_{\mathcal{E}_{\text{all}}}(e, e') < \delta$ it holds that

$$\left| \mathbb{E}_{(X,Y)}[h_f(X, Y, e)] - \mathbb{E}_{(X,Y)}[h_f(X, Y, e')] \right| \leq \int_{\mathcal{E}_{\text{all}}} \epsilon \mathrm{d}\mathbb{P}(X, Y) = \epsilon \tag{B.16}$$

by the monotonicity of expectation. This completes the proof that $G_f$ is continuous. $\square$

In this way, provided that the risks in each domain vary in a continuous way through $e$, (2.2) and (3.1) are equivalent. As a concrete example, consider an image classification setting in which the variation from domain to domain corresponds to different rotations of the images. This is the case, for instance, in the `RotatedMNIST` dataset [38, 127], wherein the training domains correspond to different rotations of the `MNIST` digits. Here, a domain transformation model $D$ can be defined by

$$D(x, e) = R(e)x \quad \text{where} \quad e \in \mathcal{E}_{\text{all}} \subseteq [0, 2\pi), \tag{B.17}$$

and where $R(e)$ is a rotation matrix. In this case, it is clear that $D$ is a continuous function of $e$ (in fact, the map is *linear*), and therefore the result in (B.3) holds.

### B.2.2 Discrete domain sets $\mathcal{E}_{\text{all}}$

When $\mathcal{E}_{\text{all}}$ is a discrete set, the conditions we require for (2.2) and (3.1) to be equivalent are even milder. In particular, the only restriction we place on the problems is that $\mathbb{Q}$ must place non-zero mass on each element of $\mathcal{E}_{\text{all}}$; that is, $\mathbb{Q}(e) > 0 \ \forall e \in \mathcal{E}_{\text{all}}$. We state this more formally below.

**Proposition B.4.** *Let us assume that $\mathcal{E}_{\text{all}}$ is discrete, and that $\mathbb{Q}$ is such that $\forall e \in \mathcal{E}_{\text{all}}$, it holds that $\mathbb{Q}(e) > 0$. Then it holds that* (2.2) *and* (3.1) *are equivalent.*

## C   Notes on KDE bandwidth selection

In our setting, we are interested in bandwidth-selection methods which: (i) work well for 1D distributions and small sample sizes $m$; and (ii) guarantee recovery of the causal predictor as $\alpha \to 1$ by satisfying $h_f \to 0 \implies \hat{\sigma}_f \to 0$, where $h_f$ is the data-dependent bandwidth and $\hat{\sigma}_f$ is the sample standard deviation (see Appendices A.2.2 and A.2.3). We thus investigated three popular bandwidth-selection methods: (1) the Gaussian-optimal rule [65], $h_f = (4/3m)^{0.2} \cdot \hat{\sigma}_f$; (2) Silverman's rule-of-thumb [65], $h_f = m^{-0.2} \cdot \min(\hat{\sigma}_f, \frac{\text{IQR}}{1.34})$, with IQR the interquartile range; and (3) the median-heuristic [128–130], which sets the bandwidth to be the median pairwise-distance between data points. Note that many sensible methods exist, as do more complete studies on bandwidth selection—see e.g. [65].

For (i), we found Silverman's rule-of-thumb [65] to perform very well, the Gaussian-optimal rule [65] to perform well, and the median-heuristic [128–130] to perform poorly. For (ii), only the Gaussian-optimal rule satisfies $h_f \to 0 \implies \hat{\sigma}_f \to 0$. Thus, in practice, we use either the Gaussian-optimal rule (particularly when causal predictor's are sought as $\alpha \to 1$), or Silverman's rule-of-thumb.

## D   Generalization bounds

This appendix states and proves our main generalization bound, Theorem D.1. Theorem D.1 applies for many possible estimates $\widehat{\mathbb{T}}_f$, and we further show how to apply Theorem D.1 to the specific case of using a kernel density estimate.

### D.1   Main generalization bound and proof

Suppose that, from each of $N$ IID environments $e_1, ..., e_N \sim \mathbb{P}(e)$, we observe $n$ IID labeled samples $(X_{i,1}, Y_{i,1}), ..., (X_{n,1}, Y_{n,1}) \sim \mathbb{P}(X^e, Y^e)$. Fix a hypothesis class $\mathcal{F}$ and confidence level $\alpha \in [0, 1]$. For any hypothesis $f : \mathcal{X} \to \mathcal{Y}$, define the *empirical risk on environment $e_i$* by

$$\widehat{\mathcal{R}}^{e_i}(f) := \frac{1}{n} \sum_{j=1}^{n} \ell\left(Y_{i,j}, f(X_{i,j})\right), \quad \text{for each} \quad i \in [N].$$

Throughout this section, we will abbreviate the distribution $F_{\mathbb{T}_f}(t) = \text{Pr}_e[\mathcal{R}^e(f) \leq t]$ of $f$'s risk by $F_f(t)$ and its estimate $F_{\widehat{\mathbb{T}}_f}$, computed from the observed empirical risks $\widehat{\mathcal{R}}^{e_1}(f), ..., \widehat{\mathcal{R}}^{e_N}(f)$, by $\widehat{F}_f$.

We propose to select a hypothesis by minimizing this over our hypothesis class:

$$\widehat{f} := \operatorname*{arg\,min}_{f \in \mathcal{F}} F_{\widehat{\mathbb{T}}_f}^{-1}(\alpha). \tag{D.1}$$

In this section, we prove a uniform generalization bound, which in particular, provides conditions under which the estimator (D.1) generalizes uniformly over $\mathcal{F}$. Because the novel aspect of the present paper is the notion of generalizing *across* environments, we will take for granted that the hypothesis class $\mathcal{F}$ generalizes uniformly *within* each environments (i.e., that each $\sup_{f \in \mathcal{F}} \mathcal{R}^{e_i}(f) - \widehat{\mathcal{R}}^{e_i}(f)$ can be bounded with high probability); myriad generalization bounds from learning theory can be used to show this.

**Theorem D.1.** *Let $\mathcal{G} := \{\widehat{F}(\mathcal{R}^{e_1}(f), \mathcal{R}^{e_2}(f), ..., \mathcal{R}^{e_N}(f)) : f \in \mathcal{F}, e_1, ..., e_n \in \mathcal{E}_{\text{all}}\}$ denote the class of possible estimated risk distributions over $N$ environments, and, for any $\epsilon > 0$, let $\mathcal{N}_\epsilon(\mathcal{G})$*

denote the $\epsilon$-covering number of $\mathcal{G}$ under $\mathcal{L}_\infty(\mathbb{R})$. Suppose the class $\mathcal{F}$ generalizes uniformly within environments; i.e., for any $\delta > 0$, there exists $t_{n,\delta,\mathcal{F}}$ such that

$$\underset{e}{\mathrm{ess\,sup}} \underset{\{(X_j,Y_j)\}_{j=1}^n \sim \mathbb{P}(X^e, Y^e)}{\Pr} \left[ \sup_{f \in \mathcal{F}} R^e(f) - \widehat{\mathcal{R}}^e(f) > t_{n,\delta,\mathcal{F}} \right] \leq \delta.$$

*Let*

$$\mathrm{Bias}(\mathcal{F}, \widehat{F}) := \sup_{f \in \mathcal{F}, t \in \mathbb{R}} F_f(t) - \mathbb{E}_{e_1,\dots,e_N}[\widehat{F}_f(t)]$$

*denote the worst-case bias of the estimator $\widehat{F}$ over the class $f$. Noting that $\widehat{F}_f$ is a function of the empirical risk CDF*

$$\widehat{Q}_f(t) := \frac{1}{N} \sum_{i=1}^N \mathbb{1}\{\mathcal{R}^{e_i}(f) \leq t\},$$

*suppose that the function $\widehat{Q}_f \mapsto \widehat{F}_f$ is L-Lipschitz under $\mathcal{L}_\infty(\mathbb{R})$. Then, for any $\epsilon, \delta > 0$,*

$$\underset{\substack{e_1,\dots,e_N \\ \{(X_j,Y_j)\}_{j=1}^n \sim \mathbb{P}(X^{e_i}, Y^{e_i})}}{\Pr} \left[ \sup_{f \in \mathcal{F}} F_f^{-1}\left( \alpha - B(\mathcal{F}, \widehat{F}) - \epsilon \right) - \widehat{F}_f^{-1}(\alpha) > t_{n, \frac{\delta}{N}, \mathcal{F}} \right] \leq \delta + 8 \mathcal{N}_{\epsilon/16}(\mathcal{G}) e^{-\frac{N\epsilon^2}{64L}}.$$

$$(\mathrm{D.2})$$

The key technical observation of Theorem D.1 is that we can pull the supremum over $\mathcal{F}$ outside the probability by incurring a $\mathcal{N}_{\epsilon/16}(\mathcal{G})$ factor increase in the probability of failure. To ensure $\mathcal{N}_{\epsilon/16}(\mathcal{G}) < \infty$, we need to limit the space of possible empirical risk profiles $\mathcal{G}$ (e.g., by kernel smoothing), incurring an additional bias term $B(\mathcal{F}, \widehat{F})$. As we demonstrate later, for common distribution estimators, such as kernel density estimators, one can bound the covering number $\mathcal{N}_{\epsilon/16}(\mathcal{G})$ in Inequality (D.2) by standard methods, and the Lipschitz constant $L$ is typically 1. Under mild (e.g., smoothness) assumptions on the family of possible true risk profiles, one can additionally bound the Bias Term, again by standard arguments.

Before proving Theorem D.1, we state two standard lemmas used in the proof:

**Lemma D.2** (Symmetrization; Lemma 2 of [131])**.** *Let $X$ and $X'$ be independent realizations of a random variable with respect to which $\mathcal{F}$ is a family of integrable functions. Then, for any $\epsilon > 0$,*

$$\Pr\left[ \sup_{f \in \mathcal{F}} f(X) - \mathbb{E}\, f(X) > \epsilon \right] \leq 2 \Pr\left[ \sup_{f \in \mathcal{F}} f(X) - f(X') > \frac{\epsilon}{2} \right].$$

**Lemma D.3** (Dvoretzky–Kiefer–Wolfowitz (DKW) Inequality; Corollary 1 of [132])**.** *Let $X_1, \dots, X_n$ be IID $\mathbb{R}$-valued random variables with CDF P. Then, for any $\epsilon > 0$,*

$$\Pr\left[ \sup_{t \in \mathbb{R}} \left| F_f(t) - \frac{1}{n} \sum_{i=1}^n \mathbb{1}\{X_i \leq t\} \right| > \epsilon \right] \leq 2 e^{-2n\epsilon^2}.$$

We now prove our main result, Theorem D.1.

*Proof of Theorem D.1.* For convenience, let $F_f(t) := \mathbb{P}_{e \sim \mathbb{P}(e)}[R^e(f) \leq t]$. In preparation for Symmetrization, for any $f \in \mathcal{F}$, let $\widehat{F}'_f$ denote $\widehat{F}_f$ computed on an independent "ghost" sample

$e'_1, ..., e'_N \sim \mathbb{P}(e)$. Then,

$$\Pr_{e_1,...,e_N} \left[ \sup_{f \in \mathcal{F}, t \in \mathbb{R}} \mathbb{E}_{e_1,...,e_N} \left[ \widehat{F}_f(t) \right] - \widehat{F}_f(t) > \epsilon \right] \tag{D.3}$$

$$\leq 2 \Pr_{\substack{e_1,...,e_N \\ e'_1,...,e'_N}} \left[ \sup_{f \in \mathcal{F}, t \in \mathbb{R}} \widehat{F}'_f(t) - \widehat{F}_f(t) > \epsilon/2 \right] \tag{D.4}$$

$$\leq 2 \Pr_{\substack{e_1,...,e_N \\ e'_1,...,e'_N}} \left[ \sup_{f \in \mathcal{F}} \left\| \widehat{F}'_f - \widehat{F}_f \right\|_\infty > \epsilon/2 \right] \tag{D.5}$$

$$\leq 2 \Pr_{\substack{e_1,...,e_N \\ e'_1,...,e'_N}} \left[ \sup_{f \in \mathcal{F}} \epsilon/8 + \left\| D\widehat{F}'_f - D\widehat{F}_f \right\|_\infty > \epsilon/2 \right] \tag{D.6}$$

$$\leq 2 \mathcal{N}_{\epsilon/16} \sup_{f \in \mathcal{F}} \Pr_{\substack{e_1,...,e_N \\ e'_1,...,e'_N}} \left[ \epsilon/8 + \left\| D\widehat{F}'_f - D\widehat{F}_f \right\|_\infty > \epsilon/2 \right] \tag{D.7}$$

$$\leq 2 \mathcal{N}_{\epsilon/16} \sup_{f \in \mathcal{F}} \Pr_{\substack{e_1,...,e_N \\ e'_1,...,e'_N}} \left[ \epsilon/4 + \left\| \widehat{F}'_f - \widehat{F}_f \right\|_\infty > \epsilon/2 \right] \tag{D.8}$$

$$= 2 \mathcal{N}_{\epsilon/16} \sup_{f \in \mathcal{F}} \Pr_{\substack{e_1,...,e_N \\ e'_1,...,e'_N}} \left[ \left\| \widehat{F}'_f - \widehat{F}_f \right\|_\infty > \epsilon/4 \right] \tag{D.9}$$

$$\leq 4 \mathcal{N}_{\epsilon/16} \sup_{f \in \mathcal{F}} \Pr_{e_1,...,e_N} \left[ \left\| \mathbb{E}_{e_1,...,e_N} \left[ \widehat{F}_f \right] - \widehat{F}_f \right\|_\infty > \epsilon/8 \right] \tag{D.10}$$

$$\leq 4 \mathcal{N}_{\epsilon/16} \sup_{f \in \mathcal{F}} \Pr_{e_1,...,e_N} \left[ \sup_{t \in \mathbb{R}} \left| F_f(t) - \frac{1}{N} \sum_{i=1}^N 1\{\mathcal{R}^e(f) \leq t\} \right| > \frac{\epsilon}{8L} \right] \tag{D.11}$$

$$\leq 8 \mathcal{N}_{\epsilon/16} \exp \left( -\frac{N\epsilon^2}{64L} \right). \tag{D.12}$$

Here, line (D.4) follows from the Symmetrization Lemma (Lemma D.2), lines (D.6) and (D.8) follow from the definition of $D$, line (D.7) is a union bound over $\widehat{\mathcal{P}}_{\epsilon/16}$, line (D.10) follows from the triangle inequality, line (D.11) follows from the Lipschitz assumption, and line (D.12) follows from the DKW Inequality (Lemma D.3).

Since $\sup_x f(x) - \sup_x g(x) \leq \sup_x f(x) - g(x)$,

$$\Pr_{e_1,...,e_N} \left[ \sup_{f \in \mathcal{F}, t \in \mathbb{R}} F_f(t) - \widehat{F}_f(t) > \epsilon + \text{Bias}(\mathcal{F}, \widehat{F}) \right]$$

$$= \Pr_{e_1,...,e_N} \left[ \sup_{f \in \mathcal{F}, t \in \mathbb{R}} F_f(t) - \widehat{F}_f(t) > \epsilon + \sup_{f \in \mathcal{F}, t \in \mathbb{R}} F_f(t) - \mathbb{E}_{e_1,...,e_N} \left[ \widehat{F}_f(t) \right] \right]$$

$$\leq \Pr_{e_1,...,e_N} \left[ \sup_{f \in \mathcal{F}, t \in \mathbb{R}} \mathbb{E}_{e_1,...,e_N} \left[ \widehat{F}_f(t) \right] - \widehat{F}_f(t) > \epsilon \right]$$

$$\leq 8 \mathcal{N}_{\epsilon/16} \exp \left( -\frac{N\epsilon^2}{64L} \right), \tag{D.13}$$

by (D.12). Meanwhile, applying the presumed uniform bound on within-environment generalization error together with a union bound over the $N$ environments, gives us a high-probability bound on the maximum generalization error of $f$ within any of the $N$ environments:

$$\Pr_{\substack{\{e_i\}_{i=1}^N \sim \mathbb{P}(e) \\ \{(X_{i,j}, Y_{i,j})\}_{j=1}^n \sim \mathbb{P}(X^{e_i}, Y^{e_i})}} \left[ \max_{i \in [N]} \sup_{f \in \mathcal{F}} \mathcal{R}^{e_i}(f) - \widehat{R}^{e_i}(f) \leq t_{n, \frac{\delta}{2N}, \mathcal{F}} \right] \leq \delta/2,$$

It follows that, with probability at least $1 - \delta/2$, for all $f \in \mathcal{F}$ and $t \in \mathbb{R}$,

$$\widehat{F}_f\left(t + t_{n,\frac{\delta}{2N},\mathcal{F}}\right) \leq \widehat{F}_{\widehat{\mathcal{R}}^{e_1}(f),...,\widehat{\mathcal{R}}^{e_1}(f)}(t),$$

where $\widehat{F}_{\widehat{\mathcal{R}}^{e_1}(f),...,\widehat{\mathcal{R}}^{e_1}(f)}(t)$ is the actually empirical estimate $\widehat{F}_f(t)$ of computed using the $N$ empirical risks $\widehat{\mathcal{R}}^{e_1}(f),...,\widehat{\mathcal{R}}^{e_N}(f)$. Plugging this into the left-hand side of Inequality (D.13),

$$\Pr_{e_1,...,e_N}\left[\sup_{f \in \mathcal{F}, t \in \mathbb{R}} F_f\left(t + t_{n,\frac{\delta}{2N},\mathcal{F}}\right) - \widehat{F}_{\widehat{\mathcal{R}}^{e_1}(f),...,\widehat{\mathcal{R}}^{e_1}(f)}(t) > \epsilon + \text{Bias}(\mathcal{F},\widehat{F})\right] \leq 8\mathcal{N}_{\epsilon/16} \exp\left(-\frac{N\epsilon}{64L}\right).$$

Setting $t = \widehat{F}^{-1}_{\widehat{\mathcal{R}}^{e_1}(f),...,\widehat{\mathcal{R}}^{e_1}(f)}(\alpha)$ and applying the non-decreasing function $F_f^{-1}$ gives the desired result:

$$\Pr_{e_1,...,e_N}\left[\sup_{f \in \mathcal{F}, t \in \mathbb{R}} F_f^{-1}\left(\alpha - \epsilon - \text{Bias}(\mathcal{F},\widehat{F})\right) - \widehat{F}^{-1}_{\widehat{\mathcal{R}}^{e_1}(f),...,\widehat{\mathcal{R}}^{e_1}(f)}(\alpha) + \geq t_{n,\frac{\delta}{2N},\mathcal{F}}\right] \leq 8\mathcal{N}_{\epsilon/16} \exp\left(-\frac{N\epsilon}{64L}\right).$$

$\square$

## D.2   Kernel density estimator

In this section, we apply our generalization bound Theorem (D.1) to the kernel density estimator (KDE)

$$\widehat{F}_h(t) = \int_{-\infty}^t \frac{1}{nh} \sum_{i=1}^n K\left(\frac{\tau - X_i}{h}\right) d\tau$$

of the cumulative risk distribution under the assumptions that:

1. the loss $\ell$ takes values in a bounded interval $[a,b] \subseteq \mathbb{R}$, and
2. for all $f \in \mathcal{F}$, the true risk profile $F_f$ is $\beta$-Hölder continuous with constant $L$, for any $\beta > 0$.

We also make standard integrability and symmetry assumptions on the kernel $K : \mathbb{R} \to \mathbb{R}$ (see Section 1.2.2 [133] for discussion of these assumptions):

$$\int_{\mathbb{R}} |K(u)| \, du < \infty, \quad \int_{\mathbb{R}} K(u) \, du = 1, \quad \int_{\mathbb{R}} |u|^{\beta}|K(u) \, du < \infty,$$

and, for each positive integer $j < \beta$,

$$\int_{\mathbb{R}} u^j K(u) \, du = 0. \tag{D.14}$$

We will use Theorem D.1 to show that, for an appropriately chosen bandwidth $h$,

$$\sup_{f \in \mathcal{F}, t \in \mathbb{R}} F_f(t) - \widehat{F}_f(t) \in O_P\left(\left(\frac{\log N}{N}\right)^{\frac{\beta}{2\beta+1}}\right).$$

We start by bounding the bias term $B(\mathcal{F},\widehat{F})$. Since

$$\mathbb{E}_{X_1,...,X_n}\left[\int_{-\infty}^t \left|\frac{1}{nh} \sum_{i=1}^n K\left(\frac{\tau - X_i}{h}\right)\right|\right] d\tau \leq \frac{1}{h} \mathbb{E}_X\left[\int_{-\infty}^{\infty} \left|K\left(\frac{\tau - X_i}{h}\right)\right|\right] d\tau$$

$$\leq \|K\|_1 < \infty,$$

applying Fubini's theorem, linearity of expectation, the change of variables $x \mapsto \tau + xh$, Fubini's theorem again, and the fact that $\int_{\mathbb{R}} K(u)\, dx = 1$,

$$
\begin{aligned}
F_f(t) - \mathbb{E}_{X_1,\ldots,X_n}\left[\widehat{F}_h(t)\right] &= F_f(t) - \mathbb{E}_{e_1,\ldots,e_N}\left[\int_{-\infty}^t \frac{1}{nh}\sum_{i=1}^n K\left(\frac{\tau - X_i}{h}\right)\right] \\
&= F_f(t) - \int_{-\infty}^t \mathbb{E}_{X_1,\ldots,X_n}\left[\frac{1}{nh}\sum_{i=1}^n K\left(\frac{\tau - X_i}{h}\right)\right] \\
&= F_f(t) - \int_{-\infty}^t \int_{\mathbb{R}} \frac{1}{h}K\left(\frac{\tau - x}{h}\right) p(x)\, dx\, d\tau \\
&= F_f(t) - \int_{-\infty}^t \int_{\mathbb{R}} K(x)p(\tau + xh)\, dx\, d\tau \\
&= F_f(t) - \int_{\mathbb{R}} K(x) \int_{-\infty}^t p(\tau + xh)\, d\tau\, dx \\
&= \int_{\mathbb{R}} K(x)\left(F_f(t) - F(t + xh)\right) dx.
\end{aligned}
$$

By Taylor's theorem for some $\pi \in [0,1]$,

$$
F(t + xh) = \sum_{j=0}^{\lfloor \beta \rfloor - 1} \frac{(xh)^j}{j!}\frac{d^j}{dt^j}F_f(t) + \frac{(xh)^{\lfloor \beta \rfloor}}{\lfloor \beta \rfloor!}\frac{d^{\lfloor \beta \rfloor}}{dt^{\lfloor \beta \rfloor}}F(t + \pi xh).
$$

Hence, by the assumption (D.14),

$$
\begin{aligned}
F_f(t) - \mathbb{E}_{X_1,\ldots,X_n}\left[\widehat{F}_h(t)\right] &= \int_{\mathbb{R}} K(x)\left(F_f(t) - \sum_{j=0}^{\lfloor \beta \rfloor - 1}\frac{(xh)^j}{j!}\frac{d^j}{dt^j}F_f(t) + \frac{(xh)^{\lfloor \beta \rfloor}}{\lfloor \beta \rfloor!}\frac{d^{\lfloor \beta \rfloor}}{dt^{\lfloor \beta \rfloor}}F(t + \pi xh)\right) dx \\
&= \int_{\mathbb{R}} K(x)\left(\frac{(xh)^{\lfloor \beta \rfloor}}{\lfloor \beta \rfloor!}\frac{d^{\lfloor \beta \rfloor}}{dt^{\lfloor \beta \rfloor}}F(t + \pi xh)\right) dx \\
&= \int_{\mathbb{R}} K(x)\frac{(xh)^{\lfloor \beta \rfloor}}{\lfloor \beta \rfloor!}\left(\frac{d^{\lfloor \beta \rfloor}}{dt^{\lfloor \beta \rfloor}}F(t + \pi xh) - \frac{d^{\lfloor \beta \rfloor}}{dt^{\lfloor \beta \rfloor}}F_f(t)\right) dx.
\end{aligned}
$$

Thus, by the Hölder continuity assumption,

$$
\begin{aligned}
\left|F_f(t) - \mathbb{E}_{X_1,\ldots,X_n}\left[\widehat{F}_h(t)\right]\right| &\leq \int_{\mathbb{R}} K(x)\frac{(xh)^{\lfloor \beta \rfloor}}{\lfloor \beta \rfloor!}\left|\frac{d^{\lfloor \beta \rfloor}}{dt^{\lfloor \beta \rfloor}}F(t + \pi xh) - \frac{d^{\lfloor \beta \rfloor}}{dt^{\lfloor \beta \rfloor}}F_f(t)\right| dx \\
&\leq \int_{\mathbb{R}} K(x)\frac{(xh)^{\lfloor \beta \rfloor}}{\lfloor \beta \rfloor!}L(\pi xh)^{\beta - \lfloor \beta \rfloor}\, dx \leq Ch^{\beta}, \qquad \text{(D.15)}
\end{aligned}
$$

where $C := \frac{L}{\lfloor \beta \rfloor!}\int_{\mathbb{R}} |x|^{\beta}|K(x)|\, dx$ is a constant.

Next, since, by the Fundamental Theorem of Calculus,

$$
\frac{d^{\lfloor \beta + 1 \rfloor}}{dt^{\lfloor \beta + 1 \rfloor}}\widehat{F}_f(t) = \frac{d^{\lfloor \beta + 1 \rfloor}}{dt^{\lfloor \beta + 1 \rfloor}}\int_{-\infty}^t \frac{1}{nh}\sum_{i=1}^N K\left(\frac{\tau - X_i}{h}\right) d\tau = \frac{1}{nh}\sum_{i=1}^N \frac{d^{\lfloor \beta \rfloor}}{dt^{\lfloor \beta \rfloor}}K\left(\frac{t - X_i}{h}\right),
$$

$\|F_f\|_{\mathcal{C}^{\beta+1}} \leq \|K_h\|_{\mathcal{C}^{\beta}} = h^{-(\beta+1)}\|K\|_{\mathcal{C}^{\beta}}$. Hence, by standard bounds on the covering number of Hölder continuous functions [134], there exists a constant $c > 0$ depending only on $\beta$ such that

$$
\mathcal{N}_{\epsilon/16}(\mathcal{N}) \leq \exp\left(c(b-a)\left(\frac{\|K\|_{\mathcal{C}^{\beta}}}{h^{\beta+1}\epsilon}\right)^{\frac{1}{\beta+1}}\right) = \exp\left(c\frac{(b-a)}{h}\left(\frac{\|K\|_{\mathcal{C}^{\beta}}}{\epsilon}\right)^{\frac{1}{\beta+1}}\right). \qquad \text{(D.16)}
$$

Finally, since $\widehat{F}_h = \widehat{Q} * K_h$ (where $*$ denotes convolution), by linearity of the convolution and Young's convolution inequality [135, p.34],

$$
\left\|\widehat{F}_h - \widehat{F}'_h\right\|_{\infty} \leq \left\|\widehat{Q} - \widehat{Q}'\right\|_{\infty}\|K_h\|_1.
$$

Since, by a change of variables, $\|K_h\|_1 = \|K\|_1 = 1$, the KDE is a 1-Lipschitz function of the empirical CDF, under $\mathcal{L}_\infty(\mathbb{R})$.

Thus, plugging Inequality (D.15), Inequality (D.16), and $L = 1$ into Theorem D.1 and taking $n \to \infty$ gives, for any $\epsilon > 0$,

$$\Pr_{e_1, \ldots, e_N} \left[ \sup_{f \in \mathcal{F}} F_f^{-1} \left( \alpha - Ch^\beta - \epsilon \right) - \widehat{F}_f^{-1}(\alpha) > 0 \right] \leq 8 \exp \left( c \frac{b-a}{h} \left( \frac{\|K\|_{\mathcal{C}^\beta}}{\epsilon} \right)^{\frac{1}{\beta+1}} \right) e^{-\frac{N\epsilon^2}{64}}.$$

Plugging in $\epsilon = \sqrt{\frac{\log \frac{1}{\delta} + c \frac{b-a}{h}}{N}}$ gives

$$\Pr_{e_1, \ldots, e_N} \left[ \sup_{f \in \mathcal{F}} F_f^{-1} \left( \alpha - Ch^\beta - \sqrt{\frac{\log \frac{1}{\delta} + c \frac{b-a}{h}}{N}} \right) - \widehat{F}_f^{-1}(\alpha) > 0 \right] \leq \delta.$$

This bound is optimized by $h \asymp \left( (b-a) \frac{\log N}{N} \right)^{\frac{1}{2\beta+1}}$, giving an overall bound of

$$\Pr_{e_1, \ldots, e_N} \left[ \sup_{f \in \mathcal{F}, t \in \mathbb{R}} F_f(t) - \widehat{F}_f(t) > ch^{\frac{\beta}{2\beta+1}} \right] \leq \delta$$

$$\Pr_{e_1, \ldots, e_N} \left[ \sup_{f \in \mathcal{F}} F_f^{-1} \left( \alpha - ch^{\frac{\beta}{2\beta+1}} + \sqrt{\frac{\log \frac{1}{\delta}}{N}} \right) - \widehat{F}_f^{-1}(\alpha) > 0 \right] \leq \delta.$$

for some $c > 0$. In particular, as $N, n \to \infty$, the EQRM estimate $\widehat{f}$ satisfies

$$F_{\widehat{f}}^{-1}(\alpha) \to \inf_{f \in \mathcal{F}} F_f^{-1}(\alpha).$$

# E Further implementation details

## E.1 Algorithm

Below we detail the EQRM algorithm. Note that: (i) any distribution estimator may be used in place of DIST so long as the functions DIST.ESTIMATE_PARAMS and DIST.ICDF are differentiable; (ii) other bandwidth-selection methods may be used on line 14, with the Gaussian-optimal rule serving as the default; and (iii) the bisection method BISECT on line 20 requires an additional parameter, the maximum number of steps, which we always set to 32.

---

**Algorithm 1:** Empirical Quantile Risk Minimization (EQRM).

---

**Input:** Predictor $f_\theta$, loss function $\ell$, desired probability of generalization $\alpha$, learning rate $\eta$, distribution estimator DIST, $M$ datasets with $D^m = \{(x_i^m, y_i^m)\}_{i=1}^{n_m}$.

1 Initialize $f_\theta$;
2 **while** *not converged* **do**

    /* Get per-domain risks (i.e. average losses)      */
3     $L^m \leftarrow \frac{1}{n_m} \sum_{i=1}^{n_m} \ell(f_\theta(x_i^m), y_i^m)$, for $m = 1, \ldots, M$ ;

    /* Estimate the parameters of $\widehat{\mathbb{T}}_f$      */
4     DIST.ESTIMATE_PARAMS($\mathbf{L}$) ;

    /* Compute the $\alpha$-quantile of $\widehat{\mathbb{T}}_f$      */
5     $q \leftarrow$ DIST.ICDF($\alpha$) ;

    /* Update $f_\theta$      */
6     $\theta \leftarrow \theta - \eta \cdot \nabla_\theta q$ ;

**Output:** $f_\theta$

7 **Procedure** GAUSS.ESTIMATE_PARAMS*($\mathbf{L}$)*

    /* Compute the sample mean and variance      */
8     $\hat{\mu} \leftarrow \frac{1}{M} \sum_{m=1}^{M} L^m$ ;
9     $\hat{\sigma}^2 \leftarrow \frac{1}{M-1} \sum_{m=1}^{M} (L^m - \hat{\mu})^2$ ;

10 **Procedure** GAUSS.ICDF*($\alpha$)*
11     **return** $\hat{\mu} + \hat{\sigma} \cdot \Phi^{-1}(\alpha)$;

12 **Procedure** KDE.ESTIMATE_PARAMS*($\mathbf{L}$)*

    /* Set bandwidth $h$ (Gaussian-optimal rule used as default)      */
13     $\hat{\sigma}^2 \leftarrow \frac{1}{M-1} \sum_{m=1}^{M} (L^m - \frac{1}{M} \sum_{j=1}^{M} L^j)^2$;
14     $h \leftarrow (\frac{4}{3M})^{0.2} \cdot \hat{\sigma}$

15 **Procedure** KDE.ICDF*($\alpha$)*

    /* Define the CDF when using $M$ Gaussian kernels      */
16     $F_m(x') \leftarrow L^m + h \cdot \Phi(x')$ ;
17     $F(x') \leftarrow \frac{1}{M} \sum_{m=1}^{M} F_m(x')$ ;

    /* Invert the CDF via bisection      */
18     mn $\leftarrow \min_m F_m^{-1}(\alpha)$ ;
19     mx $\leftarrow \max_m F_m^{-1}(\alpha)$ ;
20     **return** BISECT($F, \alpha$, mn, mx) ;

---

## E.2 ColoredMNIST

For the CMNIST results of § 6.1, we used full batches (size 25000), 400 steps for ERM pretraining, 600 total steps for IRM, VREx, EQRM, and 1000 total steps for GroupDRO, SD, and IGA. We used the original MNIST training set to create training and validation sets for each domain, and the original MNIST test set for the test sets of each domain. We also decayed the learning rate using cosine annealing/scheduling. We swept over penalty weights in $\{50, 100, 500, 1000, 5000\}$ for IRM,

VREx and IGA, penalty weights in $\{0.001, 0.01, 0.1, 1\}$ for SD, $\eta$'s in $\{0.001, 0.01, 0.1, 0.5, 1.0\}$ for GroupDRO, and $\alpha$'s in $1 - \{e^{-100}, e^{-250}, e^{-500}, e^{-750}, e^{-1000}\}$ for EQRM. To allow these values of $\alpha$, which are *very* close to 1, we used an asymptotic expression for the Normal inverse CDF, namely $\Phi^{-1}(\alpha) \approx \sqrt{-2\ln(1-\alpha)}$ as $\alpha \to 1$ [136]. This allowed us to parameterize $\alpha = 1 - e^{-1000}$ as $\ln(1-\alpha) = \ln(e^{-1000}) = -1000$, avoiding issues with floating-point precision. As is the standard for CMNIST, we used a test-domain validation set to select the best settings (after the total number of steps), then reported the mean and standard deviation over 10 random seeds on a test-domain test set. As in previous works, the hyperparameter ranges of all methods were selected by peeking at test-domain performance. While not ideal, this is quite difficult to avoid with CMNIST and highlights the problem of model selection more generally in DG—as discussed by many previous works [9, 38, 41, 115]. Finally, we note several observations from our CMNIST, WILDS and DomainBed experiments which, despite not being thoroughly investigated with their own set of experiments (yet), may prove useful for future work: (i) ERM pretraining seems an effective strategy for DG methods, and can likely replace the more delicate penalty-annealing strategies (as also observed in [115]); (ii) lowering the learning rate after ERM pretraining seems to stabilize DG methods; and (iii) EQRM often requires a lower learning rate than other DG methods after ERM pretraining, with its loss and gradients tending to be significantly larger.

## E.3 DomainBed

For EQRM, we used the default algorithm setup: a kernel-density estimator of the risk distribution with the "Gaussian-optimal" rule [65] for bandwidth selection. We used the standard hyperparameter-sampling procedure of Domainbed, running over 3 trials for 20 randomly-sampled hyperparameters per trial. For EQRM, this involved:

| Hparam | Default | Sampling |
|---|---|---|
| $\alpha$ | 0.75 | $U(0.5, 0.99)$ |
| Burn-in/anneal iters | 2500 | $10^k$, with $k \sim U(2.5, 3.5)$ |
| EQRM learning rate (post burn-in) | $10^{-6}$ | $10^k$, with $k \sim U(-7, -5)$ |

All other all hyperparameters remained as their DomainBed-defaults, while the baseline results were taken directly from the most up-to-date DomainBed tables[8]. See our code for further details.

## E.4 WILDS

We considered two WILDS datasets: iWildCam and OGB-MolPCBA (henceforth OGB). For both of these datasets, we used the architectures use in the original WILDS paper [12]; that is, for iWildCam we used a ResNet-50 architecture [137] pretrained on ImageNet [138], and for OGB, we used a Graph Isomorphism Network [139] combined with virtual nodes [140]. To perform model-selection, we followed the guidelines provided in the original WILDS paper [12]. In particular, for each of the baselines we consider, we performed grid searches over the hyperparameter ranges listed in [12] with respect to the given validation sets; see [12, Appendices E.1.2 and E.4.2] for a full list of these hyperparameter ranges.

**EQRM.** For both datasets, we ran EQRM with KDE using the Gaussian-optimal bandwidth-selection method. All EQRM models were initialized with the same ERM checkpoint, which is obtained by training ERM using the code provided by [12]. Following [12], for iWildCam, we trained ERM for 12 epochs, and for OGB, we trained ERM for 100 epochs. We again followed [12] by using a batch size of 32 for iWildCam and 8 groups per batch. For OGB, we performed grid searches over the batch size in the range $B \in \{32, 64, 128, 256, 512, 1024, 2048\}$, and we used $0.25B$ groups per batch. We selected the learning rate for EQRM from $\eta \in \{10^{-2}, 10^{-3}, 10^{-4}, 10^{-5}, 10^{-6}, 10^{-7}, 10^{-8}\}$.

**Computational resources.** All experiments on the WILDS datasets were run across two four-GPU workstations, comprising a total of eight Quadro RTX 5000 GPUs.

---

[8]https://github.com/facebookresearch/DomainBed/tree/main/domainbed/results/2020_10_06_7df6f06

# F   Connections between QRM and DRO

In this appendix we draw connections between quantile risk minimization (QRM) and distributionally robust optimization (DRO) by considering an alternative optimization problem which we call *superquantile risk minimization* [9]:

$$\min_{f \in \mathcal{F}} SQ_\alpha(R; \mathbb{T}_f) \qquad \text{where} \qquad SQ_\alpha(R; \mathbb{T}_f) := \mathbb{E}_{R \sim \mathbb{T}_f} \left[ R \mid R \geq F_{\mathbb{T}_f}^{-1}(\alpha) \right]. \qquad \text{(F.1)}$$

Here, $SQ_\alpha$ represents the *superquantile*—also known as the *conditional value-at-risk* (CVaR) or *expected tail loss*—at level $\alpha$, which can be seen as the conditional expectation of a random variable $R$ subject to $R$ being larger than the $\alpha$-quantile $F^{-1}(\alpha)$. In our case, where $R$ represents the statistical risk on a randomly-sampled environment, $SQ_\alpha$ can be seen as the expected risk in the worst $100 \cdot (1 - \alpha)\%$ of cases/domains. Below, we exploit the well-known duality properties of CVaR to formally connect (QRM) and GroupDRO [45]; see Prop. F.1 for details.

## F.1   Notation for this appendix

Throughout this appendix, for each $f \in \mathcal{F}$, we will let the risk random variable $R$ be a defined on the probability space $(\mathbb{R}_+, \mathcal{B}, \mathbb{T}_f)$, where $\mathbb{R}_+$ denotes the nonnegative real numbers and $\mathcal{B}$ denotes the Borel $\sigma$-algebra on $\mathbb{R}_+$. We will also consider the Lebesgue spaces $L^p := L^p(\mathbb{R}_+, \mathcal{B}, \mathbb{T}_f)$ of functions $h$ for which $\mathbb{E}_{r \sim \mathbb{T}_f}[|h(r)|^p]$ is finite. For conciseness, we will use the notation

$$\langle g(r), h(r) \rangle := \int_{r \geq 0} g(r) h(r) \mathrm{d}r \qquad \text{(F.2)}$$

to denote the standard inner product on $\mathbb{R}_+$. Furthermore, we will use the notation $\mathbb{U} \ll \mathbb{V}$ to signify that $\mathbb{U}$ is *absolutely continuous* with respect to $\mathbb{V}$, meaning that if $\mathbb{U}(A) = 0$ for every set $A$ for which $\mathbb{V}(A) = 0$. We also use the abbreviation "a.e." to mean "almost everywhere." Finally, the notation $\Pi_{[a,b]}(c)$ denotes the projection of a number $c$ into the real interval $[a, b]$.

## F.2   (Strong) Duality of the superquantile

We begin by proving that strong duality holds for the superquantile function $SQ_\alpha$. We note that this duality result is well-known in the literature (see, e.g., [90]), and has been exploited in the context of adaptive sampling [94] and offline reinforcement learning [141]. We state this result and proof for the sake of exposition.

**Proposition F.1** (Dual representation of $SQ_\alpha$). *If $R \in L^P$ for some $p \in (1, \infty)$, then*

$$SQ_\alpha(R; \mathbb{T}_f) = \max_{\mathbb{U} \in \mathcal{U}_f(\alpha)} \mathbb{E}_{\mathbb{U}}[R] \qquad \text{(F.3)}$$

*where the uncertainty set $\mathcal{U}_f(\alpha)$ is defined as*

$$\mathcal{U}_f(\alpha) := \left\{ \mathbb{U} \in L^q : \mathbb{U} \ll \mathbb{T}_f, \mathbb{U} \in [0, 1/1-\alpha] \ a.e. , ||U||_{L^1} = 1 \right\}. \qquad \text{(F.4)}$$

*Proof.* Note that the primal objective can be equivalently written as

$$SQ_\alpha(R; \mathbb{T}_f) = \min_{t \in \mathbb{R}} \left\{ t + \frac{1}{1 - \alpha} \langle (R - t)_+, \mathbb{T}_f \rangle \right\} \qquad \text{(F.5)}$$

where $(z)_+ = \max\{0, z\}$ [97], which in turn has the following epigraph form:

$$\min_{t \in \mathbb{R}, \ s \in L_+^p} \quad t + \frac{1}{1 - \alpha} \langle s, \mathbb{T}_f \rangle \qquad \text{(F.6)}$$

$$\text{subject to} \quad R(r) - t \leq s(r) \ \text{a.e. } r \in \mathbb{R}_+. \qquad \text{(F.7)}$$

---

[9]This definition assumes that $\mathbb{T}_f$ is continuous; for a more general treatment, see [97].

When written in Lagrangian form, we can express this problem as

$$\min_{t \in \mathbb{R},\ s \in L^p_+} \max_{\lambda \in L^q_+} \left\{ t(1 - \langle 1, \lambda \rangle) + \left\langle s, \frac{1}{1-\alpha} \mathbb{T}_f - \lambda \right\rangle + \langle R, \lambda \rangle \right\}. \tag{F.8}$$

Note that this objective is *linear* in $t$, $s$, and $\lambda$, and therefore due to the strong duality of linear programs, we can optimize over $s$, $t$, and $\lambda$ in any order [142]. Minimizing over $t$ reveals that the problem is unbounded unless $\int_{r \geq 0} \lambda(r) \mathrm{d}r = 1$, meaning that $\lambda$ is a probability distribution since $\lambda(r) \geq 0$ almost everywhere. Thus, the problem can be written as

$$\min_{s \in L^p_+} \max_{\lambda \in \mathcal{P}(\mathbb{R}_+)} \left\{ \left\langle s, \frac{1}{1-\alpha} \mathbb{T}_f - \lambda \right\rangle + \langle R, \lambda \rangle \right\} \tag{F.9}$$

where $\mathcal{P}^q(\mathbb{R}_+)$ denotes the subspace of $L^q$ of probability distributions on $\mathbb{R}_+$.

Now consider the maximization over $s$. Note that if there is a set $A \subset \mathcal{E}_{\text{all}}$ of nonzero Lebesgue measure on which $\lambda(A) \geq (1/1-\alpha)\mathbb{T}_f(A)$, then the problem is unbounded below because $s(A)$ can be made arbitrarily large. Therefore, it must be the case that $\lambda \leq (1/1-\alpha)\mathbb{T}_f$ almost everywhere. On the other hand, if $\lambda(A) \leq (1/1-\alpha)\mathbb{T}_f(A)$, then $s(A) = 0$ minimizes the first term in the objective. Therefore, $s$ can be eliminated provided that $\lambda \leq (1/1-\alpha)\mathbb{T}_f$ almost everywhere. Thus, we can write the problem as

$$\max_{\lambda \in \mathcal{P}^q(\mathbb{R}_+)} \quad \langle R, \lambda \rangle = \mathbb{E}_\lambda[R] \tag{F.10}$$

$$\text{subject to} \quad \lambda(r) \leq \frac{1}{1-\alpha} \mathbb{T}_f(r) \ \text{a.e.}\ r \geq 0. \tag{F.11}$$

Now observe that the constraint in the above problem is equivalent to $\lambda \ll \mathbb{Q}$. Thus, by defining $\mathbb{U} = \mathrm{d}\lambda/\mathrm{d}\mathbb{T}_f$ to be the Radon-Nikodym derivative of $\lambda$ with respect to $\mathbb{Q}$, we can write the problem in the form of (F.3), completing the proof. $\qquad\square$

Succinctly, this proposition shows that provided that $R$ is sufficiently smooth (i.e., an element of $L^p$), it holds that minimizing the superquantile function is equivalent to solving

$$\min_{f \in \mathcal{F}} \max_{\mathbb{U} \in \mathcal{U}_f(\alpha)} \mathbb{E}_\mathbb{U}[R] \tag{F.12}$$

which is a distributionally robust optimization (DRO) problem with uncertainty set $\mathcal{U}_f(\alpha)$ as defined in (F.4). In plain terms, for any $\alpha \in (0, 1)$, this uncertainty set contains probability distributions on $\mathbb{R}_+$ which can place no larger than $1/1-\alpha$ on any risk value.

At an intuitive level, this shows that by varying $\alpha$ in Eq. (F.1), one can interpolate between a range DRO problems. In particular, at level $\alpha = 1$, we recover the problem in (3.1), which can be viewed as a DRO problem which selects a Dirac distribution which places solely on the essential supremum of $R \sim \mathbb{T}_f$. On the other hand, at level $\alpha = 0$, we recover a problem which selects a distribution that equally weights each of the risks in different domains equally. A special case of this is the GroupDRO formulation in [45], wherein under the assumption that the data is partitioned into $m$ groups, the inner maximum in (F.12) is taken over the $(m - 1)$-dimensional simplex $\Delta_m$ (see, e.g., equation (7) in [45]).

# G  Additional analyses and experiments

## G.1  Linear regression

In this section we extend § 6.1 to provide further analyses and discussion of EQRM using linear regression datasets based on Ex. A.3. In particular, we: (i) extend Fig. 3 to include plots of the predictors' risk CDFs (G.1.1); and (ii) discuss the ability of EQRM to recover the causal predictor when $\sigma_1^2$, $\sigma_2^2$ and/or $\sigma_Y^2$ change over environments, compared to IRM [9] and VREx [41] (G.1.2).

Table 5: Recovering the causal predictor for linear regression tasks based on Ex. A.3. A tick means that it is *possible* to recover the causal predictor, under further assumptions.

| Changing | Domain Scedasticity | Invariant Risk | Invariant Function ($\beta_{\text{cause}}$) | IRM | VREx | EQRM |
|---|---|---|---|---|---|---|
| $\sigma_1$ | *Homo*scedastic | ✓ | ✓ | ✓ | ✓ | ✓ |
| $\sigma_2$ | *Homo*scedastic | ✓ | ✓ | ✓ | ✓ | ✓ |
| $\sigma_Y$ | *Hetero*scedastic | ✗ | ✓ | ✓ | ✗ | ✗ |

### G.1.1 Risk CDFs as risk-robustness curves

As an extension of Fig. 3, in particular the PDFs in Fig. 3 **B**, Fig. 6 depicts the risk CDFs for different predictors. Here we see that a predictor's risk CDF depicts its risk-robustness curve, and also that each $\alpha$ results in a predictor $f_\alpha$ with minimal $\alpha$-quantile risk. That is, for each desired level of robustness (i.e. probability of the upper-bound on risk holding, y-axis), the corresponding $\alpha$ has minimal risk (x-axis).

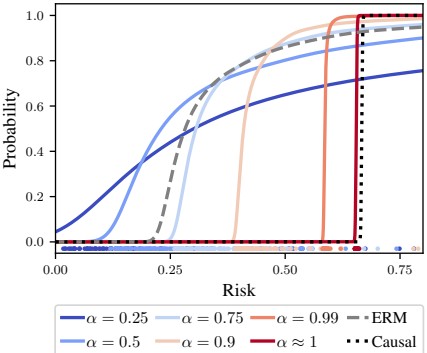

Figure 6: **Extension of Fig. 3 showing the risk CDFs (i.e. risk-robustness curves) for different predictors.** For each risk upper-bound ($x$), we see the corresponding probability of it holding under the training domains ($y$). Note that, for each level of robustness ($y$, i.e. probability that the risk upper-bound holds), the corresponding $\alpha$ has the lowest upper-bound on risk ($x$). Also note that these CDFs correspond to the PDFs of Fig. 3 (**B**).

### G.1.2 Invariant risks vs. invariant functions

We now compare seeking invariant *risks* to seeking invariant *functions* by analyzing linear regression datasets, based on Ex. A.3, in which $\sigma_1^2$, $\sigma_2^2$ and/or $\sigma_Y^2$ change over domains. This is turn allows us to compare EQRM (invariant risks), VREx [41] (invariant risks), and IRM [9] (invariant functions).

**Domain-skedasticity.** For recovering the causal predictor, the key difference between using invariant *risks* and invariant *functions* lies in the assumption about *domain-skedasticity*, i.e. the "predicatability" of $Y$ across domains. In particular, the causal predictor only has invariant risks in *domain-homoskedastic* cases and not in *domain-heteroskedastic* cases, the latter describing scenarios in which the predictability of $Y$ (i.e. the amount of irreducible error or intrinsic noise) varies across domains, meaning that the risk of the causal predictor will be smaller on some domains than others. Thus, methods seeking the causal predictor through invariant risks must assume domain homoskedasticity [41, 54]. In contrast, methods seeking the causal predictor through invariant *functions* need not make such a domain-homoskedasticity assumption, but instead the slightly weaker assumption of the conditional mean $\mathbb{E}[Y|\text{Pa}(Y)]$ being invariant across domains. As explained in the next paragraph and summarized in Table 5, this translates into the coefficient $\beta_{\text{cause}}$ being invariant across domains for the linear SEM of Ex. A.3.

**Mathematical analysis.** We now analyze the risk-invariant solutions of Ex. A.3. We start by expanding the structural equations of Ex. A.3 as:

$$X_1 = N_1,$$
$$Y = N_1 + N_Y,$$
$$X_2 = N_1 + N_Y + N_2.$$

We then note that the goal is to learn a model $\widehat{Y} = \beta_1 \cdot X_1 + \beta_2 \cdot X_2$, which has residual error

$$\widehat{Y} - Y = \beta_1 \cdot N_1 + \beta_2 \cdot (N_1 + N_Y + N_2) - N_1 - N_Y$$
$$= (\beta_1 + \beta_2 - 1) \cdot N_1 + (\beta_2 - 1) \cdot N_Y + \beta_2 \cdot N_2.$$

Then, since all variables have zero mean and the noise terms are independent, the risk (i.e. the MSE loss) is simply the variance of the residuals, which can be written as

$$\mathbb{E}[(\widehat{Y} - Y)^2] = (\beta_1 + \beta_2 - 1)^2 \cdot \sigma_1^2 + (\beta_2 - 1)^2 \cdot \sigma_Y^2 + \beta_2^2 \cdot \sigma_2^2.$$

Here, we have that, when:

- **Only $\sigma_1$ changes:** the only way to keep the risk invariant across domains is to set $\beta_1 + \beta_2 = 1$. The minimal invariant-risk solution then depends on $\sigma_y$ and $\sigma_2$:
  - if $\sigma_y < \sigma_2$, the minimal invariant-risk solution sets $\beta_1 = 1$ and $\beta_2 = 0$ (causal predictor);
  - if $\sigma_y > \sigma_2$, the minimal invariant-risk solution sets $\beta_1 = 0$ and $\beta_2 = 1$ (anti-causal predictor);
  - if $\sigma_y = \sigma_2$, then any solution $(\beta_1, \beta_2) = (c, 1-c)$ with $c \in [0,1]$ is a minimal invariant-risk solution, including the causal predictor $c = 1$, anti-causal predictor $c = 0$, and everything in-between.
- **Only $\sigma_2$ changes:** the invariant-risk solutions set $\beta_2 = 0$, with the minimal invariant-risk solution also setting $\beta_1 = 1$ (causal predictor).
- **$\sigma_1$ and $\sigma_2$ change:** *the* invariant-risk solution sets $\beta_1 = 1, \beta_2 = 0$ (causal predictor).
- **Only $\sigma_Y$ changes:** the invariant-risk solutions set $\beta_2 = 1$, with the minimal invariant-risk solution also setting $\beta_1 = 0$ (anti-causal predictor).
- **$\sigma_1$ and $\sigma_Y$ change:** *the* invariant-risk solution sets $\beta_1 = 0, \beta_2 = 1$ (anti-causal predictor).
- **$\sigma_2$ and $\sigma_Y$ change:** there is no invariant-risk solution.
- **$\sigma_1, \sigma_2$ and $\sigma_Y$ change:** there is no invariant-risk solution.

**Empirical analysis.** To see this empirically, we refer the reader to Table 5 of Krueger et al. [41, App. G.2], which compares the invariant-risk solution of VREx to the invariant-function solution of IRM on the synthetic linear-SEM tasks of Arjovsky et al. [9, Sec. 5.1], which calculate the MSE between the estimated coefficients $(\hat{\beta}_1, \hat{\beta}_2)$ and those of the causal predictor $(1, 0)$.

**Different goals, solutions, and advantages.** We end by emphasizing the fact that the invariant-risk and invariant-function solutions have different pros and cons depending both on the goal and the assumptions made. If the goal is the recover the causal predictor or causes of $Y$, then the invariant-function solution has the advantage due to weaker assumptions on domain skedasticity. However, if the goal is learn predictors with stable or invariant performance, such that they perform well on new domains with high probability, then the invariant-risk solution has the advantage. For example, in the domain-heteroskedastic cases above where $\sigma_Y$ changes or $\sigma_Y$ and $\sigma_1$ change, the invariant-function solution recovers the causal predictor $\beta_1 = 1, \beta_2 = 0$ and thus has arbitrarily-large risk as $\sigma_Y \to \infty$ (i.e. in the worst-case). In contrast, the invariant-risk solution recovers the anti-causal predictor $\beta_1 = 0, \beta_2 = 1$ and thus has fixed risk $\sigma_2^2$ in all domains.

### G.2 DomainBed

In this section, we include the full per-dataset DomainBed results. We consider the two most common model-selection methods of the DomainBed package—training-domain validation set and test-domain validation set (oracle)—and compare EQRM to a range of baselines. Implementation details for these experiments are provided in § E.3 and our open-source code.

### G.2.1 Model selection: training-domain validation set

**VLCS**

| Algorithm | C | L | S | V | Avg |
|---|---|---|---|---|---|
| ERM | 97.7 ± 0.4 | 64.3 ± 0.9 | 73.4 ± 0.5 | 74.6 ± 1.3 | 77.5 |
| IRM | 98.6 ± 0.1 | 64.9 ± 0.9 | 73.4 ± 0.6 | 77.3 ± 0.9 | 78.5 |
| GroupDRO | 97.3 ± 0.3 | 63.4 ± 0.9 | 69.5 ± 0.8 | 76.7 ± 0.7 | 76.7 |
| Mixup | 98.3 ± 0.6 | 64.8 ± 1.0 | 72.1 ± 0.5 | 74.3 ± 0.8 | 77.4 |
| MLDG | 97.4 ± 0.2 | 65.2 ± 0.7 | 71.0 ± 1.4 | 75.3 ± 1.0 | 77.2 |
| CORAL | 98.3 ± 0.1 | 66.1 ± 1.2 | 73.4 ± 0.3 | 77.5 ± 1.2 | 78.8 |
| MMD | 97.7 ± 0.1 | 64.0 ± 1.1 | 72.8 ± 0.2 | 75.3 ± 3.3 | 77.5 |
| DANN | 99.0 ± 0.3 | 65.1 ± 1.4 | 73.1 ± 0.3 | 77.2 ± 0.6 | 78.6 |
| CDANN | 97.1 ± 0.3 | 65.1 ± 1.2 | 70.7 ± 0.8 | 77.1 ± 1.5 | 77.5 |
| MTL | 97.8 ± 0.4 | 64.3 ± 0.3 | 71.5 ± 0.7 | 75.3 ± 1.7 | 77.2 |
| SagNet | 97.9 ± 0.4 | 64.5 ± 0.5 | 71.4 ± 1.3 | 77.5 ± 0.5 | 77.8 |
| ARM | 98.7 ± 0.2 | 63.6 ± 0.7 | 71.3 ± 1.2 | 76.7 ± 0.6 | 77.6 |
| VREx | 98.4 ± 0.3 | 64.4 ± 1.4 | 74.1 ± 0.4 | 76.2 ± 1.3 | 78.3 |
| RSC | 97.9 ± 0.1 | 62.5 ± 0.7 | 72.3 ± 1.2 | 75.6 ± 0.8 | 77.1 |
| EQRM | 98.3 ± 0.0 | 63.7 ± 0.8 | 72.6 ± 1.0 | 76.7 ± 1.1 | 77.8 |

**PACS**

| Algorithm | A | C | P | S | Avg |
|---|---|---|---|---|---|
| ERM | 84.7 ± 0.4 | 80.8 ± 0.6 | 97.2 ± 0.3 | 79.3 ± 1.0 | 85.5 |
| IRM | 84.8 ± 1.3 | 76.4 ± 1.1 | 96.7 ± 0.6 | 76.1 ± 1.0 | 83.5 |
| GroupDRO | 83.5 ± 0.9 | 79.1 ± 0.6 | 96.7 ± 0.3 | 78.3 ± 2.0 | 84.4 |
| Mixup | 86.1 ± 0.5 | 78.9 ± 0.8 | 97.6 ± 0.1 | 75.8 ± 1.8 | 84.6 |
| MLDG | 85.5 ± 1.4 | 80.1 ± 1.7 | 97.4 ± 0.3 | 76.6 ± 1.1 | 84.9 |
| CORAL | 88.3 ± 0.2 | 80.0 ± 0.5 | 97.5 ± 0.3 | 78.8 ± 1.3 | 86.2 |
| MMD | 86.1 ± 1.4 | 79.4 ± 0.9 | 96.6 ± 0.2 | 76.5 ± 0.5 | 84.6 |
| DANN | 86.4 ± 0.8 | 77.4 ± 0.8 | 97.3 ± 0.4 | 73.5 ± 2.3 | 83.6 |
| CDANN | 84.6 ± 1.8 | 75.5 ± 0.9 | 96.8 ± 0.3 | 73.5 ± 0.6 | 82.6 |
| MTL | 87.5 ± 0.8 | 77.1 ± 0.5 | 96.4 ± 0.8 | 77.3 ± 1.8 | 84.6 |
| SagNet | 87.4 ± 1.0 | 80.7 ± 0.6 | 97.1 ± 0.1 | 80.0 ± 0.4 | 86.3 |
| ARM | 86.8 ± 0.6 | 76.8 ± 0.5 | 97.4 ± 0.3 | 79.3 ± 1.2 | 85.1 |
| VREx | 86.0 ± 1.6 | 79.1 ± 0.6 | 96.9 ± 0.5 | 77.7 ± 1.7 | 84.9 |
| RSC | 85.4 ± 0.8 | 79.7 ± 1.8 | 97.6 ± 0.3 | 78.2 ± 1.2 | 85.2 |
| EQRM | 86.5 ± 0.4 | 82.1 ± 0.7 | 96.6 ± 0.2 | 80.8 ± 0.2 | 86.5 |

**OfficeHome**

| Algorithm | A | C | P | R | Avg |
|---|---|---|---|---|---|
| ERM | 61.3 ± 0.7 | 52.4 ± 0.3 | 75.8 ± 0.1 | 76.6 ± 0.3 | 66.5 |
| IRM | 58.9 ± 2.3 | 52.2 ± 1.6 | 72.1 ± 2.9 | 74.0 ± 2.5 | 64.3 |
| GroupDRO | 60.4 ± 0.7 | 52.7 ± 1.0 | 75.0 ± 0.7 | 76.0 ± 0.7 | 66.0 |
| Mixup | 62.4 ± 0.8 | 54.8 ± 0.6 | 76.9 ± 0.3 | 78.3 ± 0.2 | 68.1 |
| MLDG | 61.5 ± 0.9 | 53.2 ± 0.6 | 75.0 ± 1.2 | 77.5 ± 0.4 | 66.8 |
| CORAL | 65.3 ± 0.4 | 54.4 ± 0.5 | 76.5 ± 0.1 | 78.4 ± 0.5 | 68.7 |
| MMD | 60.4 ± 0.2 | 53.3 ± 0.3 | 74.3 ± 0.1 | 77.4 ± 0.6 | 66.3 |
| DANN | 59.9 ± 1.3 | 53.0 ± 0.3 | 73.6 ± 0.7 | 76.9 ± 0.5 | 65.9 |
| CDANN | 61.5 ± 1.4 | 50.4 ± 2.4 | 74.4 ± 0.9 | 76.6 ± 0.8 | 65.8 |
| MTL | 61.5 ± 0.7 | 52.4 ± 0.6 | 74.9 ± 0.4 | 76.8 ± 0.4 | 66.4 |
| SagNet | 63.4 ± 0.2 | 54.8 ± 0.4 | 75.8 ± 0.4 | 78.3 ± 0.3 | 68.1 |
| ARM | 58.9 ± 0.8 | 51.0 ± 0.5 | 74.1 ± 0.1 | 75.2 ± 0.3 | 64.8 |
| VREx | 60.7 ± 0.9 | 53.0 ± 0.9 | 75.3 ± 0.1 | 76.6 ± 0.5 | 66.4 |
| RSC | 60.7 ± 1.4 | 51.4 ± 0.3 | 74.8 ± 1.1 | 75.1 ± 1.3 | 65.5 |
| EQRM | 60.5 ± 0.1 | 56.0 ± 0.2 | 76.1 ± 0.4 | 77.4 ± 0.3 | 67.5 |

**TerraIncognita**

| Algorithm | L100 | L38 | L43 | L46 | Avg |
|---|---|---|---|---|---|
| ERM | 49.8 ± 4.4 | 42.1 ± 1.4 | 56.9 ± 1.8 | 35.7 ± 3.9 | 46.1 |
| IRM | 54.6 ± 1.3 | 39.8 ± 1.9 | 56.2 ± 1.8 | 39.6 ± 0.8 | 47.6 |
| GroupDRO | 41.2 ± 0.7 | 38.6 ± 2.1 | 56.7 ± 0.9 | 36.4 ± 2.1 | 43.2 |
| Mixup | 59.6 ± 2.0 | 42.2 ± 1.4 | 55.9 ± 0.8 | 33.9 ± 1.4 | 47.9 |
| MLDG | 54.2 ± 3.0 | 44.3 ± 1.1 | 55.6 ± 0.3 | 36.9 ± 2.2 | 47.7 |
| CORAL | 51.6 ± 2.4 | 42.2 ± 1.0 | 57.0 ± 1.0 | 39.8 ± 2.9 | 47.6 |
| MMD | 41.9 ± 3.0 | 34.8 ± 1.0 | 57.0 ± 1.9 | 35.2 ± 1.8 | 42.2 |
| DANN | 51.1 ± 3.5 | 40.6 ± 0.6 | 57.4 ± 0.5 | 37.7 ± 1.8 | 46.7 |
| CDANN | 47.0 ± 1.9 | 41.3 ± 4.8 | 54.9 ± 1.7 | 39.8 ± 2.3 | 45.8 |
| MTL | 49.3 ± 1.2 | 39.6 ± 6.3 | 55.6 ± 1.1 | 37.8 ± 0.8 | 45.6 |
| SagNet | 53.0 ± 2.9 | 43.0 ± 2.5 | 57.9 ± 0.6 | 40.4 ± 1.3 | 48.6 |
| ARM | 49.3 ± 0.7 | 38.3 ± 2.4 | 55.8 ± 0.8 | 38.7 ± 1.3 | 45.5 |
| VREx | 48.2 ± 4.3 | 41.7 ± 1.3 | 56.8 ± 0.8 | 38.7 ± 3.1 | 46.4 |
| RSC | 50.2 ± 2.2 | 39.2 ± 1.4 | 56.3 ± 1.4 | 40.8 ± 0.6 | 46.6 |
| EQRM | 47.9 ± 1.9 | 45.2 ± 0.3 | 59.1 ± 0.3 | 38.8 ± 0.6 | 47.8 |

**DomainNet**

| Algorithm | clip | info | paint | quick | real | sketch | Avg |
|---|---|---|---|---|---|---|---|
| ERM | 58.1 ± 0.3 | 18.8 ± 0.3 | 46.7 ± 0.3 | 12.2 ± 0.4 | 59.6 ± 0.1 | 49.8 ± 0.4 | 40.9 |
| IRM | 48.5 ± 2.8 | 15.0 ± 1.5 | 38.3 ± 4.3 | 10.9 ± 0.5 | 48.2 ± 5.2 | 42.3 ± 3.1 | 33.9 |
| GroupDRO | 47.2 ± 0.5 | 17.5 ± 0.4 | 33.8 ± 0.5 | 9.3 ± 0.3 | 51.6 ± 0.4 | 40.1 ± 0.6 | 33.3 |
| Mixup | 55.7 ± 0.3 | 18.5 ± 0.5 | 44.3 ± 0.5 | 12.5 ± 0.4 | 55.8 ± 0.3 | 48.2 ± 0.5 | 39.2 |
| MLDG | 59.1 ± 0.2 | 19.1 ± 0.3 | 45.8 ± 0.7 | 13.4 ± 0.3 | 59.6 ± 0.2 | 50.2 ± 0.4 | 41.2 |
| CORAL | 59.2 ± 0.1 | 19.7 ± 0.2 | 46.6 ± 0.3 | 13.4 ± 0.4 | 59.8 ± 0.2 | 50.1 ± 0.6 | 41.5 |
| MMD | 32.1 ± 13.3 | 11.0 ± 4.6 | 26.8 ± 11.3 | 8.7 ± 2.1 | 32.7 ± 13.8 | 28.9 ± 11.9 | 23.4 |
| DANN | 53.1 ± 0.2 | 18.3 ± 0.1 | 44.2 ± 0.7 | 11.8 ± 0.1 | 55.5 ± 0.4 | 46.8 ± 0.6 | 38.3 |
| CDANN | 54.6 ± 0.4 | 17.3 ± 0.1 | 43.7 ± 0.9 | 12.1 ± 0.7 | 56.2 ± 0.4 | 45.9 ± 0.5 | 38.3 |
| MTL | 57.9 ± 0.5 | 18.5 ± 0.4 | 46.0 ± 0.1 | 12.5 ± 0.1 | 59.5 ± 0.3 | 49.2 ± 0.1 | 40.6 |
| SagNet | 57.7 ± 0.3 | 19.0 ± 0.2 | 45.3 ± 0.3 | 12.7 ± 0.5 | 58.1 ± 0.5 | 48.8 ± 0.2 | 40.3 |
| ARM | 49.7 ± 0.3 | 16.3 ± 0.5 | 40.9 ± 1.1 | 9.4 ± 0.1 | 53.4 ± 0.4 | 43.5 ± 0.4 | 35.5 |
| VREx | 47.3 ± 3.5 | 16.0 ± 1.5 | 35.8 ± 4.6 | 10.9 ± 0.3 | 49.6 ± 4.9 | 42.0 ± 3.0 | 33.6 |
| RSC | 55.0 ± 1.2 | 18.3 ± 0.5 | 44.4 ± 0.6 | 12.2 ± 0.2 | 55.7 ± 0.7 | 47.8 ± 0.9 | 38.9 |
| EQRM | 56.1 ± 1.3 | 19.6 ± 0.1 | 46.3 ± 1.5 | 12.9 ± 0.3 | 61.1 ± 0.0 | 50.3 ± 0.1 | 41.0 |

**Averages**

| Algorithm | VLCS | PACS | OfficeHome | TerraIncognita | DomainNet | Avg |
|---|---|---|---|---|---|---|
| ERM | 77.5 ± 0.4 | 85.5 ± 0.2 | 66.5 ± 0.3 | 46.1 ± 1.8 | 40.9 ± 0.1 | 63.3 |
| IRM | 78.5 ± 0.5 | 83.5 ± 0.8 | 64.3 ± 2.2 | 47.6 ± 0.8 | 33.9 ± 2.8 | 61.6 |
| GroupDRO | 76.7 ± 0.6 | 84.4 ± 0.8 | 66.0 ± 0.7 | 43.2 ± 1.1 | 33.3 ± 0.2 | 60.9 |
| Mixup | 77.4 ± 0.6 | 84.6 ± 0.6 | 68.1 ± 0.3 | 47.9 ± 0.8 | 39.2 ± 0.1 | 63.4 |
| MLDG | 77.2 ± 0.4 | 84.9 ± 1.0 | 66.8 ± 0.6 | 47.7 ± 0.9 | 41.2 ± 0.1 | 63.6 |
| CORAL | 78.8 ± 0.6 | 86.2 ± 0.3 | 68.7 ± 0.3 | 47.6 ± 1.0 | 41.5 ± 0.1 | 64.6 |
| MMD | 77.5 ± 0.9 | 84.6 ± 0.5 | 66.3 ± 0.1 | 42.2 ± 1.6 | 23.4 ± 9.5 | 63.3 |
| DANN | 78.6 ± 0.4 | 83.6 ± 0.4 | 65.9 ± 0.6 | 46.7 ± 0.5 | 38.3 ± 0.1 | 62.6 |
| CDANN | 77.5 ± 0.1 | 82.6 ± 0.9 | 65.8 ± 1.3 | 45.8 ± 1.6 | 38.3 ± 0.3 | 62.0 |
| MTL | 77.2 ± 0.4 | 84.6 ± 0.5 | 66.4 ± 0.5 | 45.6 ± 1.2 | 40.6 ± 0.1 | 62.9 |
| SagNet | 77.8 ± 0.5 | 86.3 ± 0.2 | 68.1 ± 0.1 | 48.6 ± 1.0 | 40.3 ± 0.1 | 64.2 |
| ARM | 77.6 ± 0.3 | 85.1 ± 0.4 | 64.8 ± 0.3 | 45.5 ± 0.3 | 35.5 ± 0.2 | 61.7 |
| VREx | 78.3 ± 0.2 | 84.9 ± 0.6 | 66.4 ± 0.6 | 46.4 ± 0.6 | 33.6 ± 2.9 | 61.9 |
| EQRM | 77.8 ± 0.6 | 86.5 ± 0.2 | 67.5 ± 0.1 | 47.8 ± 0.6 | 41.0 ± 0.3 | 64.1 |

### G.2.2 Model selection: test-domain validation set (oracle)

**VLCS**

| Algorithm | C | L | S | V | Avg |
|---|---|---|---|---|---|
| ERM | 97.6 ± 0.3 | 67.9 ± 0.7 | 70.9 ± 0.2 | 74.0 ± 0.6 | 77.6 |
| IRM | 97.3 ± 0.2 | 66.7 ± 0.1 | 71.0 ± 2.3 | 72.8 ± 0.4 | 76.9 |
| GroupDRO | 97.7 ± 0.2 | 65.9 ± 0.2 | 72.8 ± 0.8 | 73.4 ± 1.3 | 77.4 |
| Mixup | 97.8 ± 0.4 | 67.2 ± 0.4 | 71.5 ± 0.2 | 75.7 ± 0.6 | 78.1 |
| MLDG | 97.1 ± 0.5 | 66.6 ± 0.5 | 71.5 ± 0.1 | 75.0 ± 0.9 | 77.5 |
| CORAL | 97.3 ± 0.2 | 67.5 ± 0.6 | 71.6 ± 0.6 | 74.5 ± 0.0 | 77.7 |
| MMD | 98.8 ± 0.0 | 66.4 ± 0.4 | 70.8 ± 0.5 | 75.6 ± 0.4 | 77.9 |
| DANN | 99.0 ± 0.2 | 66.3 ± 1.2 | 73.4 ± 1.4 | 80.1 ± 0.5 | 79.7 |
| CDANN | 98.2 ± 0.1 | 68.8 ± 0.5 | 74.3 ± 0.6 | 78.1 ± 0.5 | 79.9 |
| MTL | 97.9 ± 0.7 | 66.1 ± 0.7 | 72.0 ± 0.4 | 74.9 ± 1.1 | 77.7 |
| SagNet | 97.4 ± 0.3 | 66.4 ± 0.4 | 71.6 ± 0.1 | 75.0 ± 0.8 | 77.6 |
| ARM | 97.6 ± 0.6 | 66.5 ± 0.3 | 72.7 ± 0.6 | 74.4 ± 0.7 | 77.8 |
| VREx | 98.4 ± 0.2 | 66.4 ± 0.7 | 72.8 ± 0.1 | 75.0 ± 1.4 | 78.1 |
| RSC | 98.0 ± 0.4 | 67.2 ± 0.3 | 70.3 ± 1.3 | 75.6 ± 0.4 | 77.8 |
| EQRM | 98.2 ± 0.2 | 66.8 ± 0.8 | 71.7 ± 1.0 | 74.6 ± 0.3 | 77.8 |

**PACS**

| Algorithm | A | C | P | S | Avg |
|---|---|---|---|---|---|
| ERM | 86.5 ± 1.0 | 81.3 ± 0.6 | 96.2 ± 0.3 | 82.7 ± 1.1 | 86.7 |
| IRM | 84.2 ± 0.9 | 79.7 ± 1.5 | 95.9 ± 0.4 | 78.3 ± 2.1 | 84.5 |
| GroupDRO | 87.5 ± 0.5 | 82.9 ± 0.6 | 97.1 ± 0.3 | 81.1 ± 1.2 | 87.1 |
| Mixup | 87.5 ± 0.4 | 81.6 ± 0.7 | 97.4 ± 0.2 | 80.8 ± 0.9 | 86.8 |
| MLDG | 87.0 ± 1.2 | 82.5 ± 0.9 | 96.7 ± 0.3 | 81.2 ± 0.6 | 86.8 |
| CORAL | 86.6 ± 0.8 | 81.8 ± 0.9 | 97.1 ± 0.5 | 82.7 ± 0.6 | 87.1 |
| MMD | 88.1 ± 0.8 | 82.6 ± 0.7 | 97.1 ± 0.5 | 81.2 ± 1.2 | 87.2 |
| DANN | 87.0 ± 0.4 | 80.3 ± 0.6 | 96.8 ± 0.3 | 76.9 ± 1.1 | 85.2 |
| CDANN | 87.7 ± 0.6 | 80.7 ± 1.2 | 97.3 ± 0.4 | 77.6 ± 1.5 | 85.8 |
| MTL | 87.0 ± 0.2 | 82.7 ± 0.8 | 96.5 ± 0.7 | 80.5 ± 0.8 | 86.7 |
| SagNet | 87.4 ± 0.5 | 81.2 ± 1.2 | 96.3 ± 0.8 | 80.7 ± 1.1 | 86.4 |
| ARM | 85.0 ± 1.2 | 81.4 ± 0.2 | 95.9 ± 0.3 | 80.9 ± 0.5 | 85.8 |
| VREx | 87.8 ± 1.2 | 81.8 ± 0.7 | 97.4 ± 0.2 | 82.1 ± 0.7 | 87.2 |
| RSC | 86.0 ± 0.7 | 81.8 ± 0.9 | 96.8 ± 0.7 | 80.4 ± 0.5 | 86.2 |
| EQRM | 88.3 ± 0.6 | 82.1 ± 0.5 | 97.2 ± 0.4 | 81.6 ± 0.5 | 87.3 |

**OfficeHome**

| Algorithm | A | C | P | R | Avg |
|---|---|---|---|---|---|
| ERM | 61.7 ± 0.7 | 53.4 ± 0.3 | 74.1 ± 0.4 | 76.2 ± 0.6 | 66.4 |
| IRM | 56.4 ± 3.2 | 51.2 ± 2.3 | 71.7 ± 2.7 | 72.7 ± 2.7 | 63.0 |
| GroupDRO | 60.5 ± 1.6 | 53.1 ± 0.3 | 75.5 ± 0.3 | 75.9 ± 0.7 | 66.2 |
| Mixup | 63.5 ± 0.2 | 54.6 ± 0.4 | 76.0 ± 0.3 | 78.0 ± 0.7 | 68.0 |
| MLDG | 60.5 ± 0.7 | 54.2 ± 0.5 | 75.0 ± 0.2 | 76.7 ± 0.5 | 66.6 |
| CORAL | 64.8 ± 0.8 | 54.1 ± 0.9 | 76.5 ± 0.4 | 78.2 ± 0.4 | 68.4 |
| MMD | 60.4 ± 1.0 | 53.4 ± 0.5 | 74.9 ± 0.1 | 76.1 ± 0.7 | 66.2 |
| DANN | 60.6 ± 1.4 | 51.8 ± 0.7 | 73.4 ± 0.5 | 75.5 ± 0.9 | 65.3 |
| CDANN | 57.9 ± 0.2 | 52.1 ± 1.2 | 74.9 ± 0.7 | 76.2 ± 0.2 | 65.3 |
| MTL | 60.7 ± 0.8 | 53.5 ± 1.3 | 75.2 ± 0.6 | 76.6 ± 0.6 | 66.5 |
| SagNet | 62.7 ± 0.5 | 53.6 ± 0.5 | 76.0 ± 0.3 | 77.8 ± 0.1 | 67.5 |
| ARM | 58.8 ± 0.5 | 51.8 ± 0.7 | 74.0 ± 0.1 | 74.4 ± 0.2 | 64.8 |
| VREx | 59.6 ± 1.0 | 53.3 ± 0.3 | 73.2 ± 0.5 | 76.6 ± 0.4 | 65.7 |
| RSC | 61.7 ± 0.8 | 53.0 ± 0.9 | 74.8 ± 0.8 | 76.3 ± 0.5 | 66.5 |
| EQRM | 60.0 ± 0.8 | 54.4 ± 0.7 | 76.5 ± 0.4 | 77.2 ± 0.5 | 67.0 |

**TerraIncognita**

| Algorithm | L100 | L38 | L43 | L46 | Avg |
|---|---|---|---|---|---|
| ERM | 59.4 ± 0.9 | 49.3 ± 0.6 | 60.1 ± 1.1 | 43.2 ± 0.5 | 53.0 |
| IRM | 56.5 ± 2.5 | 49.8 ± 1.5 | 57.1 ± 2.2 | 38.6 ± 1.0 | 50.5 |
| GroupDRO | 60.4 ± 1.5 | 48.3 ± 0.4 | 58.6 ± 0.8 | 42.2 ± 0.8 | 52.4 |
| Mixup | 67.6 ± 1.8 | 51.0 ± 1.3 | 59.0 ± 0.0 | 40.0 ± 1.1 | 54.4 |
| MLDG | 59.2 ± 0.1 | 49.0 ± 0.9 | 58.4 ± 0.9 | 41.4 ± 1.0 | 52.0 |
| CORAL | 60.4 ± 0.9 | 47.2 ± 0.5 | 59.3 ± 0.4 | 44.4 ± 0.4 | 52.8 |
| MMD | 60.6 ± 1.1 | 45.9 ± 0.3 | 57.8 ± 0.5 | 43.8 ± 1.2 | 52.0 |
| DANN | 55.2 ± 1.9 | 47.0 ± 0.7 | 57.2 ± 0.9 | 42.9 ± 0.9 | 50.6 |
| CDANN | 56.3 ± 2.0 | 47.1 ± 0.9 | 57.2 ± 1.1 | 42.4 ± 0.8 | 50.8 |
| MTL | 58.4 ± 2.1 | 48.4 ± 0.8 | 58.9 ± 0.6 | 43.0 ± 1.3 | 52.2 |
| SagNet | 56.4 ± 1.9 | 50.5 ± 2.3 | 59.1 ± 0.5 | 44.1 ± 0.6 | 52.5 |
| ARM | 60.1 ± 1.5 | 48.3 ± 1.6 | 55.3 ± 0.6 | 40.9 ± 1.1 | 51.2 |
| VREx | 56.8 ± 1.7 | 46.5 ± 0.5 | 58.4 ± 0.3 | 43.8 ± 0.3 | 51.4 |
| RSC | 59.9 ± 1.4 | 46.7 ± 0.4 | 57.8 ± 0.5 | 44.3 ± 0.6 | 52.1 |
| EQRM | 57.0 ± 1.5 | 49.5 ± 1.2 | 59.0 ± 0.3 | 43.4 ± 0.6 | 52.2 |

**DomainNet**

| Algorithm | clip | info | paint | quick | real | sketch | Avg |
|---|---|---|---|---|---|---|---|
| ERM | 58.6 ± 0.3 | 19.2 ± 0.2 | 47.0 ± 0.3 | 13.2 ± 0.2 | 59.9 ± 0.3 | 49.8 ± 0.4 | 41.3 |
| IRM | 40.4 ± 6.6 | 12.1 ± 2.7 | 31.4 ± 5.7 | 9.8 ± 1.2 | 37.7 ± 9.0 | 36.7 ± 5.3 | 28.0 |
| GroupDRO | 47.2 ± 0.5 | 17.5 ± 0.4 | 34.2 ± 0.3 | 9.2 ± 0.4 | 51.9 ± 0.5 | 40.1 ± 0.6 | 33.4 |
| Mixup | 55.6 ± 0.1 | 18.7 ± 0.4 | 45.1 ± 0.5 | 12.8 ± 0.3 | 57.6 ± 0.5 | 48.2 ± 0.4 | 39.6 |
| MLDG | 59.3 ± 0.1 | 19.6 ± 0.2 | 46.8 ± 0.2 | 13.4 ± 0.2 | 60.1 ± 0.4 | 50.4 ± 0.3 | 41.6 |
| CORAL | 59.2 ± 0.1 | 19.9 ± 0.2 | 47.4 ± 0.2 | 14.0 ± 0.4 | 59.8 ± 0.2 | 50.4 ± 0.4 | 41.8 |
| MMD | 32.2 ± 13.3 | 11.2 ± 4.5 | 26.8 ± 11.3 | 8.8 ± 2.2 | 32.7 ± 13.8 | 29.0 ± 11.8 | 23.5 |
| DANN | 53.1 ± 0.2 | 18.3 ± 0.1 | 44.2 ± 0.7 | 11.9 ± 0.1 | 55.5 ± 0.4 | 46.8 ± 0.6 | 38.3 |
| CDANN | 54.6 ± 0.4 | 17.3 ± 0.1 | 44.2 ± 0.7 | 12.8 ± 0.2 | 56.2 ± 0.4 | 45.9 ± 0.5 | 38.5 |
| MTL | 58.0 ± 0.4 | 19.2 ± 0.2 | 46.2 ± 0.1 | 12.7 ± 0.2 | 59.9 ± 0.1 | 49.0 ± 0.0 | 40.8 |
| SagNet | 57.7 ± 0.3 | 19.1 ± 0.1 | 46.3 ± 0.5 | 13.5 ± 0.4 | 58.9 ± 0.4 | 49.5 ± 0.2 | 40.8 |
| ARM | 49.6 ± 0.4 | 16.5 ± 0.3 | 41.5 ± 0.8 | 10.8 ± 0.1 | 53.5 ± 0.3 | 43.9 ± 0.4 | 36.0 |
| VREx | 43.3 ± 4.5 | 14.1 ± 1.8 | 32.5 ± 5.0 | 9.8 ± 1.1 | 43.5 ± 5.6 | 37.7 ± 4.5 | 30.1 |
| RSC | 55.0 ± 1.2 | 18.3 ± 0.5 | 44.4 ± 0.6 | 12.5 ± 0.1 | 55.7 ± 0.7 | 47.8 ± 0.9 | 38.9 |
| EQRM | 55.5 ± 1.8 | 19.6 ± 0.1 | 45.9 ± 1.9 | 12.9 ± 0.3 | 61.1 ± 0.0 | 50.3 ± 0.1 | 40.9 |

**Averages**

| Algorithm | VLCS | PACS | OfficeHome | TerraIncognita | DomainNet | Avg |
|---|---|---|---|---|---|---|
| ERM | 77.6 ± 0.3 | 86.7 ± 0.3 | 66.4 ± 0.5 | 53.0 ± 0.3 | 41.3 ± 0.1 | 65.0 |
| IRM | 76.9 ± 0.6 | 84.5 ± 1.1 | 63.0 ± 2.7 | 50.5 ± 0.7 | 28.0 ± 5.1 | 60.6 |
| GroupDRO | 77.4 ± 0.5 | 87.1 ± 0.1 | 66.2 ± 0.6 | 52.4 ± 0.1 | 33.4 ± 0.3 | 63.3 |
| Mixup | 78.1 ± 0.3 | 86.8 ± 0.3 | 68.0 ± 0.2 | 54.4 ± 0.3 | 39.6 ± 0.1 | 65.4 |
| MLDG | 77.5 ± 0.1 | 86.8 ± 0.4 | 66.6 ± 0.3 | 52.0 ± 0.1 | 41.6 ± 0.1 | 64.9 |
| CORAL | 77.7 ± 0.2 | 87.1 ± 0.5 | 68.4 ± 0.2 | 52.8 ± 0.2 | 41.8 ± 0.1 | 65.6 |
| MMD | 77.9 ± 0.1 | 87.2 ± 0.1 | 66.2 ± 0.3 | 52.0 ± 0.4 | 23.5 ± 9.4 | 61.4 |
| DANN | 79.7 ± 0.5 | 85.2 ± 0.2 | 65.3 ± 0.8 | 50.6 ± 0.4 | 38.3 ± 0.1 | 63.8 |
| CDANN | 79.9 ± 0.2 | 85.8 ± 0.8 | 65.3 ± 0.5 | 50.8 ± 0.6 | 38.5 ± 0.2 | 64.1 |
| MTL | 77.7 ± 0.5 | 86.7 ± 0.2 | 66.5 ± 0.4 | 52.2 ± 0.4 | 40.8 ± 0.1 | 64.8 |
| SagNet | 77.6 ± 0.1 | 86.4 ± 0.4 | 67.5 ± 0.2 | 52.5 ± 0.4 | 40.8 ± 0.2 | 65.0 |
| ARM | 77.8 ± 0.3 | 85.8 ± 0.2 | 64.8 ± 0.4 | 51.2 ± 0.5 | 36.0 ± 0.2 | 63.1 |
| VREx | 78.1 ± 0.2 | 87.2 ± 0.6 | 65.7 ± 0.3 | 51.4 ± 0.5 | 30.1 ± 3.7 | 62.5 |
| RSC | 77.8 ± 0.6 | 86.2 ± 0.5 | 66.5 ± 0.6 | 52.1 ± 0.2 | 38.9 ± 0.6 | 64.3 |
| EQRM | 77.8 ± 0.2 | 87.3 ± 0.2 | 67.0 ± 0.4 | 52.2 ± 0.7 | 40.9 ± 0.3 | 65.1 |

### G.3 WILDS

In Figure 7, we visualize the test-time risk distributions of IRM and GroupDRO relative to ERM, as well as $EQRM_\alpha$ for select values[10] of $\alpha$. In each of these figures, we see that IRM and GroupDRO tend to have heavier tails than any of the other algorithms.

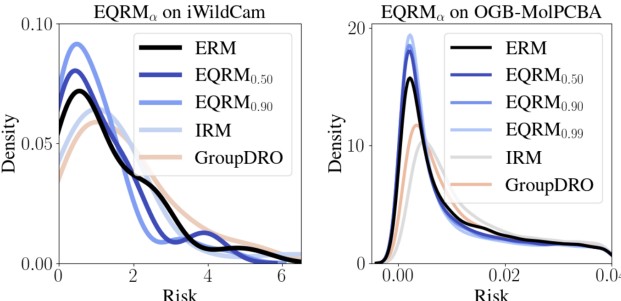

Figure 7: **Baseline test risk distributions on iWildCam and OGB-MolPCBA.** We supplement Figure 4 by providing comparisons to two baseline algorithms: IRM and GroupDRO. In each case, $EQRM_\alpha$ tends to display superior tail performance relative to ERM, IRM, and GroupDRO.

**Other performance metrics.** In the main text, we studied the tails of the *risk* distributions of predictors trained on `iWildCam` and `OGB`. However, in the broader DG literature, there are a number of other metrics that are used to assess performance or OOD-generalization. In particular, for `iWildCam`, past work has used the macro $F_1$ score as well as the average accuracy across domains to assess OOD generalization; for `OGB`, the standard metric is a predictor's average precision over test domains [12]. In Tables 6 and 7, we report these metrics and compare the performance of our algorithms to ERM, IRM, and GroupDRO. Below, we discuss the results in each of these tables.

To begin, consider Table 6. Observe that ERM achieves the best *in-distribution* (ID) scores relative to any of the other algorithms. However, when we consider the *out-of-distribution* columns, we see that EQRM offers better performance with respect to both the macro $F_1$ score and the mean accuracy. Thus, although our algorithms are not explicitly trained to optimize these metrics, their strong performance on the tails of the risk distribution appears to be correlated with strong OOD performance with these alternative metrics. We also observe that relative to ERM, EQRM suffers smaller accuracy drops between ID and OOD mean accuracy. Specifically, ERM dropped 5.50 points, whereas EQRM dropped by an average of 2.38 points.

Next, consider Table 7. Observe again that ERM is the strongest-performing *baseline* (first band of the table). Also observe that EQRM performs similarly to ERM, with validation and test precision tending to cluster around 28 and 27 respectively. However, we stress that these metrics are *averaged* over their respective domains, whereas in Tables 2 and 3, we showed that EQRM performed well on the more difficult domains, i.e. when using *tail* metrics.

Table 6: WILDS metrics on `iWildCam`.

| Algorithm | Macro $F_1$ ($\uparrow$) | | Mean accuracy ($\uparrow$) | |
|---|---|---|---|---|
| | ID | OOD | ID | OOD |
| ERM | **49.8** | 30.6 | **77.0** | 71.5 |
| IRM | 23.4 | 15.2 | 59.6 | 64.1 |
| GroupDRO | 34.3 | 22.1 | 66.7 | 67.7 |
| $QRM_{0.25}$ | 18.3 | 11.4 | 54.3 | 58.3 |
| $QRM_{0.50}$ | 48.1 | 33.8 | 76.2 | 73.5 |
| $QRM_{0.75}$ | 49.5 | 31.8 | 76.1 | 72.0 |
| $QRM_{0.90}$ | 48.6 | 32.9 | 77.1 | 73.3 |
| $QRM_{0.99}$ | 45.9 | 30.8 | 76.6 | 71.3 |

Table 7: WILDS metrics on `OGB-MolPCBA`.

| Algorithm | Mean precision ($\uparrow$) | |
|---|---|---|
| | Validation | Test |
| ERM | 28.1 | 27.3 |
| IRM | 15.4 | 15.5 |
| GroupDRO | 23.5 | 22.3 |
| $QRM_{0.25}$ | 28.1 | 27.3 |
| $QRM_{0.50}$ | **28.3** | **27.4** |
| $QRM_{0.75}$ | 28.1 | 27.1 |
| $QRM_{0.90}$ | 27.9 | 27.2 |
| $QRM_{0.99}$ | 28.1 | 27.4 |

---

[10]We display results for fewer values of $\alpha$ in Figure 7 to keep the plots uncluttered.

# H   Limitations of our work

As discussed in the first paragraph of § 7, the main limitation of our work is that, for $\alpha$ to *precisely* approximate the probability of generalizing with risk below the associated $\alpha$-quantile value, we must have a large number of i.i.d.-sampled domains. Currently, this is rarely satisfied in practice, although § 7 describes how new data-collection procedures could help to better-satisfy this assumption. We believe that our work, and its promise of machine learning systems that generalize with high probability, provides sufficient motivation for collecting real-world datasets with a large number of i.i.d.-sampled domains. In addition, we hope that future work can explore ways to relax this assumption, e.g., by leveraging knowledge of domain dependencies like time.