# OpenReview forum: "Probable Domain Generalization via Quantile Risk Minimization"
_NeurIPS.cc/2022/Conference — NeurIPS 2022 Accept_

### Official Review · Reviewer_xBHw · 2022-07-11

**Rating:** 7
**Confidence:** 3
**Soundness:** 3 good
**Presentation:** 4 excellent
**Contribution:** 3 good

**Summary:**

This work advocates for achieving high-probability generalization across domains, which is in between of the average-case (domain) generaliaztion for which ERM empirically works well, and the worst-case problem which motivates most of previous work in this area.  It assumes the training and test domains are i.i.d. samples from a shared distribution, and proposes to minimize the $\alpha$-quantile and superquantile of the risk distribution of predictors. The authors establish consistency guarantees for the proposed objectives, and connect them to other distributionally robust optimization and invariant prediction formulations.  Experiments show that the proposed method outperforms ERM in terms of the (super)quantile risk.

**Post-rebuttal update**: I have read the authors' response to my and other reviewers' questions and find them satisfactory.  I'm happy to see the new DomainBed experiment, which shows the promise of the proposed method in settings with a smaller number of domain, but the reason for such improvement isn't fully clear.  While I don't believe such discussions are in any sense necessary for the present work (and will not change my score in any case), I think this claim would be better supported if there are more experiments in the small-number-of-domain setting, or some theoretical discussion on the probability of improvement over ERM in such cases.

**Questions:**

- Section 7 claims that we can deploy the trained model to "all [test domains] where it will be successful with probability ≈ α".  How can we identify such domains?  Are you assuming we have a labeled validation set in all test domains?

- It may also be helpful to move some of the IRM / GroupDRO results in appendix to the text.

**Limitations:**

Yes.

**Strengths And Weaknesses:**

The main strength of this work lies in its new formulation; while various relaxiations of worst-case generalization have been proposed before, the new formulation appears easier to implement and analyze, especially for nonlinear models, while still being intuitive and interpretable when its assumptions are adequate.  It thus brings new promise for achieving domain generalization in nonlinear models, as is also evidenced by the experiments.

It should be noted that the method relies on the assumption of i.i.d. domains, and requires that we have a relatively large number of training domains so that quantiles can be estimated to a suitable precision.  The limitations are understandable, given the challenging nature of the problem, and the authors have been very transparent about them, but they will put the contributions better in context.

And while the real-world experiment shows the promise of the method in scenarios where a large number of domains are available, it would strengthen the paper if evaluations are also provided in scenarios where only a handful of domains are available, such as some of the DomainBed benchmarks.

---

> ### Author Response · Authors · 2022-08-02
> **Response to Reviewer xBHw**
>
> Thanks for your positive review of our work!  We have addressed your main points below:
>
>  - *"while the real-world experiment shows the promise of the method in scenarios where a large number of domains are available, it would strengthen the paper if evaluations are also provided in scenarios where only a handful of domains are available, such as some of the DomainBed benchmarks."*
>     - We agree with the reviewer here, and so we have added results on DomainBed (see Table 2).  Encouragingly, on \texttt{PACS} we beat state-of-the-art algorithms across all three model-selection methods, with a notable 2-percentage-point improvement for the cross-validation selection method.  And on \texttt{VLCS}, we match the state-of-the-art, with a slight improvement of nearly 1-percentage-point for the oracle selection method.  For a full comparison across a range of standard baselines with standard deviations across each domain, please see Appendix G.2.
>
>  - *"Section 7 claims that we can deploy the trained model to `all [test domains] where it will be successful with probability $\approx \alpha$'. How can we identify such domains? Are you assuming we have a labeled validation set in all test domains?"*
>     - The wording in this part of the submitted version was confusing, and we have accordingly revised it.  What we meant is as follows: Assume we have labeled training data from sufficiently many i.i.d.-sampled US hospitals, and say we train a predictor with a fixed value of $\alpha$ on this training data.  Then our results guarantee that with probability $\alpha$ over the draw of a new, unseen US hospital, our predictor will perform well on the data from this new hospital.  This is guaranteed by our theoretical results (i.e., Thm.\ 4.1) under the assumptions that (i) the domains are i.i.d.-sampled and (ii) that there are sufficiently many training domains, and (iii) in each of these training domains, we have sufficiently many labeled data-points.
>  - *"It may also be helpful to move some of the IRM / GroupDRO results in appendix to the text."*
>     - We agree, and have accordingly moved the suggested baselines from Appendix C to Section 6 in the main text.

---

### Official Review · Reviewer_yw1U · 2022-07-11

**Rating:** 6
**Confidence:** 4
**Soundness:** 3 good
**Presentation:** 3 good
**Contribution:** 3 good

**Summary:**

This paper introduces a quantile risk minimization (QRM) and its empirical version (EQRM) that aim to learn predictors that can provide minimal risk in a domain generalization with probability $\alpha$. Interestingly, it has been shown in the paper that EQRM turns to the causal predictor when $\alpha \to 1$. The experimental results in the paper show the importance of quantile performance in evaluating domain generalization methods. These results also show that the proposed QRM can produce better results on both real and synthetic data sets compared to some strong baselines.

**Questions:**

Please see my comments on the weaknesses part of the paper.

**Strengths And Weaknesses:**

Originality: The idea of introducing a probable domain generalization (by using quantile risk minimization) is novel and interesting due to the lack of a theoretical guarantee for a generalization bound or learnability of current domain generalization algorithms.

Quality: It seems that the main theoretical contribution of the paper is Proposition 3.1. However, could the authors provide a detailed definition of the distribution $\mathbb T_f$ and a representation of each element $R$ of $\mathbb T_f$?  In the formulation of (QRM), does the condition $Pr_{R \sim \mathbb T_f} \{ R \leq t \} \geq \alpha$ need to be satisfied $\forall R$? Otherwise, it would be very hard for me to verify that statement. More importantly, does QRM or EQRM learn an invariant predictor/representation across domains? If so, how does that work, and are there any theoretical guarantees for that?

Clarity: The paper is well-written and easy to follow.

Significance: As mentioned in the paper, the theoretical result in Proposition 4.2 is only true for linear predictors -- which really restricts the application of the proposed method in domain generalization. Note that the result for an invariant optimal predictor across domains has been introduced in a very well-known paper [1].

[1] @article{arjovsky2019invariant,
  title={Invariant risk minimization},
  author={Arjovsky, Martin and Bottou, L{\'e}on and Gulrajani, Ishaan and Lopez-Paz, David},
  journal={arXiv preprint arXiv:1907.02893},
  year={2019}
}

---

> ### Author Response · Authors · 2022-08-02
> **Response to Reviewer yw1U (Pt 1)**
>
> Thanks for your positive review of our work!  We have addressed your main points below:
>
>  - *"It seems that the main theoretical contribution of the paper is Proposition 3.1."*
>     - Proposition 3.1 is not the main theoretical contribution of the paper---it simply says that setting $\\alpha=1$ recovers the worst-case problem, while setting $\\alpha=0.5$ recovers the average-case problem. Our main theoretical contributions are:
>         - *Theorem 4.1 (and its full version Theorem D.2):* Our main generalization bound establishes consistency guarantees for QRM, giving it a theoretical soundness in the nonlinear setting that is lacking in many existing DG algorithms (IRM, VREx, CORAL, etc.).
>         - *Proposition 4.3 (see updated version of the paper):*  QRM learns an invariant predictor with minimal risk, i.e.\ a minimal invariant-risk predictor, as $\\alpha \\to 1$.
>         - *Theorem 4.4  (see updated version of the paper):* Learning a minimal invariant-risk predictor is sufficient to recover the causal predictor under assumptions which are both weaker and more general than those of Peters et al. (2016, Theorem 2) and Kreuger et al. (2021, Theorem 1). Together, Proposition 4.3 and Theorem 4.4 provide the conditions under which QRM recovers the causal predictor as $\\alpha \\to 1$.
>
>  - *"could the authors provide a detailed definition of the distribution $\mathbb{T}\_f$ and a representation of each element $R$ of $\mathbb{T}_f$?"*
>     - For a more formal definition of $\mathbb{T}\_f$, see Appendix B.1 and in particular Eq. (B.3), where we define $\mathbb{T}\_f$ as the pushforward measure of $\mathbb{Q}$ through a function which -- given a fixed predictor $f$ -- maps an environment index $e$ to the risk $\\mathcal{R}^e(f)$ of $f$ in environment $e$.  In plain words, this means that $\mathbb{T}\_f$ is a distribution over the risks that $f$ can encounter in different environments.  Or, put another way, for any predictor $f$ and $t \\geq 0$, we have $\\Pr\_{e\\sim\\mathbb{Q}}\{[\\mathcal{R}^e(f) \leq t]\} = \\Pr\_{R\\sim\\mathbb{T}\_f}\{[R\\leq t]\}$ (c.f., Eq. (B.4)).  We chose to include these details in the appendix rather than in the main text for ease of exposition, although we would be happy to move it up to the main text if the reviewer feels that it would improve the clarity of our derivation.
>
>
>  - *"In the formulation of (QRM), does the condition $\Pr_{R \sim \mathbb{T}_f} R \leq t \geq \alpha$ need to be satisfied $\forall R$? Otherwise, it would be very hard for me to verify that statement."*
>
>
>    - Let's first rewrite (QRM) in a more intuitive (albeit equivalent) way:
>         $$\\begin{alignat}{2}
>             &\\min\_{f\in\mathcal{F}}  \\min\_{t\in\\mathbb R} && \quad t \\
>             &\quad \\text{subject to}&&\quad \Pr\_{R\\sim\mathbb{T}\_f}\{ [R \\leq t]\} \\geq \\alpha.
>         \\end{alignat}$$
>         The constraint says that with probability at least $\alpha$ over the draw of $R$ from $\mathbb{T}\_f$, it must hold that $R \\leq t$.  Therefore, if $\\alpha=1$, the constraint needs to hold for (almost) every risk $R \\sim\\mathbb{T}_f$.  Otherwise, if $\\alpha < 1$, the constraint need only hold for an $\\alpha$-fraction of the risks $R$ drawn from $\\mathbb{T}\_f$ (or, equivalently, for an $\alpha$-fraction of the environments $e\\sim\\mathbb{Q}$).  In our experiments, we report results for various values of $\alpha$ to show the impact of varying this parameter.  In line with your suggestions, we will add a few descriptive lines after the QRM objective, before Prop. 3.1. We have also added a few more informal lines on Prop. 3.1. to highlight the role of $\alpha$ in this result.
>
>  - *"More importantly, does QRM or EQRM learn an invariant predictor/representation across domains? If so, how does that work, and are there any theoretical guarantees for that?"*
>     - Yes, QRM learns an invariant predictor with minimal risk, i.e.\ a minimal invariant-risk predictor, and we provide theoretical guarantees for this.  We have now moved material on these guarantees---previously in Appendix A.2---to Section 4.1. In particular, Proposition 4.3 states how QRM learns a minimal invariant-risk predictor when using Gaussian and KDE estimators of the risk distribution. The easiest way to see this is to inspect Eq. (A.3) of Appendix A.2.1, observing that $\\Phi^{-1}(\\alpha) \\to \\infty$ as $\alpha \to 1$. Thus, for Gaussian estimators of the risk distribution, QRM eventually puts all its weight on the risk standard deviation over domains as $\\alpha \\to 1$. A similar idea applies to KDE estimators of the risk distribution, as detailed in Appendix A.2.2.

---

> > ### Comment · Reviewer_yw1U · 2022-08-08
> > **Maintaining my original score**
> >
> > Thank you for answering my questions by clarifying the paper's theoretical contributions. I encourage the authors to add those details in the next revisions of the paper.
> >
> > I decide to maintain the original score for the paper.
> >
> > Kind regards.

---

> > > ### Author Response · Authors · 2022-08-08
> > > **Response**
> > >
> > > Thank you for maintaining your positive review of our work. Just to be clear, these details and clarifications have already been included---see lines 138--144, 184--198, and 248--250. We are happy to answer any additional questions that may further improve your score/perception of our work.

---

### Official Review · Reviewer_WQN2 · 2022-07-12

**Rating:** 5
**Confidence:** 4
**Soundness:** 3 good
**Presentation:** 2 fair
**Contribution:** 3 good

**Summary:**

In this work, the authors proposed the Quantile Risk Minimization to have the *probable* DG, where the authors argue that DG should both seek and evaluate predictors based on their ability to perform well with high probability, rather than in the worst-case or on-average, which are widely adopted in concurrent works. The authors also showed that the empirical QRM can recover the causal predictor for the desired generalization probability. The paper is technically sound and generally well-written.

**Questions:**


1. Theorem 4.1 indicates that the number of samples and domains should be sufficiently large. If so, no matter adopting what criteria ('worst case' or 'on average'), the generalization can succeed since we have enough domains and training samples. Thus, I'm wondering what are the exact benefits opt for quantile risk minimization?

2. Following the first question: If the number of samples and domains should be sufficiently large, will the generalization property rely on the model complexity?

3. I would expect more experimental results comparing recent baselines (e.g. the ones provided in DomainBed) on more standard DG datasets (e.g. the PACS, Office-Home, DomainNet etc.).

4. a minor question:  In line 101, it seems that the difference between the assumption of *a distribution over environments* and the general assumption in many concurrent DG works that there exists a meta distribution between domains is not clear. Some explanations or discussions can help.

**Limitations:**

The motivation is not quite clear, and empirical evaluations are not sufficient.
====
After rebuttal comments: I would like to thank the authors for further providing the explanations and experimental results in the rebuttal phase.

**Strengths And Weaknesses:**

Originality: the originality of the paper lies in the proposal of a new objective for the domain generalization problems. I found the contribution fair and somewhat limited.

Strengths:

1. The theoretical analysis was complete, I checked most parts of the appendix, and it seems technically sound.

2. The empirical evaluations were conducted on diverse scenarios including regression, digits and some real-world applications.

Weaknesses:

1. The motivation for adopting quantile risk minimization is not quite clear. The benefits of why opt for Quantile Risk Minimization rather than the 'worst-case' or 'on-average' are not clear.

2. The authors' claim on "Recovering the causal predictor" was not formally elaborated.

3. The empirical comparison was only conducted on a very limited number of baselines. For example, the experiments on ColorMNIST missed many recent baselines, and for the experiments on the *real-world* datasets, only the ERM was compared. This makes the effectiveness of the proposed method hardly to be evaluated.

4. The organization of the presentation needs improvements. Many key components of the paper are listed in the appendix rather than in the main text. The paper itself should be self-contented.

---

> ### Author Response · Authors · 2022-08-02
> **Response to reviewer**
>
> *"I found the contribution fair and somewhat limited."*
>
>  - We are slightly disappointed to read this, but believe it may stem from some misunderstandings which have been clarified below, namely: (i) the motivation for---and significance of---high-probability generalization, in particular how it differs from average and worst-case optimization; (ii) the fact that, while the theory needs many domains, QRM performs very strongly in practice with only a few domains; and (iii) the significance of our causal recovery results.
>  - To further clarify our contributions, we restate them here along with a note on their significance. We:
>     - Propose QRM for learning predictors that perform well with high probability, rather than in the worst case (overly-conservative) or on average (not conservative enough)---see the first point below under ``Weaknesses''.
>     - Provide a generalization bound which establishes consistency guarantees for QRM (Theorem 4.3). While we could have left out this theoretical analysis, and simply presented a practical method with strong empirical results (akin to IRM, VREx, CORAL, etc.), we opted to include a large-sample analysis in order to improve the soundness and theoretical grounding of our method, and believe it constitutes a major contribution.
>     - Prove that QRM learns an invariant predictor as $\\alpha \\to 1$ (Proposition 4.3), and that the causal predictor is often the only (minimal) invariant predictor (Theorem 4.4). These results not only prove that QRM recovers the causal predictor under weaker conditions than previous works, but ultimately establish new fundamental connections between invariance and causality.
>     - Demonstrate the importance of more holistic quantile-focused evaluations in DG, and that QRM performs well in practice---on CMNIST, WILDS, and DomainBed (few domains, newly added).
>
> **Weaknesses:**
>
>  - *"The motivation for adopting quantile risk minimization is not quite clear. The benefits of why opt for Quantile Risk Minimization rather than the 'worst-case' or 'on-average' are not clear."*
>     - We apologize that this was not clear. Given its importance for our method, we will now attempt to clarify the benefits of performing well with high probability, rather than in the worst-case or on-average. In short, optimizing for the worst-case tends to be overly conservative in practice, discarding useful information "just-in-case" we see an extremely-unlikely shift where this information "hurts". In contrast, optimizing for the average-case is not conservative enough, allowing very strong performances on one half of our domains to "cancel out" very weak performances on the other. As we argue in the paper, neither of these fit the purpose of DG. Ultimately, we want a predictor which performs well with high probability (e.g. on 99\% of domains), rather than in the worst case (e.g.\ on shifts with probability $10^{-100000...}$), or on average (performing well on one half but poorly on the other).
>     - As a simple example, imagine we have labeled medical data from 100 US hospitals and then train predictors which perform well: (i) in the worst-case; (ii) on average; and (iii) with high-probability. On deploying these predictors to all 4000 US hospitals, we see that: (i) the worst-case model has quite poor performance, detecting cancer with 60\% accuracy in all 4000 hospitals; (ii) the average model performs well on 3000 hospitals (95\% accuracy) but very poorly on the remaining 1000 hospitals (40\% accuracy); and (iii) QRM with $\\alpha=0.99$ performs well on 99\% of hospitals (80\% accuracy on 3960 hospitals) and poorly on just 1\% of hospitals (50\% accuracy on the remaining 40 hospitals). In this way, $\alpha$ selects different accuracy-robustness trade-offs in an interpretable way, allowing us to meet application-specific requirements on accuracy and/or robustness, e.g.\ a medical system may need to have accuracy $>75$\% on at least $95$\% of hospitals for approval.
>  - *"The authors' claim on `Recovering the causal predictor' was not formally elaborated."*
>     - While the formal elaboration of our causal recovery results was given in Appendix A.2, we agree with the reviewer this elaboration should be in the main text. We have thus moved this formal elaboration up from the appendix---see the highlighted text in Section 4.1.

---

> > ### Author Response · Authors · 2022-08-02
> > **Response to reviewer (pt 2)**
> >
> >  - *"The empirical comparison was only conducted on a very limited number of baselines. For example, the experiments on ColorMNIST missed many recent baselines, and for the experiments on the real-world datasets, only the ERM was compared. This makes the effectiveness of the proposed method hardly to be evaluated."*
> >
> >     - *CMNIST:* we have added SD [1] and IGA [2], resulting in a total of 6 baselines. We believe this is sufficient to illustrate strong performance on CMNIST, particularly since our method approaches oracle performance.
> >     - *WILDS:* we have moved comparisons with IRM and GroupDRO---which were deferred to Tables 7--10 of Appendix C.2---to the main text. We have also added comparisons with CORAL.
> >     - *DomainBed:* We have now added additional empirical comparisons on DomainBed, which allows us to compare our method to 13 of the most recent DG baselines.
> >
> >  - *"Many key components of the paper are listed in the appendix rather than in the main text. The paper itself should be self-contented."*
> >     - On reflection, we tend to agree with the reviewer here. In response, we have moved the following material from the Appendix to the main text:
> >         - Formal explanations and discussions of our causal recovery results from Appendix A.2 to Section 4.1.
> >         - Fig. 4 of Appendix B to Section 3.
> >         - Ex. A.3 of Appendix A to Section 6.1.
> >         - Additional WILDS baselines from Appendix C.2 to Section 6.2.
> >
> >
> > **Questions:**
> >
> >  - *"Theorem 4.1 indicates that the number of samples and domains should be sufficiently large. If so, no matter adopting what criteria ('worst case' or 'on average'), the generalization can succeed since we have enough domains and training samples. Thus, I'm wondering what are the exact benefits opt for quantile risk minimization?"*
> >     - First, while the theory requires many domains, our DomainBed results demonstrate that QRM performs well in practice given few domains. Please see our general response to all reviewers for further discussion and clarification on this point.
> >     - Second, even with a large number of domains, the objective matters---it defines what it means to "succeed", as demonstrated in our hospital example above.
> >  - *"Following the first question: If the number of samples and domains should be sufficiently large, will the generalization property rely on the model complexity?"*
> >     - Indeed, generalization depends both on the number of samples and domains and on the complexity of models being considered. This occurs at two levels:
> >         - The complexity of the predictor model $\\mathcal{F}$ and the number of samples per domain together determine the within-domain generalization error ($\\epsilon_1$ in Theorem 4.1). For example, a large neural network requires more samples to learn than a linear model.
> >         - The complexity of the risk distribution model and the number of domains together determine the probability (($\\alpha - \\epsilon_2$) in Theorem 4.1) that a new test domain will satisfy the generalization bound. For example, a nonparametric risk distribution requires more training domains to guarantee a prescribed generalization probability $\alpha$ than a Gaussian risk distribution.
> >     - Our full generalization bound (Theorem G.1 in Appendix G) explains this in greater detail, with specific terms ($t\_{n,\\delta,\\mathcal{F}}$ and $\\mathcal{N}\_\\epsilon(\\mathcal{G})$, respectively) measuring complexity of the predictor and risk distribution classes.
> >  - *"I would expect more experimental results comparing recent baselines (e.g. the ones provided in DomainBed) on more standard DG datasets (e.g. the PACS, Office-Home, DomainNet etc.)"*
> >     - We have now provided many additional baselines and reported results on DomainBed datasets---see our response above on the number of baselines, under ``Weaknesses''.
> >     - As discussed in L322-331 (orig. submission) and demonstrated in Tables 2--5 on WILDS (orig. submission), comparing DG methods in terms of average performance can be very misleading. For this reason, we had focused our evaluations on large-domain datasets where quantile performance could be evaluated. However, we now agree that providing DomainBed results is useful, and have done so.
> >  - *"a minor question: In line 101, it seems that the difference between the assumption of a distribution over environments and the general assumption in many concurrent DG works that there exists a meta distribution between domains is not clear. Some explanations or discussions can help."*
> >     - We are unsure of which works you are referring to---could you please point us to such works which have an explicit distribution over domains? As far as we know, there are very few works in domain generalization which make such an assumption.
> >
> >
> > We hope that the reviewer now agrees that the motivation is clear and the empirical evaluations sufficient. If so, we kindly ask that they consider adjusting their score.

---

> > > ### Author Response · Authors · 2022-08-02
> > > **Response to reviewer (pt 3)**
> > >
> > > [1] Pezeshki, M., Kaba, O., Bengio, Y., Courville, A. C., Precup, D., \& Lajoie, G. (2021). Gradient starvation: A learning proclivity in neural networks. In *Advances in Neural Information Processing Systems 34*, 1256-1272.
> > >
> > > [2] Koyama, M., \& Yamaguchi, S. (2020). Out-of-distribution generalization with maximal invariant predictor. *OpenReview*.

---

> ### Author Response · Authors · 2022-08-08
> **Discussion phase ends tomorrow**
>
> Dear Reviewer WQN2, we believe that our response addresses your concerns and hope you agree. As the discussion phase ends tomorrow, we would appreciate a brief response indicating whether or not we have indeed addressed your concerns, or if there are any remaining concerns.

---

> ### Comment · Reviewer_WQN2 · 2022-08-09
> **Thanks for the responses**
>
> I've read the authors' rebuttal, and I appreciate the authors' efforts to add more results. However, I'm still not confused on several points:
>
> 1. On the rebuttal of why not *worst-case*, the authors provide an example with not minor probability $10^{-1000...}$ , which I think is not convincing. To my knowledge, the criteria of *worst-case* or *on-average* are still built upon the PAC learning framework; thus, one can tolerate some extreme cases that happen. Even though we adopt the worst-case criteria, we are handling the worst domain, which is assumed to be *not far* from the other domains.
>
> 2. As for the rebuttal on *sufficient domains and samples*, it seems there are inconsistencies between the theory and practice. Besides, the authors' response, "our DomainBed results demonstrate that QRM performs well in practice given few domains." may not be sufficient to answer the question. Since extra DomainBed results were conducted on PACS and VLCS, which are relatively simple (totally of four domains with some domains easily having >90% accuracy); thus the results may not benefit from the theoretical analysis. If possible, I would like to see some further analysis in this respect

---

> > ### Author Response · Authors · 2022-08-09
> > **Further clarifications to your new questions (1/2)**
> >
> > Thank you for your continued engagement with our paper. We sincerely appreciate the feedback you've given thus far. In response to your feedback, we have (a) clarified our contributions, (b) discussed the relationship between PAC learning and QRM (see below), (c) moved material up from the appendix to clarify our theoretical results on recovering the causal predictor, (d) added additional baselines, and (e) added additional datasets.  We hope that you agree that this has strengthened our contribution, and therefore we ask whether you would consider increasing your score.
> >
> > Furthermore, we address each of your new questions below:
> >
> > _"To my knowledge, the criteria of worst-case or on-average are still built upon the PAC learning framework; thus, one can tolerate some extreme cases that happen. Even though we adopt the worst-case criteria, we are handling the worst domain, which is assumed to be not far from the other domains."_
> >
> > We believe the reviewer may be conflating the $\delta$ parameter in PAC learning and our $\alpha$ parameter. These bound the probabilities of *different sources of error*. Specifically, $\delta$ bounds the probability of obtaining a poor or non-representative *training* sample, while $\alpha$ bounds the probability of obtaining an extreme *test* sample. These are very different; for example, typically $\delta \to 0$ as the sample size $n \to \infty$, whereas $\alpha$ does not. Moreover, unlike QRM, we emphasize that **PAC learning provides no guarantee on the proportion of test data for which it performs well---it only bounds _average_ error**. We will add these details to the related work to make clear the difference compared to PAC learning.
> >
> > To provide more intuition, we will now try to elaborate on this point, and would be happy to add these details to the main text.
> >
> > * **PAC learnability for worst-case problems is often impossible.** First things first, we hope that the reviewer agrees that formulating DG as a worst-case problem leads to extreme conservatism, which is undesirable.  One way to see this is through the lens of PAC learning.  For instance, in Lemma 2 of [1], the authors show that having finite VC-dimension is not sufficient to learn a classifier that is optimal for worst case problems.  This is in stark contrast to the Fundamental Theorem of Learning Theory (see, e.g. Thm. 6.7 in [2]), which says that for an average-case problem, finite VC-dimension is equivalent to PAC learnability.
> >
> > * **ERM is not robust to distribution shifts.**  Given that the worst-case problem is too conservative, in this paper, we formulate (QRM), which relaxes this worst-case problem.  The main idea here is to formulate a problem that is more robust than the average-case (e.g. ERM), since algorithms based on the average-case have been shown to generalize poorly out-of-distribution (e.g. as in DomainBed).
> >
> > * **QRM is (much) more robust than ERM, and it still satisfies the PAC learnability criteria.** While it may be unintuitive, for any value of $\alpha<1$, QRM actually satisfies the standard PAC-learnability criteria.  One notable result in this area is Prop. 4.2 in [3], which shows (in the slightly different, though analogous setting of perturbation-based adversarial robustness) that for any $\alpha<1$, (QRM) can match the sample complexity of ERM while providing significantly more robustness.  This result also shows that the corresponding worst-case problem (i.e. $\alpha=1$) can have a sample complexity that is exponentially larger than ERM/QRM.  This is directly in line with our sample complexity bound in Thm. 4.1.
> >
> > * **Summary.**  On the one hand, despite PAC learning having an $\epsilon$ tolerance on the _average_ error, worst-case learning is still impossible in general.  On the other hand, while the average-case learning of ERM is possible, we know from numerous empirical works that it is not robust enough to distribution shifts.  Therefore, from the standpoint of PAC learning, we have argued here that QRM is a PAC learner for DG in the same sense as ERM.  This is backed up by experiments on numerous datasets which demonstrate the robustness of QRM.

---

> > > ### Author Response · Authors · 2022-08-09
> > > **Changing of the goalposts on DomainBed**
> > >
> > > _"As for the rebuttal on sufficient domains and samples, it seems there are inconsistencies between the theory and practice."_
> > >
> > > * **Gaps between theory and practice.**  Firstly, we note that almost every paper has gaps between theory and practice, and in this case we agree that our theory does not completely explain why we get strong results on DomainBed. However, we urge the reviewer to consider that: **(i) no algorithm has theoretical guarantees for DomainBed**; and **(ii) most DG algorithms provide no theory at all in the nonlinear setting, regardless of the number of domains [5,6,7,8]**. In fact, a recent paper shows that **IRM can provably fail** when it is not given access to sufficiently many domains [4].
> > >
> > > * **A lack of theory does not diminish our empirical contribution on DomainBed.** Since **there does not exist an algorithm that has theoretical guarantees on DomainBed**, our strong empirical results on these DomainBed datasets are as strong a contribution as any other paper which has used these datasets. We hope the reviewer agrees that these results are compelling from an empirical point of view, given that QRM outperforms state-of-the-art algorithms, particularly on PACS.
> > >
> > > * **Our theory aligns with the experiments in Section 6.3.** We emphasize that in Section 6.3, we present a complete set of experiments where our assumptions do hold, and in those cases, we demonstrate strong performance relative to each of the baselines we considered.
> > >
> > > * **Changing of the goalposts.** Finally, we would like to remind the reviewer that **you specifically asked to see experiments on DomainBed**:
> > >
> > > > "I would expect more experimental results comparing recent baselines (e.g. the ones provided in DomainBed) on more standard DG datasets (e.g. the PACS, Office-Home, DomainNet etc.)."
> > >
> > > And in response, we provided these experiments.  We ask the reviewer: **If you believed that experiments on DomainBed would not be a relevant contribution to our paper, why did you ask for these experiments in the first place?**  We worked hard during the rebuttal to provide these experiments, and we would appreciate if the reviewer would empathize with our position given your earlier comments.
> > >
> > > [1] Montasser, O., Hanneke, S., & Srebro, N. (2019, June). Vc classes are adversarially robustly learnable, but only improperly. In Conference on Learning Theory (pp. 2512-2530). PMLR.
> > >
> > > [2] Shalev-Shwartz, S., & Ben-David, S. (2014). Understanding machine learning: From theory to algorithms. Cambridge university press.
> > >
> > > [3] Robey, A., Chamon, L. F., Pappas, G. J., & Hassani, H. (2022). Probabilistically Robust Learning: Balancing Average-and Worst-case Performance. In International Conference on Machine Learning.
> > >
> > > [4] Rosenfeld, E., Ravikumar, P. K., & Risteski, A. (2020). The Risks of Invariant Risk Minimization. In International Conference on Learning Representations.
> > >
> > > [5] Arjovsky, M., Bottou, L., Gulrajani, I., \& Lopez-Paz, D. (2019). Invariant risk minimization. *arXiv preprint arXiv:1907.02893*.
> > >
> > > [6] Krueger, D., Caballero, E., Jacobsen, J. H., Zhang, A., Binas, J., Zhang, D., ... \& Courville, A. (2021). Out-of-distribution generalization via risk extrapolation (REx). In *International Conference on Machine Learning* (pp. 5815-5826).
> > >
> > > [7] Sagawa, S., Koh, P. W., Hashimoto, T. B., \& Liang, P. (2019). Distributionally Robust Neural Networks. In *International Conference on Learning Representations*.
> > >
> > > [8] Sun, B., & Saenko, K. (2016). Deep coral: Correlation alignment for deep domain adaptation. In *European Conference on Computer Vision* (pp. 443-450). Springer, Cham.

---

> > > > ### Comment · Reviewer_WQN2 · 2022-08-09
> > > > **Concerns addressed**
> > > >
> > > > I would like to thank the authors for the detailed rebuttal. Most of my concerns are addressed. I'm happy to increase my rating.

---

### Official Review · Reviewer_MTnN · 2022-07-13

**Rating:** 6
**Confidence:** 4
**Soundness:** 3 good
**Presentation:** 2 fair
**Contribution:** 3 good

**Summary:**

The paper presents a probable domain generalisation framework for finding predictors that generalise with high probability instead of generalisation in the worst or average case. It proposes two algorithms: QRM, SQRM, which are based on minimising $\alpha$ quantile of training risks with the interpretation that the predictor so trained would generalise well to at least $\alpha$ fraction of test domains. The last statement holds because the train and test domains are sampled from the same (latent) underlying distribution of domains.

----------
**Post rebuttal**.
I thank the authors for clarifying most of my questions.
The merits of the paper are better understood when there is risk-robustness tradeoff such as in their synthetic setting or the two real-world settings: iWildCam, OGB-MolPCBA. Their synthetic experiment has varying noise levels per environment, and iWildCam likely has varying noise or difficulty level per domain, but I do not see any reason for risk-robustness tradeoff in DomainBed experiments. Therefore, their results on DomainBed are arguable. On CMNIST, they claim to recover causal predictor, as further elaborated in Appendix A, but when the number of training domains is so small as two, $T_F$ estimate is noisy, which is not accounted for in their theorem proofs of Appendix. Besides, in the case of CMNIST, it is unclear why any of the existing methods cannot recover a function with invariant loss across domains. Some of the reviewers (including me) are requesting for a better explanation/intuition on their DomainBed results perhaps because some gains could also arise from implementation differences.
Also, the contribution of SQRM to this paper is unclear. Authors note that it could lead to interesting future work, but I feel it does not contribute much to this work (this is a relatively minor issue though).
This paper has solid contributions, but I believe they could have been expressed more crisply with better experiment design and crisper writing/presentation. For this reason, I am changing my earlier assessment to a weak (but not enthusiastic) accept.

**Questions:**

Please respond to questions marked with (Q) in the section above.

**Ethics Review Area:**

["I don’t know"]

**Limitations:**

Yes.

**Strengths And Weaknesses:**

## Strengths
Probable DG is a very sensible and a practical idea. Training data could include domains with varying difficulty levels and imposing invariance like constraints on them could be detrimental.

## Weaknesses/comments
**Writing**
The writing is unclear and some important information should have been in the main content rather than in the appendix. Let me explain,

* L36-37: (Q) what is “standard DG formulation”, need citations. From the description, it resembles (DG) of L76-77, which inturn resembles DRO.
* L101-110: I do not see why talking about “esssup” is important. It clearly is just DG as remarked in L113: (3.1) is equivalent to DG.
Would help if Preposition 3.1 and eq 3.3 are described more, I could not follow.
* L152-154: Not a good idea to describe in main content a figure in Appendix. Should have been here if it is important.
* L163-165: I do not see why the note on finance is interesting here. Could have been in related work.
* L266-268: “A: each $\alpha$ is optimal at the corresponding quantile” needs further description. I believe the authors are using “optimal” in a different sense?
* L261: “Ex. A.3” which shows SCM of linear setting should have been in the main content.

**Evaluation metric**
The paper advocates to go beyond average or worst accuracy measures for DG and suggest looking at quantiles of test risk/accuracy. Tables 2-5 shows 7 quantiles and average risk. From what I understand, authors judge superior method to be the one that minimises risk on all quantiles, but such a method would also have lower average risk on test domains. So, it seems the authors are going for a more nuanced comparison of quantiles. (Q) They should mention explicitly how quantiles are contrasted to reveal the superior method. In my opinion, evaluation should go hand-in-hand with a human specified quantile level or some such.

**Missing comparisons**
(Q) I expect to see comparisons with other DG methods in Tables 2-5.
(Q) And also std. dev. for all numbers.

**QRM vs SQRM**
The way SQRM is formulated, SQRM is not the same as QRM. Moreover, QRM is never compared with SQRM. In tables 2-5, QRM has consistently lower risk than SQRM. If the contribution of SQRM is efficiency or some such, that should have been discussed. I fail to see the contribution of SQRM.

**ColorMNIST**
(Q) What’s the value of $\alpha$ and how did you pick it?
(Q) As $\alpha\rightarrow 1$, QRM (eq 4.1) is the same as (3.1), which is the same as DRO, correct? (Q) Why does QRM perform so much better than DRO/GrpDRO in Table 1?

**DG formulation**
The paper misrepresents existing DG work. The related work did some justice by having a section on “Robust DG” and “other DG”, but the main content completely disregarded “other DG” and went on to call min-max formulation of risk objective as the standard. This should be fixed.

---

> ### Author Response · Authors · 2022-08-02
> **Reviewer response**
>
> Thank you for your feedback!  We have addressed your main points below:
>
> **Writing:**
>
>  - The standard DG formulation
>     - *"What is 'standard DG formulation', need citations."*
>         - It is common practice [1--6] to formulate domain generalization as optimizing the worst-case performance over some perturbation set of possible test domains $\mathcal{E}_\text{all}$, which gives rise to the minimax formulation in (DG). We have added references [1-6] to the paper for clarity. We also agree that the word "standard" could be misleading as there are certainly other prominent approaches to addressing the domain generalization setting outside of the minimax formulation; we discuss many of these works in our related work section.  To avoid confusion, we have softened our language by changing the word ``standard'' to "common", which is consistent with the language used in past work (see, e.g., $\\S1$ of [4]).
>     - *"Resembles DRO"*
>         - The minimax formulation of (DG) is one of the most general and common ways of specifying the DG problem, wherein the uncertainty set can be used to encode specific assumptions regarding the set of possible all domains $\mathcal{E}_\\text{all}$. In particular, GroupDRO assumes $\mathcal{E}_\\text{all}$ contains all mixtures of the training distributions, e.g., all convex combinations of the training risks [2]. Other works make different assumptions about how the domains in $\mathcal{E}_\\text{all}$ differ, e.g. covariate shift [6,7], label shift [8,9], concept shift [10], measurement shift [11], etc.  And so, in short, we agree that the (DG) formulation does indeed resemble DRO, although it is a more general way of specifying the DG problem.
>  - *"L101-110: I do not see why talking about 'esssup' is important"*
>     - *"It clearly is just DG as remarked in L113: (3.1) is equivalent to DG."*
>         - In L113 we state that (3.1) is equivalent to (DG) *in many cases*. However, as in lines 106–110, (3.1) is the measure-theoretic analogue of (DG), meaning it ignores domains of measure zero under $\\mathbb{Q}$. For example, if we had a discrete distribution over domains $\\mathbb{Q}$, then (3.1) would ignore domains with zero probability of occurring under $\\mathbb{Q}$, while (DG) would not. This ability to ignore zero-probability domains/shifts is critical as it lays the foundation for ignoring low-probability domains/shifts in (3.2) and ultimately in (QRM)---those with a combined probability no greater than $1 - \\alpha$. Thus, (3.1) is not always equivalent to (DG), and their distinction (due to esssup) is important. E.g., $\\mathcal{E}_{\\text{all}}$ contains zero-probability domains for (3.1), or low-probability domains for (3.2).
>         - To clarify this important distinction, we have added another line after (3.1).
>     - *"Would help if Preposition 3.1 and eq 3.3 are described more, I could not follow."*
>         - We have added an additional line before Prop. 3.1 to aid understanding, and slightly updated the language in Prop. 3.1.
>  - *"L163-165: I do not see why the note on finance is interesting here. Could have been in related work."*
>     - We believe that drawing this low-cost connection early is interesting for those familiar with value at risk (VaR) and conditional value at risk (CVaR) in the financial optimization literature. Further connections are indeed discussed in the related work.
>
>  - *"L266-268: 'A: each $\\alpha$ is optimal at the corresponding quantile' needs further description."*
>     - On reflection, we tend to agree with the reviewer that this point is a bit cryptic. We have thus updated it to: *'A: for each true quantile (x-axis), the corresponding $\\alpha$ has the lowest risk (y-axis) '*.
>  - Moving content from the Appendix to the main text.
>     - *"L152-154: Not a good idea to describe in main content a figure in Appendix."*
>         - We agree and have moved this figure (Fig. 4 of Appendix B in original submission) to the main text.
>     - *"L261: `Ex. A.3' which shows SCM of linear setting should have been in the main content."*
>         - We have put Ex. A.3 in the main text.

---

> > ### Author Response · Authors · 2022-08-02
> > **Reviewer response part 2**
> >
> > **Evaluation protocol/metric(s):**
> >
> >  - *"From what I understand, authors judge superior method to be the one that minimises risk on all quantiles, but such a method would also have lower average risk on test domains."*
> >     - It is very unlikely that *any* method would achieve minimum risk on all quantiles, given that there is generally a risk-robustness trade-off. As different methods and parameter settings will perform better at different quantiles, which one is deemed "superior" depends on the application. To this end, with agree with your suggestion below of a human-specified quantile-level-of-interest, and in fact suggested the same in L348--351. We have now added a line on human-specified quantile levels to the final paragraph of Section 6.2.
> >
> >  - *"They should mention explicitly how quantiles are contrasted to reveal the superior method. In my opinion, evaluation should go hand-in-hand with a human specified quantile level or some such."*
> >     - We strongly agree! In fact, this is one of the most important advantages of our method compared to other baselines: due to its interpretability, we can select $\\alpha$ to meet application-specific requirements. As in L348--351: "$\\alpha$ approximates the probability with which our predictor will generalize with risk below the associated $\alpha$-quantile value [...] Such interpretability may aid practitioners meet application-specific requirements on performance and/or robustness". That is, if we know in advance that we care about the $0.95$-quantile, we simply set $\\alpha=0.95$. Moreover, given sufficiently many domains and samples, our generalization bound proves that (E)QRM with $\\alpha=0.95$ will be optimal for this $0.95$-quantile.
> >
> >
> > **Baseline comparisons:**
> >
> >  - *"I expect to see comparisons with other DG methods in Tables 2-5. And also std. dev. for all numbers."*
> >     - Comparisons against IRM and GroupDRO were deferred to Tables 7--10 of Appendix C.2, but we have now brought these back up to the main text. We have also added the CORAL baseline.
> >     - We will add standard deviations over 5 random seeds in the next version (we unfortunately ran out of compute time for the rebuttal phase, focusing our available compute on adding the DomainBed experiments).
> >
> > **QRM vs. SQRM**
> >
> >  - *"SQRM is not the same as QRM"*
> >     - This is correct, they are not the same. As in L193--203, the superquantile is an upper-bound on the quantile and thus acts as an approximation or surrogate with more desirable mathematical properties.
> >
> >  - *"QRM is never compared with SQRM"*
> >     - We have added SQRM to the CMNIST results (Table 1). In the camera ready version, we will offer a complete comparison between QRM and SQRM on WILDS and DomainBed.  However, due to a limited computational budget, we chose to use our computational resources during the rebuttal period to obtain results for QRM on DomainBed, as these experiments were requested by several reviewers.
> >  - *"If the contribution of SQRM is efficiency or some such, that should have been discussed. I fail to see the contribution of SQRM."*
> >     - We appreciate your feedback, and we agree that the contribution of SQRM could be clearer.  The contribution of SQRM is twofold. Firstly, SQRM presents a *convex* alternative to the nonconvex problem in (QRM), which is advantageous from an optimization perspective (although it offers a relaxation of (QRM) which may not be tight). Secondly, it allows us to explicitly connect our problem to existing work.  In particular, we note that, as proved in Appendix F, the optimization problem in (SQRM) satisfies strong duality with the GroupDRO objective [2] (c.f., footnote 4 on page 9 of [2], which alludes to this connection).  This connection facilitates an interpretation of our problem through the lens of existing work, and exploring this connection further is a promising direction for future work.  We have now clarified this in Section 4.2.

---

> > > ### Author Response · Authors · 2022-08-02
> > > **Reviewer response part 3**
> > >
> > > **ColoredMNIST:**
> > >
> > >  - *"What’s the value of $\\alpha$ and how did you pick it?"*
> > >     - As in Appendix D.2, we followed the standard DomainBed procedure of sampling $\\alpha$ from a range of values (see L1088--1090), then selecting the best performing one using test-domain model selection (as is the standard for CMNIST).
> > >     - However, your comments made us reflect on the appropriateness of this DomainBed sampling procedure for CMNIST. In particular, the sampling ranges of other methods were chosen to maximize average performance across *all* DomainBed datasets, while ours were chosen purely for CMNIST. We have thus re-run these experiments with a more appropriate hyperparameter sweep, which in turn has elucidated the importance of ERM pretraining on CMNIST (see Table 1), inline with recent work [12, Table 1]. We thank the reviewer for their question, which sparked these revised experiments.
> > >  - *"As $\\alpha \\to 1$, QRM (eq 4.1) is the same as (3.1), which is the same as DRO, correct?"*
> > >     - No, this is not correct. (QRM) is not the same as (3.1), and (3.1) is not (always) the same as DRO. In regards to the former, (QRM) seeks predictors that generalize with probability $\\alpha$, while (3.1) seeks predictors that generalize with probability 1. In regards to the latter, (3.1) is the measure-theoretic analgoue of (DG), ignoring domains of measure zero under $\\mathbb{Q}$, while DRO is one special case of (DG). See our response to your 'esssup' question above.
> > >  - *"Why does QRM perform so much better than DRO/GrpDRO in Table 1?"*
> > >     - First, as clarified above, these are fundamentally different objectives. Second, as in L151--157 (original submission), smoothing the empirical risk distribution using KDE permits risk "extrapolation" [4] beyond our largest training risk, which ultimately permits invariant prediction---an important generalization capability unavailable to DRO. In Appendix A.2 (orig. subm.), we explained how setting $\\alpha \\to 1$ forces the risk variance over domains to 0, resulting in an invariant-risk predictor. As in L291--292 (orig. subm.), "Note that $\\alpha=0.5$ gives the best worst-case risk over the two training domains—the preferred solution of DRO---while $\\alpha \\to 1$ sacrifices risk for increased robustness or invariance".
> > >     - To clarify this *critical* point in the main text, we have:
> > >         - Allocated a separate paragraph to the importance of KDE smoothing in Section 4.1, bringing up the up the smoothing figure from the Appendix.
> > >         - Expanded the discussion and statement of our causal recovery results in Section 4.1, explaining in more detail how QRM learns an invariant-risk predictor, and the connection to causality.
> > >
> > > If the reviewer agrees that the weaknesses and questions have been adequately addressed, we kindly ask that they consider adjusting their score.

---

> ### Author Response · Authors · 2022-08-08
> **Discussion phase ends tomorrow**
>
> Dear Reviewer MTnN, we believe that our response addresses your concerns and hope you agree. As the discussion phase ends tomorrow, we would appreciate a brief response indicating whether or not we have indeed addressed your concerns, or if there are any remaining concerns.

---

### Author Response · Authors · 2022-08-02
**General response to all reviewers**

We sincerely thank all reviewers for their time and thoughtful feedback.

We were pleased to read that *"Probable DG is a very sensible and a practical idea"* (MTnN) which is *"novel and interesting due to the lack of a theoretical guarantee for a generalization bound or learnability of current domain generalization algorithms"*  (yw1U), with its main strength lying *"in its new formulation [which] appears easier to implement and analyze, especially for nonlinear models, while still being intuitive and interpretable [...]. It thus brings new promise for achieving domain generalization in nonlinear models, as is also evidenced by the experiments."* (xBHw). We were also pleased to read that our paper was *"well-written"* (WQN2, yw1U) and *"easy to follow"* (yw1U), the theoretical analysis was *"complete [and] technically sound"* (WQN2), while the *"empirical evaluations were conducted on diverse scenarios including regression, digits and some real-world applications"* (WQN2).

The main concern of reviewers surrounds the number of domains required for our method (WQN2, xBHw), which has in turn prompted requests for further empirical comparisons (MTnN, WQN2, xBHw). To address this, we have taken the following actions (note that new or revised material is highlighted in yellow in the updated manuscript):

 - **Clarified the domain requirements.** On reflection, we realize that theoretical statements like "given sufficiently many domains and samples", and the fact that most of our experiments contained a large number of domains, may have given the impression that our method requires more domains than existing methods. However, this is not the case, as illustrated by our newly-added results on DomainBed, as well as our previous results on CMNIST. To clarify this in the main text, we have:

    - Added a few lines after our main theoretical result, Theorem 4.1, clarifying that:

        1. While the generalization bound requires many domains, our DomainBed results demonstrate that QRM performs well in practice given few domains.
        2. While many domains are required for $\\alpha$ to *precisely* estimate the true quantile, $\\alpha$ can still be used as an interpretable conservativeness-hyperparameter given only a few domains, albeit with a less precise interpretation.

    - Added a few lines to the related work, clarifying that:

        1. Many of the best-performing DG methods provide no large-sample analyses or theoretical guarantees in the nonlinear setting, e.g.\ IRM, VREx, CORAL, while QRM has both strong theoretical results and strong empirical results.
        2. The few that do provide such guarantees/analyses do so by making strong assumptions about the type of shift that occurs, e.g. covariate shift. We make the weaker and more flexible assumption of iid-sampled domains, which ultimately makes use of the data/domains to determine the types of shifts that are *probable*.

 - **Added more empirical comparisons.**
    - *CMNIST:* added 2 more baselines, bringing the total to 6.
    - *WILDS:* moved comparisons with IRM and GroupDRO---which were deferred to Tables 8--11 of Appendix G.3---to the main text. We also added comparisons with CORAL.
    - *DomainBed:* Added additional experiments on DomainBed, where our method is compared to 13 of the most recent DG baselines.  These results show state-of-the-art performance for QRM on both of the datasets we used: PACS and VLCS.


Another concern shared by reviewers (MTnN, WQN2, xBHw) was that many important items were left to the Appendix. We tend to agree with this point, and have taken the following actions to address it:

 - Causal recovery results and explanations have been moved from Appendix A.2 to Section 4.1.
 - Fig. 4 of Appendix B to Section 3 (KDE smoothing).
 - Ex. A.3 of Appendix A to Section 6.1 (linear SCM).
 - Additional WILDS baselines from Appendix C.2 to Section 6.2.


We believe that these clarifications, additional results, and promotion of Appendix material have improved the paper, and kindly ask the reviewers to take this into account when considering score adjustments. If the reviewers have any further questions or comments, we are very happy to respond during the discussion phase.

---

### Meta-Review · Area_Chair_fwE2 · 2022-08-26

**Recommendation:** Accept
**Confidence:** Certain

**Metareview:**

The authors have convinced the reviewers of the merits of the paper after extensive and detailed rebuttal and discussion. The reviewers have clearly converged towards acceptance.

**Award:**

No

---

### Decision · Program_Chairs · 2022-09-14

Accept